# Effect of changing vegetation and precipitation on denudation (part 1): Predicted vegetation composition and cover over the last 21 thousand years along the Coastal Cordillera of Chile

Christian Werner[1], Manuel Schmid[2], Todd A Ehlers[2], Juan Pablo Fuentes-Espoz[3], Jörg Steinkamp[1], Matthew Forrest[1], Johan Liakka[4], Antonio Maldonado[5], Thomas Hickler[1,6]

[1] Senckenberg Biodiversity and Climate Research Centre (SBiK-F), Senckenberganlage 25, 60325 Frankfurt, Germany
[2] University of Tuebingen, Department of Geosciences, Wilhelmstrasse 56, 72074 Tuebingen, Germany
[3] Department of Silviculture and Nature Conservation, University of Chile, Av. Santa Rosa 11315, La Pintana, Santiago RM, Chile
[4] Nansen Environmental and Remote Sensing Center, Bjerknes Centre for Climate Research, Thormøhlens gate 47, N-5006 Bergen, Norway
[5] Centro de Estudios Avanzados en Zonas Áridas (CEAZA), Raúl Bitrán 1305, La Serena, Chile
[6] Department of Physical Geography, Geosciences, Goethe-University, Altenhoeferallee 1, 60438 Frankfurt, Germany

*Correspondence to:* Christian Werner (christian.werner@senckenberg.de)

**Abstract**

Vegetation is crucial for modulating rates of denudation and landscape evolution as it stabilizes and protects hillslopes and intercepts rainfall. Climate conditions and atmospheric $CO_2$ concentration ($[CO_2]$) influence the establishment and performance of plants and thus have a direct influence on vegetation cover. In addition, vegetation dynamics (competition for space, light, nutrients and water) and stochastic events (mortality and fires) determine the state of vegetation, response times to environmental perturbations, and the successional development. In spite of this, state-of-art reconstructions of past transient vegetation changes have not been accounted for in landscape evolution models. Here, a widely used dynamic vegetation model (LPJ-GUESS) was used to simulate vegetation composition/ cover and surface runoff in Chile for the Last Glacial Maximum (LGM), Mid Holocene (MH) and present day (PD). In addition, we conducted transient vegetation simulations from LGM to PD for four sites of the Coastal Cordillera of Chile at a spatial and temporal resolution adequate for coupling with landscape evolution models.

A new landform mode was introduced to LPJ-GUESS to enable a better simulation of vegetation dynamics and state at sub-pixel resolution and to allow for future coupling to landscape evolution models operating at different spatial scales. Using a regionally-adapted parametrization, LPJ-GUESS was capable of reproducing present day potential natural vegetation along the strong climatic gradients of Chile and simulated vegetation cover was also in line with satellite-based observations. Simulated vegetation during the LGM differed markedly from PD conditions. Coastal cold temperate rainforests where displaced northward by about 5° and the tree line and vegetation zones were at lower elevations than at PD. Transient vegetation simulations indicate a marked shift in vegetation composition starting with the past-glacial warming that coincides with a rise in $[CO_2]$. Vegetation cover between the sites ranged from 13% (LGM: 8%) to 81% (LGM: 73%) for the northern Pan de Azúcar and southern Nahuelbuta sites, respectively, but did not vary by more than 10% over the 21,000 yr simulation. A sensitivity study suggests that $[CO_2]$ is an important driver of vegetation changes and, thereby, potentially landscape evolution. Comparisons with other paleoclimate model driver highlight the importance of model input on simulated vegetation.

In the near future, we will directly couple LPJ-GUESS to a landscape evolution model (see companion paper) to build a fully-coupled dynamic-vegetation/ landscape evolution model that is forced with paleoclimate data from atmospheric general circulation models.

## 1. Introduction

On the macro scale, it has been suggested that sediment yields from rivers exhibit a non-linear relationship with changing vegetation (Langbein and Schumm, 1958). Although this relationship is controversial (e.g. Riebe et al., 2001; Gyssels et al., 2005), previous work highlights that vegetation is likely a first order control on catchment denudation rates (Acosta et al., 2015; Collins et al., 2004; Istanbulluoglu and Bras, 2005; Jeffery et al., 2014). While relatively simple vegetation descriptions have been included in landscape evolution modelling (LEM) studies (Collins et al., 2004; Istanbulluoglu and Bras, 2005), these descriptions do not include explicit representations of plant competition for water, light and nutrients or stand dynamics which are key to determine the progression of vegetation state.

Dynamic Global Vegetation Models (DGVMs) were created as state-of-art tools for representing the distribution of vegetation types, vegetation dynamics (forest succession and disturbances by, e.g., fire), vegetation structure and biogeochemical exchanges of carbon water and other elements between the soil, the vegetation and the atmosphere (Prentice et al., 2007; Snell et al., 2014). Interactions with the climate system have been a special focus, including both transient response to climatic changes and using DGVMs as land-surface schemes of Earth system models (i.e. Cramer et al., 2001; Bonan, 2008; Reick et al., 2013; Yu et al., 2016). DGVMs are instrumental for understanding the impact of future climate change on vegetation (i.e. Morales et al., 2007; Hickler et al., 2012) as well as studying feedbacks between changing vegetation on climate (i.e. Raddatz et al., 2007; Brovkin et al., 2009). In addition, DGVMs have been utilised to better understand past vegetation changes, ranging from the Eocene (Liakka et al., 2014; Shellito and Sloan, 2006) and Late Miocene (Forrest et al., 2015) to the Last Glacial Maximum (LGM; ~21,000 BP) to the Mid Holocene (MH, ~6,000 BP) (i.e. Harrison and Prentice, 2003; Allen et al., 2010; Prentice et al., 2011; Bragg et al., 2013; Huntley et al., 2013; Hopcroft et al., 2017). Using these models, it has been shown that vegetation often responds with substantial time lags to changes in climate (Hickler et al., 2012, Huntley et al., 2013). Such transient changes are likely to influence erosion rates and catchment denudation. Acosta et al. (2015) showed that [10]Be-derived mean catchment denudation rates are lower for steeper but vegetated hillslopes in the Rwenzori Mountains and the Kenya Rift Flanks than the erosion rates for sparsely-vegetated, lower-gradient hillslopes within the Kenya Rift zone.

Jeffery et al. (2014) investigated how interdependent climate and vegetation properties affect Central Andean topography. They found that mean hill slope gradient correlates most strongly with percent vegetation cover, and that climate influences on topography are mediated by vegetation. On a shorter timescale, Vanacker et. al. (2007) determined that the removal of natural vegetation due to land use change increases sediment yield from catchments significantly, while catchments with high vegetation-cover, natural or artificial, return to their natural benchmark erosion rates after reforestation.

Past vegetation changes are however not only the results of changes in climate. The atmospheric $CO_2$ concentration $[CO_2]$ has varied substantially through Earth's history (i.e. Brook 2008) and is an important limiting factor of photosynthesis and plant growth (i.e. Hickler et al., 2015). Glacial $[CO_2]$ of approx. 180 ppm is close to the $CO_2$ compensation point of about 150 ppm for $C_3$ plants (Lovelock and Whitfield, 1982), implying that the majority of all plants on Earth were severely $CO_2$-limited in the LGM relative to today. Vegetation models tend to overestimate the cover of forest during the last glacial if they do not account for the strong limiting effect of $[CO_2]$ (Harrison and Prentice, 2003). Furthermore, changes in $[CO_2]$ also affect stomatal conductance and, thereby, plant water stress, plant productivity, and the hydrological cycle (Gerten et al., 2005). Although the magnitude of so-called "$CO_2$ fertilization effects" is still highly debated (Hickler et al., 2015), physiological effects of $[CO_2]$ might be an important driver of landscape evolution.

While DGVMs are in principle very widely applicable, simulations setups do require modification and calibration for particular applications. For regional applications, DGVMs should be tested against present-day data in the study region and process representations should be adapted to specific conditions (Hickler et al., 2012, Seiler et al., 2014). Climate data for simulations of paleovegetation often originates from Global Climate Models (GCMs), which have a rather coarse grid cell resolution. Hence, spatial downscaling is necessary to derive climatic drivers at an adequate scale.

This study is part of the German EarthShape priority research program (www.earthshape.net), which investigates how biota shapes Earth surface processes along the climatic gradient in the Coastal Cordillera of Chile. Here, we describe the climate data processing and vegetation modelling approach, and report results of simulations for the last 21,000 years (part 1). Specifically, we developed a) a regionally-adapted setup of LPJ-GUESS that also includes improvements in sub-grid representation of vegetation (required for future coupling), b) simulated potential natural vegetation (PNV) for Chile for present day (PD), MH and LGM climate conditions, and c) conducted transient simulations for four focus sites (Fig. 1a) in monthly resolution spanning the full period from LGM to PD. Furthermore, we d) investigate the effect of [$CO_2$] and the use of different paleoclimate data for vegetation simulations of the LGM, and e) explore the relationship between vegetation state, vegetation cover and simulated surface runoff. A companion paper (part 2, Schmid et al., this issue) presents a sensitivity analysis of how transient climate and vegetation impact catchment denudation. This component is evaluated through implementation of transient vegetation effects for hillslopes and rivers in a LEM. Although the approaches presented in these two companion papers are not fully coupled, the results of predicted vegetation cover change derived by our vegetation simulations provide the basis for magnitudes of change in vegetation cover implemented in the companion paper. Together - these two papers provide a conceptual basis for understanding how transient climate and vegetation could impact catchment denudation. As a follow-up to the two presented studies we plan to couple the vegetation and LEMs.

## 2. Background

Climate and vegetation are key controls of surface processes that shape landscapes as precipitation enables the transport of sediment down-slope, while vegetation cover has the ability to protect hillslopes from erosion due to root cohesion, obstruction of overland flow and protection from splash erosion. Vegetation characteristics (i.e. composition and cover, that vary substantially by lifeform, rooting and phenological strategies) are not constant in space and time and vary with climate, topography, and soils. Furthermore, the environmental forcing like temperature, precipitation, radiation and [$CO_2$] change over longer time scales and lead to different vegetation assemblages, potentially differing vegetation cover, and thus protection from erosion.

### 2.1. Climate of Chile

The location of Chile at the Pacific Ocean, its vast meridional extent of 4,345 km, and its steep topographic longitudinal profile (Fig. 1b) result in highly variable climatic conditions. Large-scale subsidence of air masses over the southeast Pacific Ocean and other regional factors (e.g. relative cold coastal ocean currents) yield extremely arid deserts conditions in northern Chile with as little as 2-20 mm of annual precipitation (Garreaud and Aceituno, 2007). In the south, a 1000 km long narrow band of Mediterranean-type climate exists at the western side of the Andes (Armesto et al., 2007). According to a general bioclimatic classification by Luebert and Pliscoff (2017), the tropical/ Mediterranean boundary is located at 23° S (coast) to 27-28° S (inland) and the Mediterranean/ temperate boundary at 36° S (in both Coastal and Andes mountain ranges) to 39° S (Central Depression), while the temperate/ boreal boundary traverses from 50.5° S to 56° S. From a climatologic standpoint, the Mediterranean bioclimatic region has a warm-temperate climate dominated by winter rain (variation in mean annual precipitation from north to south between 300 to 1500 mm, respectively) and hot dry summers with dry periods varying from seven months (north) to less than four months (south) in duration (Uribe et al, 2012). To the south, mid-latitude westerly winds and orographic uplift by the coastal mountains and Andes lead to annual precipitation of up to 3000 mm and 5000 mm, respectively (Veblen et al., 1996). El Niño occurrences generally lead to above average precipitation rates in the austral winter and spring in the Mediterranean zone and reduced precipitation at 38° S to 41° S in the following austral summer (Garreaud and Aceituno, 2007; Montecinos and Aceituno, 2003).

## 2.2. Vegetation of Chile

Considering the ecological divisions of South America (Young et al., 2007), Chile is represented by three noticeable areas: the Peruvian-Chilean desert, Mediterranean Chile, and the Moist Pacific temperate area. They reflect the main floristic characteristics and vegetation types of the country in which the distribution of vegetation is constrained by thermal and hydrological climatic factors that vary according to latitude and longitude (see Table 1 for a mapping of characteristic species to the biome classification and modelled vegetation types). In the North, longitudinal variations in climate are the result of geomorphological and ombroclimatic changes. Between 17° S and 28° S, the coastal zone is exposed to the influence of fog and orographic precipitation, allowing vegetation such as columnar cacti (*Eulychnia* genera) and a diverse group of shrubs (e.g. *Nolana* , *Heliotropium*, *Euphorbia*, *Tetragonia*)  and succulents (e.g. *Deuterocohnia*, *Tillandsia*, *Puya*, *Neoporteria*) to develop (Luebert and Pliscoff, 2017). To the west, and between the Coastal and Andes mountain ranges, a hyper-arid desert zone exists, which is characterized by the absence of rainfall or coastal-fog water inputs, and therefore no vascular plants are usually found. Only with punctual water inputs due to local substrate conditions, such as the presence of a water table, some halophytic shrubs (e.g. *Prosopis* sp) can develop (Luebert and Pliscoff, 2017). Vegetation in the Andes mountain range is constrained by altitude due to the decrease in temperature and the increase in precipitation. The major development in vegetation is observed at intermediate altitudes in which shrubs tend to dominate (Luebert and Pliscoff, 2017).

Vegetation increases to the south with increasing winter precipitation (Rundel et al., 2007), allowing Mediterranean-type shrubland and woodland ecosystems to develop. In these ecosystems various plant species, commonly denominated sclerophyllous plants, have small, rigid, xeromorphic leaves adapted to hot, dry summers and wet, cool winters (Young et al, 2007). Sclerophyllous woodlands and forests extend from 30-31° S to 37.5-38° S (Fig. 1a) and range from xeric thorn savanna elements to dense herbaceous cover in the Central Depression, to evergreen sclerophyllous trees and tall shrubs, at higher elevations in which mesic conditions predominate. In these ecosystems, slope aspect can modify local moisture conditions, affecting the structure and composition of vegetation (Luebert and Pliscoff, 2017). To the South, these vegetation types transition into the temperate deciduous *Nothofagus* forests (*Maule* or *Nothofagus* parklands, dominant species: *Nothofagus obliqua*, *N. glauca* and *N. alessandrii*) at the Coastal and lower Andes ranges, before forming a broader vegetation zone (Donoso, 1982; Villagrán, 1995). The deciduous *Nothofagus* forests then grade into mixed deciduous-evergreen *Nothofagus* forests at 36° S (Young et al., 2007).  With increasingly hydric conditions evergreen broadleaf species begin to dominate forest stands at approximately 40° S and form the Valdivian rainforest (the northernmost rainforest type, ranging from 37°45' S to 43°20' S) with high biomass and arboreal biodiversity and evergreen, deciduous and needleleaf species (Veblen, 2007). Further south, the less diverse North Patagonian rainforest is mainly dominated by *Nothofagus betuloides* (Veblen, 2007) and transitions into the Magellanic rainforest at approx. 47.5° S and Magellanic moorland with water-logged soils and poor nutrition status at the coast (Arroyo et al., 2005). *Cold Deciduous Forests* stretch from 35° S to 55° S along the Andes covering cooler and dryer sites as compared to the coastal rainforests. These forests occur at altitudes of approx. 1300 m and gradually descend to sea level in Tierra del Fuego (Pollmann, 2005). In Tierra del Fuego and east of the low Andes in southern Patagonia a gramineous steppe exists (Moreira-Muñoz, 2011), and high-Andean steppe also extends north at higher altitudes.

## 3. Methods

### 3.1. Vegetation model

The Lund-Potsdam-Jena General Ecosystem Simulator (LPJ-GUESS; Smith et al., 2001, 2014) is a state-of-the-art dynamic vegetation model that also simulates detailed stand dynamics using a gap-model approach (Bugmann, 2001; Hickler et al., 2004). The model is developed by an international community of scientists and has been used in more than 200 peer-

reviewed international publications, including model evaluations against a large variety of benchmarks including vegetation type distribution, vegetation structure and productivity, as well as carbon and water cycling, at regional to global scales (http://www.nateko.lu.se/lpj-guess). Vegetation development and functioning is based on the explicit simulation of photosynthesis rates, stomatal conductance, phenology, allometric calculations, and carbon and nutrient allocation. The model simulates growth and competition of different plant functional types (PFT, see Bonan et al., 2002) based on their competition for space, water, nutrients and light. Population dynamics are then simulated as stochastic processes that are influenced by current resource status, life-history and demography for each PFT. To enable a representative description of average site conditions within a landscape each grid cell is simulated as a number of replicate patches in order to allow different (stochastic) disturbance histories and development (successional) stages (see Hickler et al., 2004; Wramneby et al., 2008). Fire occurrence is determined by the model using temperature, fuel load and soil moisture levels (Thonicke et al., 2001). Soil hydrology in LPJ-GUESS is represented by a simple two-layer bucket model with percolation between layers and deep drainage (see Gerten et al., 2004).

Vegetated surface area and runoff are affected by a range of model parameters in LPJ-GUESS. In 'cohort'-mode, an average individual of each PFT with a given age and development status is used to characterise vegetation state. Depending on PFT-specific parameters (i.e. maximum crown area, sapling density, allometric properties, leaf-to-sapwood area), age and competition for light, water, space, nutrients and demographic processes (establishment and mortality) individual cohorts can develop different states.

Using Eqn. 1, we approximate the fraction A of the land surface covered by vegetation with foliar projected cover (FPC) - the vertical projection of leaf area onto the ground (see Wramneby et al., 2010). In LPJ-GUESS it is derived from daily leaf area index (LAI, leaf area to ground area ratio, $m^2\ m^{-2}$) summed for all simulated PFTs (n: number of PFTs) using Lambert-Beer extinction law (originally proposed by Monsi and Saeki in 1953 for estimation of light extinction in plant canopies, see translation in Monsi and Saeki, 2005; Prentice et al., 1993).

$$A[\%] = \left( 1.0 - \exp\left( -0.5 * \sum_{i=0}^{n} PFT_{LAI} \right) \right) * 100$$

(1)

Thus, depending on the composition of PFTs and the disturbance regime varying levels of ground cover can be simulated that not only reflect the environmental conditions, but also vegetation diversity and development.

In addition, the hydrological cycle is also affected by PFT-specific interception and transpiration rates that are a function of PFT-specific parameters and development stage. Thus vegetation is modulating infiltration via interception (that is a function of vegetation cover) and runoff (via plant uptake and transpiration of water) under the given environmental constraints. In LPJ-GUESS water enters the top soil layer as precipitation until this layer is fully saturated (excess water is lost as surface runoff and evaporation removes water from a 20cm sub-horizon of the top layer). During precipitation days, water can percolate from the top to the lower layer until the lower layer is saturated (excess water is lost as drainage). In addition, water of the lower layer can drain as baseflow with a fixed drainage rate (Gerten et al., 2004; Seiler et al., 2015). The model does neither consider lateral water movement between grid cells nor routing in a stream network (in this study we report the surface runoff component only).

### 3.2. Landform classification

To bridge the gap in spatial resolution between LPJ-GUESS (typical spatial resolution of 0.5°x0.5°) and the LEM LandLab (Hobley et al., 2017; Schmid et al., this issue, typical spatial resolution ~100 m) and so facilitate future coupling of these models, we introduced the concept of landform disaggregation of grid cell conditions to smaller sub-pixel entities (Fig. 2,

Fig. B1). The advantages of introducing sub-pixel entities as opposed to simply performing higher resolution simulations are twofold. Firstly, higher-resolution simulations require climate forcing data of the desired output resolution that are not available for the intended simulation periods and region (and are not generally available for past time periods). Secondly, using sub-pixel entities incurs smaller additional computation costs than higher resolution simulations.

Sub-pixel entities, hereafter termed 'landforms', were derived for each grid cell using SRTM1-based elevation models (Kobrick and Crippen, 2017). Pixels from the elevation model (30 m spatial resolution) were classified based on their elevation (200 m bands) and their association with topographic features (ridges, mid-slope positions, valleys, and plains - based on slope and aspect) and similar pixels were grouped to form the landforms. The elevation of the landforms was used to modify the temperature at a given landform. Using the elevation difference between the average elevation of a landform as

derived from the high-resolution elevation model and the reference elevation obtained from the 0.5°x0.5° grid, a temperature delta was calculated using the lapse rate of the International Standard Atmosphere of -6.5 °C km$^{-1}$ (e.g. Vaughan 2015). However, it has to be noted that this lapse rate is a global average rate that can substantially differ from local conditions and over multiple time scales (ranging from sub-daily to climatological) as it is controlled by various atmospheric therodynamics and dynamics (i.e. radiative conditions, moisture content, large-scale circulation conditions). While a higher lapse would

potentially be a better approximation for drier sites (e.g. Pan de Azúcar), this might not be the case for other sites or past times with different atmospheric conditions and the lack of defining environmental data and a mismatch of scales prohibits the use in our simulations.

The slope and aspect were utilized to adjust the incoming radiation received by the landform (see Appendix B). The general topographic features were used to modify the depth of the lower soil layer (deeper soils at valleys and plains, shallow soils at

ridges) and to identify areas (valleys and plains) with a newly implemented time-buffered deep-water storage pool only accessible by tree PFTs. The classification resulted in 2 to 56 (mean: 17) landforms for each grid cell depending on topographic complexity.

In the proposed future coupled model, we envisage that the landform classification will be performed using elevation information from the LEM. The resulting per-landform (but non-spatially explicit) vegetation simulation results will be

matched back to spatially explicit grid cells in the LEM, thus bridging the scale gap between the two models.

In the simulations presented here, all classified landforms with an area >1 % of the total land area in a grid cell were simulated using 15 replicate patches each, and simulation results were aggregated by area-weighting the results to the grid cell level. For a summary of the implementation details see Fig. B1.

### 3.3. Parametrization of plant functional types and biome classification

In a previous study Escobar Avaria (2013) implemented a first regional simulation for Chilean ecosystems (also using LPJ-GUESS) for present day (PD) climate conditions using a region-specific parametrization, which the presented study adapts and builds upon. Eleven PFTs - three shrub types, seven tree types and one herbaceous type - were defined in order to describe the major vegetation communities of Chile (Table C1). The definition of these PFTs are generally based on the proposed macro-units of Chilean vegetation (Luebert and Pliscoff, 2017) and follow the concept of representative/ or

dominant species for describing a physiognomic unit. Apart from growth habit and associated traits, PFTs were designed to differentiate between leaf morphology and strategy, shade-tolerant and shade-intolerant varieties, their adaption to water access (mesic, xeric type), and root distribution. An overview of major eco-zones, associated PFTs and representative species is given in Table 1.

Using a biomization approach (see Prentice and Guiot, 1996 for the general concept), we classified the simulated PFTs into

discrete vegetation types (referred to as biomes in the manuscript, see Fig. C1 for details of the classification procedure). Based on LAI thresholds and the ratio of certain key PFTs or PFT groups (i.e. boreal tree PFTs, xeric PFTs) to their peers, a cascading decision tree was implemented that leads to eleven biomes resembling the general vegetation zones of Chile (Fig.

1a), but also includes additional biome classes to capture the finer nuances of transitions between semi-arid and mesic vegetation communities and open woodlands (biomes are highlighted with *italic* in the text). To keep the number of vegetation types reasonable we designed multiple decision paths for biomes that exist as dense forest ecosystems but also transition into lower canopy woodlands or transition into more open woodlands (i.e. *Magellanic Forest, Cold Deciduous*

*Forest*; see Fig. C1). The classification was conducted at the landform level after the simulated 15 patches were averaged and the grid cell classification was derived from picking the area-dominant class of the landforms of a grid cell.

### 3.4. Environmental driving data and modelling protocol

The climate forcing data for LPJ-GUESS is derived from TraCE-21ka (Liu et al., 2009), which is a transient coupled atmosphere-ocean simulation from LGM to PD using the Community Climate System Model version 3 (CCSM3; Collins et

al., 2006). In this study we present time-slice simulations for the LGM, MH and PD (1960-1989), using perpetual climate forcing data from 30-year monthly climatologies (Fig. 3). In addition, we show results from a transient model simulation that utilises the full time-series data from the LGM to PD. All simulations were preceded by a 500-year spin-up period with de-trended climate data until vegetation and soils reached steady-state. For time-slice simulations the last 30 years of the simulation were used.

Monthly temperature, precipitation, and downward shortwave radiation from the TraCE-21ka dataset (resolution T31; ~3.7°) were downscaled to 0.5°x0.5° spatial resolution and bias-corrected using a monthly climatology from the ERA-Interim reanalysis (Dee et al., 2011, years 1979-2014). We used an additive bias-correction for the temperature and multiplicative corrections for the precipitation and shortwave radiation (see Hempel et al., 2013; this technique was also used in e.g. O'ishi and Abe-Ouchi, 2011). The multiplicative corrections for the precipitation and radiation are necessary because these fields

cannot have negative values. The resulting bias-corrected anomalies were subsequently downscaled to the ERA-Interim grid using a bilinear interpolation technique. The number of rain days within each month (used by LPJ-GUESS to distribute monthly precipitation totals to daily time steps internally) was derived from the monthly mean precipitation in the TraCE-21ka data and the day-to-day precipitation variability from ERA-Interim (see Appendix A). [$CO_2$] for each simulation year was obtained from Monnin et al. (2001) and Meinshausen et al. (2017). A comparison climate dataset for the LGM (referred

to as ECHAM5 in the manuscript) was provided by Mutz et al. (2018). Soil texture data used for bare-ground initialization of the model was obtained from the ISRIC-WISE soil dataset (Batjes, 2012) and a default soil depth of 1.5 m (0.5 m topsoil, 1 m subsoil) was assumed.

We simulated vegetation dynamics using the process-based dynamic vegetation model LPJ-GUESS (Smith et al., 2001) version 3.1 (Smith et al., 2014) with the model additions outlined above. The model runs were carried out without nitrogen

limitation, using the CENTURY carbon cycle model (see Smith et al., 2014). Patch destroying disturbance and establishment intervals were defined as 100 and 5 years, respectively, and fire dynamics were enabled. Further details of the PFT specific parametrizations are given in Table B1. The transient site-scale model runs were conducted for the four focus sites of the EarthShape SPP only due to i) computing resource constraints, ii) comparability with other EarthShape SPP work (see Schmid et al., this issue), and iii) better interpretability.

### 4. Results

In this study we present data from two types of simulations. First, we show results for time-slice (LGM, MH, PD) simulations. Direct model results (simulated LAI of individual PFTs) are presented first (Sect. 4.1) and then aggregated to a biome representation for easier visualisation and comparison (Sect. 4.2). Then, foliar projected cover and surface runoff are investigated (Sect. 4.3). Then, we present results from the transient LGM-to-PD site simulations (Sect. 4.4) and finally a

sensitivity analysis of the effect of [$CO_2$] levels under LGM climate conditions on vegetation composition and cover (Sect. 4.5).

## 4.1. Distribution of simulated plant functional types

Vegetation communities (expressed as assemblages of PFTs in LPJ-GUESS) establish spatially depending on a) environmental controls, b) competition, and c) stochastic events (i.e. fire incidents and mortality). An overview of the simulated vegetation distribution expressed as the simulated LAI for each PFT under PD climate conditions is given in Fig. 4 (for key PFT properties see Table C1, the PFT distribution maps for MH and LGM are given in the supplemental material as Fig. S1 and Fig. S2 for completeness). Temperate broadleaved evergreen trees (TeBE$_{tm}$, TeBE$_{itm}$; t = shade-tolerant, it = shade-intolerant; m = mesic) dominate the coastal and central areas between latitudes 40° S to 46° S but also extend north into the Mediterranean zone. Further to the north, temperate broadleaved summergreen PFTs (TeBS$_{tm}$, TeBS$_{itm}$) occur, with the shade-tolerant type dominating a relative small area between 37° S and 40° S. Northward, and at coastal areas, the sclerophyllous temperate evergreen PFT (TeBE$_{itscl}$; scl = sclerophyllous) starts to dominate, and with dryer conditions the total LAI (LAI$_{tot}$) is dominated by evergreen and raingreen shrubs (TeE$_s$, TeR$_s$; s = shrub). South of 40° S and, on higher terrain also further north, boreal broadleaved summergreen and evergreen tree PFTs (BBS$_{itm}$, BBE$_{itm}$) as well as boreal evergreen shrubs (BE$_s$) establish. On the Andean ranges, and, to a lesser extent also as secondary components in the lowlands and coastal ranges from 35° S to 50° S, temperate needleleaved evergreen trees (TeNE) are simulated. Herbaceous vegetation (C3G) dominates LAI$_{tot}$ at high altitudes along the Andean ranges and is also a substantial contributor to total LAI in the Mediterranean zone (30° S to 38° S). To a lesser extent it contributes in most other regions but the hyper-arid desert. LAI$_{tot}$ is highest in the zone 36° S to 50° S. While LAI decreases substantially at sea level towards the Atacama Desert, higher values of LAI extend further north at higher altitudes (see inset in Fig. 4).

## 4.2. Distribution of simulated biomes at LGM, MH and present

The simulated biome distribution changes spatially (Fig. 5a), vertically (Fig. 5b) and over time. Under PD climate, the simulated Valdivian temperate rainforests extend from 38° S to 46° S at the coast and transitions into the *Magellanic Forests/ Woodlands*, that are dominated by the boreal PFTs. A small zone of *Deciduous 'Maule' Forest* occurs to the north of the *Valdivian Rainforest* at mesotemperate climates With even dryer and warmer conditions the *Sclerophyllous Woodland* type establishes, and as the fraction of trees and LAI$_{tot}$ is reduced even further with increasing temperatures and even lower annual rainfall, eventually give way to shrub dominated *Matorral* and finally the *Arid Shrubland* type. The *Cold Deciduous Forest* type is classified for parts of Tierra del Fuego, and on higher elevations of the lower Andes in Patagonia. It also forms larger zones at altitude between 30° S and 40° S. *Mesic Woodland* occurs between 34° S and 30° S at altitude (above the *Sclerophyllous Woodland* zone dominating the lowlands) and high-Andean *Steppe* occurs between 18° S and 30° S. A cold desert is present above the tree line in Patagonia and the highest Andean ranges, whereas hot desert (LAI$_{tot}$ <0.2) is simulated for areas from 20° S to 26° S. The model was able to simulate the general distribution of biomes of Chile for most regions (Fig. 1a, 5), but accuracy for the occurrence of deciduous PFTs was low (*Deciduous 'Maule' Forest*, *Cold Deciduous Forest*, also previously reported by Escobar Avaria, 2013).

Owing to the similarity in climatic conditions (Fig. 3), the distribution simulated for the MH does not differ substantially from PD (Table 2). The northern border of *Sclerophyllous Woodland* shifts to approx. 34° S, giving way to a *Matorral* zone. In addition, the *Cold Deciduous Forest* biome covers larger areas in Patagonia at the expense of *Magellanic Woodland* (+27.5%; -9.1%; Table 2). The spatial and vertical distribution of biomes for the LGM is however markedly different (Fig. 5a,b). The substantially lower temperatures (Fig. 3) lead to an expansion of cold deserts up to 45° S (coastal areas) and 40° S (higher altitudes), respectively. The boreal PFT dominated *Magellanic Woodland* biome is substantially reduced in extend (16.5% (LGM) vs. 32.6% (PD) of simulated area, Table 2) and shifted further north (40° S to 45° S at the coast, 43° S to 35°

S at higher altitudes inland). The area covered by temperate rainforest is restricted to a small lowland area from 36° S to 40° S, and the larger areas of *Cold Deciduous Forest* at altitude are also substantially smaller (Fig. 5b; Table 2). Lowland *Steppe* and *Mesic Woodland* biomes are simulated instead of *Matorral* and *Sclerophyllous Woodlands* and desert covers larger areas of the high-Andes to the north.

**4.3. Foliar projected cover and surface runoff**

The percentage of ground covered, and thus shielded from strong denudation, and surface runoff (as a major driver of erosion rates) are both influenced by the composition and state of vegetation communities. We therefore evaluate the regional and temporal changes of these important variables as simulated by LPJ-GUESS for the LGM, MH and PD time slices. The simulated LAI of PFTs can be aggregated and converted to foliar projected cover (see Eq. (1)) and thus allows us to estimate the surface covered by vegetation. It should however be noted that this is only an approximation of true ground cover as small-scale vegetation variations are not simulated location-specific (spatial lumping effects are not considered for instance). However, the implementation of a sub-grid landform scale was, in part, motivated to improve the models' prediction of smaller scale differences as it should allow to differentiate different sub-grid conditions for the major landforms within one simulation cell (see Sect. 5.1 and Appendix B for further information). Low FPC clearly coincides with distribution of hyper- to semi-arid biomes (Fig. 5a, 6a). Under PD climate conditions, average FPC for the semi-arid and Mediterranean biome types *Arid Shrubland*, *Matorral*, and *Sclerophyllous Woodland* are 16%, 35%, and 66%, respectively (Table 3) and cover increases southward with increases in annual precipitation rates (Fig. 3b). South of 35° S, FPC values >70% are simulated for most grid cells except for high-altitude locations, the glacier fields of North Patagonia, and parts of the Magellanic moorland at the coast (see also Table 3).

Simulated FPC was lower than satellite-based estimates by MODIS Vegetation Continuous Field product (Dimiceli et al., 2017), likely due to methodological differences of foliar projected cover and total satellite-observed vegetation cover, but the general patterns were represented (Fig. 7). The most distinct regional discrepancy can be observed at coastal areas between 30° S to 36° S and the Andean highlands in the north of the model domain. For the entire model domain the mean average error (MAE) of FPC was 11.9%. Best agreements of 6.1% and 6.3% MAE were achieved for the *Arid Shrubland* and *Valdivian Rainforest* biomes, while the discrepancies were greatest for *Mixed Forest* and *Cold Deciduous Forest* biomes with 22.4% and 22.1% MAE, respectively (see Fig. 4a for biome locations, note that LPJ-GUESS often tends to underestimate FPC of deciduous PFTs). A strong positive correlation between annual rainfall and FPC can be observed for all semi-arid, Mediterranean, and seasonally dry temperature eco zones (Fig. 8a) and increases in annual rainfall lead to a strong rise of ground covered until 250-300 mm rainfall per year (*Arid Shrubland*: +14.5–16.3% 100 mm$^{-1}$; *Matorral*: +12.9-16.5% 100 mm$^{-1}$; *Steppe*: +13.5-17.8% 100 mm$^{-1}$; Fig. 8a). At these levels, Mediterranean and temperate woodland biomes start to dominate but increases in precipitation raise FPC only by +2.7-3.3% 100 mm$^{-1}$ and +5.4-6.4% 100 mm$^{-1}$ (*Sclerophyllous Woodland, Mesic Woodland*).

Similar spatial pattern of FPC can be observed for the MH (Fig. 6a) and for the relationship to precipitation rates (Fig. 8a). No significant difference in FPC to PD conditions are apparent for the Atacama Desert and most temperate forest areas from 36° S to 42° S. FPC at high-Andes locations of the north and large parts of the *Magellanic Forest* and *Cold Deciduous Forest* biomes in the south is 10-20% lower than under PD climate, whereas for the Mediterranean zone FPC was reduced by 5-10% (Fig. 6b, Table 3). The reduction of MH FPC (47° S - 54° S) coincides with a zone of lower temperature (> -0.5 °C, see Fig. 6b) and the areal extent of TeBE PFTs (Fig. S1). Lower [$CO_2$] at MH compared to PD might also be the reason for a general reduction of biomass productivity but since [$CO_2$] levels of all regions are the same at any time might be compensated by differences of the other climate drivers.

Due to lower temperatures and reduced precipitation rates at high altitudes and in the southern part of the country (Fig. 3), FPC at LGM in these areas is simulated to be strongly reduced (< -30%, cold desert conditions for large areas south of 46°

S) but FPC was lower for all areas of Chile (Fig. 6b, Table 3) – likely also the effect of substantially lower $[CO_2]$ (see Sect. 4.5 for a sensitivity analysis of the effect of $[CO_2]$). While the general correlation of FPC to precipitation can also be observed for LGM (Fig. 8a), the variability in vegetation cover in mesic and xeric woodlands appears to be larger – indicating the potential for greater variabilities of erosion rates within the same biome. The strongest correlations between annual precipitation and FPC were observed for *Sclerophyllous Woodland* ($r^2$-adj. 0.84, 0.91, and 0.9 for LGM, MD, and PD respectively; $p < 0.001$).

Annual average runoff varies greatly from north to south (Fig. 6c) and coincide with annual precipitation (Fig. 3, 8b). For PD and MH climate, LPJ-GUESS simulated almost no surface runoff for arid and Mediterranean areas of Chile to approx. 32° S (see also Table 3). Thus, correlation between precipitation and runoff in the *Steppe* and *Arid Shrubland* biomes was low (adj. $r^2$: 0.16-0.27, 0.12-0.37, 0.11-0.35 for LGM, MH, and PD respectively; $p < 0.01$). For all other systems (excl. *Matorral and Mixed Forests*), correlation coefficients were generally > 0.85 for all time-slices; $p < 0.001$.

Runoff rates increase gradually southward and reach their peak (> 2000mm $yr^{-1}$) in areas of hyper-humid conditions along the Pacific coast. MH runoff rates are higher for areas of the Northern Patagonian and Magellanic rainforests (40° S – 46° S), but lower for coastal areas of the Magellanic moorlands (Fig. 6d). LGM runoff rates are higher for most areas (Table 3) and especially south of 34° S, with strongest differences occurring from 40° S to 46° S. However, we want to highlight that the low temporal variability of the TraCE-21ka precipitation data likely leads to a substantial underrepresentation of episodes of high hygric variability (see discussion).

### 4.4. Transient changes of simulated vegetation from LGM to present

In this section we present transient simulation results for grid cells that contain the EarthShape focus sites (Fig. 9). The results are given for a single landform of the simulated grid cells in order to preserve successional transitions between PFTs that might otherwise be lost through averaging. The field site location and the represented area of the chosen landform within the grid cell are marked in the insets of Fig. 9a-d. The diversity of simulated vegetation cover for all landforms of the four EarthShape focus sites is illustrated in Fig. B2. The area-averaged mean of landform enabled LPJ-GUESS simulations closely matches the default model results at sites Sta. Gracia and La Campana, but FPC at sites Pan de Azúcar and Nahuelbuta is approx. 3% greater than the default simulation result (Fig. B2a). However, results from all four sites show that inter-landform variability of FPC varies by 10%, 15%, 25% and 18% for sites Pan de Azúcar, Sta. Gracia, La Campana and Nahuelbuta, respectively.

Temperatures and $[CO_2]$ start to increase at 18,000 BP and a marked pull-back in temperatures during the Antarctic Cold Reversal (~14,500 BP) is present in the TraCE-21ka data for all four sites (Fig. 9a-d). Annual precipitation at the site Pan de Azúcar (21.11° S 70.55° W, 320 m a.s.l.) is extremely low (38-40 mm, hyper-arid) for the entire simulation period (Fig. 9a). Annual average temperature for the presented landform increases from 13.9 °C (LGM) to 16.8 °C (PD). As a result of these arid conditions only evergreen and raingreen shrubs and herbaceous vegetation can establish, and LAI, and consequently FPC, remains very low ($LAI_{tot}$ <0.3). For most episodes of the simulation this location is classified as desert according to the implemented biomization scheme, with only two periods switching to another state (*Arid Shrubland*, $LAI_{tot}$ >0.2) at approx. 17,000 to 15,000 BP, and more permanently, the late Holocene. Fire return intervals (expressed as number of years between fire incidents) fluctuate greatly as fuel production is substantially limited by low vegetation growth. No surface runoff is simulated and FPC ranges from 8% (LGM) to 13% (PD).

Climatic conditions for site Sta. Gracia (29.75° S 71.16° W, 579 m a.s.l.) show a similar temporal progression from LGM to PD, but average temperatures are lower and range from 11.9 °C (LGM) to 14.9 °C (PD) (Fig. 9b). Annual precipitation does not change substantially from LGM to PD (152 vs. 122 mm). Vegetation from LGM to approx. 18,000 BP is dominated by herbaceous vegetation, but (raingreen) shrubs increase with rising temperatures, leading to a biome shift from *Steppe* to

*Matorral*. Fire return intervals decrease with increased (arboral) litter production due to encroachment of shrubs but FPC only increases by 7% from 33% (LGM) to 40% (PD). Simulated surface runoff is insignificant for most simulation years.

Temperatures at the La Campana site (32.93° S 71.09° W, 412 m a.s.l.) increase from 11.0 °C (LGM) to 14.0 °C (PD), but annual precipitation amounts decrease from 446 mm (LGM) to approx. 320 mm slightly increasing again to 355 mm at PD.

The simulated LGM vegetation for site La Campana is dominated by herbaceous vegetation and small fractions of temperate evergreen deciduous trees mixed with small fractions of boreal shrubs leading to a steppe biome classification with short episodes of mesic woodlands (Fig. 9c). With the decrease of annual precipitation and increasing temperatures (approx. 17,500 BP) *Sclerophyllous Woodlands* displace the deciduous trees and evergreen and raingreen shrubs start to appear. The fire-return intervals shorten in this phase and reach values of less than 15 years for the remaining simulation. Raingreen

shrubs expand at approx. 12,000 BP and push back on herbaceous vegetation and, in part, evergreen shrubs. The LAI of sclerophyllous broadleaved evergreen trees and shrubs increases further during the last 5,000 simulation years, which leads to a shift in our biome classification from *Matorral* ($LAI_{tot}$ >0.5) to *Sclerophyllous Woodland* ($LAI_{woody}$ >1). Despite pronounced changes in vegetation composition, FPC only increases from approx. 51% (LGM) to 59% (PD), which translates to a relative stable vegetation cover for these regions over time and thus likely a low effect of biome shifts on erosive

processes.

Climatic conditions at the site Nahuelbuta (37.81° S 73.01° W, 1234 m a.s.l.) are markedly different from the three previous ones, as this location receives substantially higher annual precipitation throughout the time-series (> 1200 mm) and average temperatures at this latitude and elevation are substantially lower (5.1 °C for LGM and 8.6 °C for PD, Fig. 9d). Note however, that the landform is only representative for a small fraction of the 0.5°x0.5° grid cell as it is located on

mountainous terrain, whereas most areas in the cell are covered by coastal lowlands with higher annual temperatures and thus the site simulation results presented here differ from the total grid cell results presented in previous sections (see marked landform cover in inset, Fig 9d and Fig. B2a,b). LPJ-GUESS simulates a diverse composition of PFTs and a transition from boreal, *Magellanic Woodland* conditions at LGM, to a period of *Cold Deciduous Forest* (17,500 - 12,000 BP), followed by 12,000 years of *Valdivian Rainforest* and *Mesic Woodland* alternations. During the LGM, boreal broadleaved evergreen,

deciduous tree and shrub PFTs dominate and form a forest. Annual precipitation (>1300 mm) and surface runoff (>440 mm) is high during that period. With rising temperatures, the boreal shrubs and evergreen tree PFTs are displaced by temperate needleleaved evergreen trees (TeNE) and increases in herbaceous vegetation. Temperate evergreen PFTs establish approx. 17,000 BP and, after another retreat at approx. 14,500 BP (coinciding with the Antarctic Cold Reversal), start to dominate the forest at this location. Fire frequency is low for the first 4,000 simulation years and only rises to approx. one fire in 100

years afterwards. FPC remains at constantly high values (>75%) indicating a largely closed forest for the entire simulation period and thus low dynamics of erosive processes due to constant high vegetation cover in this area.

## 4.5 Sensitivity of foliar projected cover to [$CO_2$]

The effectiveness of photosynthesis, and thus the plants ability to build-up biomass, has a large dependence on [$CO_2$] (Farquhar et al., 1980; Hickler et al., 2015). Increases in vegetation biomass (expressed in LAI) were observed in our

simulations but coincide with simultaneous rise of temperatures and [$CO_2$] (Fig. 9b-d). To assess the direct effect of changes in [$CO_2$], we conducted a sensitivity simulation under LGM climate conditions. We compared the default LGM simulations (TraCE-21ka, [$CO_2$] = ~180 ppm) with a pre-industrial [$CO_2$] of 280 ppm (Fig. 10). This change leads to an expansion of vegetated areas (high-Andean steppe), an increase of forest biomes at the expense of herbaceous vegetation (i.e., northward expansion of *Mesic Woodland*, *Cold Deciduous Forests*, and *Valdivian Rainforest* and the establishment of small pockets of

*Sclerophyllous Woodland*) (Fig. 10a). In all vegetated areas FPC increases with higher [$CO_2$], most notably between 30 - 40° S (+ 5-10% FPC).

## 5. Discussion

The aim of this study was to demonstrate that a dynamic vegetation model with suitable modifications can simulate the state and transient changes of vegetation structure and composition at a temporal and spatial resolution that is suitable for coupling with LEMs that operate on higher spatial resolutions. To bridge the spatial scales and retain a high computationally efficiency, we introduced the sub-gridcell landform types in the DGVM LPJ-GUESS. Using this novel approach we are able to better reproduce observed spatial heterogeneity with existing DVMs. Furthermore, a regional parameterisation of Chilean vegetation allowed us to simulated the present-day vegetation of Chile (Fig. 1a, 4, 5, 7; see also details in Sect. 5.1 below). In our PD simulations we found the largest regional discrepancy of observed and simulated vegetation cover to occur at coastal areas between 30° S to 36° S, which might be attributed to the lack of fog-precipitation in our model. Fog precipitation is strongly dependent on distance to the coast, local topography, wind fields, and stratification of the troposphere (Lehnert et al., 2018) and can potentially contribute significant amounts of precipitation (Garreaud et al., 2008). However, a model representation of fog in LPJ-GUESS is difficult due to the scale mismatch and a lack of required input variables to determine the occurrence. Second, precipitation amounts in the model drivers might be too low as the coarse spatial resolution of the original input data leads to an underrepresentation of orographic precipitation effects (i.e. Leung and Ghan, 1998).We also want to note that the MODIS VCF product was found to overestimate cover in sparsely vegetated areas (Sexton et al., 2013) and thus should rather be treated as a general guideline. Furthermore, LPJ-GUESS was applied to simulate the PNV whereas MODIS VCF observes actual vegetation that includes anthropogenic land-use (agricultural fields, degradation due to grazing, etc.).

Substantial changes in fire frequencies were observed in the transition simulations (Fig. 9b,c). LPJ-GUESS simulated those dynamics by considering available fuel which, in-turn, is a function of present PFT composition and litter production. These rapid removals of vegetation cover have the potential to expose bare soil to erosion process and could be critical in coupled simulations (a feature usually not considered in traditional landscape evolution modelling setups, i.e. Istanbulluoglu and Bras, 2005).

LPJ-GUESS explicitly simulates the soil moisture available for vegetation through a simple hydrological cycle (Gerten et al., 2004), and it is thus possible to calculate changes in runoff due to changes in transpiration, evaporation and percolation at a site. Such information, in particular surface runoff, may also be provided to a LEM in a coupled model configuration. In this initial stage of our studies we did not yet use surface runoff in the LEM (see Schmid et al., this issue), but we envision that it will be incorporated in the planned coupled DVM-LEM model after proper evaluation. We want however caution the reader that the model used in this study uses a simple two-layer bucket model which cannot truly represent catchment scale hydrological features (i.e. no lateral flow, routing, or variable water table) but has been successfully coupled to mechanistic hydrological models (Pappas et al., 2015).

### 5.1 Effects of landform simulation mode

As mentioned previously, the novel landform approach illustrated in this study (see Appendix B) was motivated by the fact that standard DVM do not account for sub-grid heterogeneity controlled by topography and micro-climatic conditions. A common solution is to run these models at the same spatial resolution as a LEM, but this is generally computationally wasteful and often violates the scale assumptions of DVMs (i.e. patch size assumption in LPJ-GUESS, see Appendix B). The most striking advantage of the landform implementation regarding the realism of simulated FPC is that it allows to simulated sub-grid topographic units (i.e. valleys, ridges, slopes) in a computationally efficient way as similar topographic locations are grouped to landforms and only simulated once (or rather n times for the patches of a given landform, see Fig. B1). North and south facing landforms receive a modified amount of solar radiation depending on their average slope and aspect orientation (affecting photosynthesis and transpiration and thus also water availability) and previous studies have showed that this can lead to distinct differences (i.e. Sternberg and Shoshany, 2001). Variable soil depth of the landforms is associated with

topographic position (ridges are assumed to feature more shallow soils while valleys have deeper soils due to alluvial material) thus also providing differing water storage potential that can especially relevant in dryer locations (Kosmas et al., 2000). Local temperature differences are also considered as higher landforms are exposed to colder temperatures depending on the elevation difference to the reference elevation of the grid cell (see Appendix B). This leads to a more diverse PFT composition within a grid cell as altitudinal zonation of vegetation within a grid cell can be represented (see Fig. 5b and Fig. B2b).

However, it has to be noted that we currently do not account for precipitation variations in the landform approach as a good representation of these effects would require many additional model drivers (i.e. wind speed and direction, sea surface temperature for local fog precipitation, see Gerreaud et al., 2008, 2016; Lehnert et al., 2018) that are either not available at appropriate resolutions or for the time periods considered in this study. Future work may try to incorporate sophisticated statistical downscaling schemes (i.e. Karger et al., 2017) to address this shortcoming.

As can be observed (Fig. B2a), the implemented landform approach has differing effects for the four focus sites. The area-weighted average FPC at site Sta. Gracia closely resembles the results from the default simulation mode (apart from landform 810 of high altitudes that only covers 1.7% of the grid cell area, Fig. B2b). Average-landform and default results for site La Campana also differ only marginally. However, here a set of landforms of higher altitudes has a substantially lower FPC than the average (~ - 15%). Variation at the hyper-arid site Pan de Azúcar is lower (as is the FPC), but generally higher than the default simulation. The larger vegetation cover simulated sing the landform approach also better aligns with MODIS observations for the site, where the default model underestimates satellite-observed cover. The higher FPC in the new model setup is likely a result of deeper soil profiles of flat and valley landforms that allow a longer water storage versus the default uniform 1.5m soil assumption of LPJ-GUESS. The larger variability of FPC at site Nahuelbuta can be attributed to the relatively large altitudinal variation in this grid cell (coast to mountainous terrain) and is thus likely a temperature effect. From these results it can be postulated that the stronger heterogeneity of simulated FPC using the landform approach will lead to more diverse denudation rates for the topographic units when linked to a LEM and should better resemble observed patterns of vegetation associations in the landscape (we will investigate this in detail in a future study of a coupled DVM-LEM setup).

In order to account for micro-site differences of temperature versus the average climate information per grid cell we stratified the landforms into elevation bins (Sect. 3.1 and Appendix B). Site temperature is an important environmental control for LPJ-GUESS as it impacts vegetation establishment (controlled for individual PFTs by their specific bioclimatic limits), photosynthesis rate, evapotranspiration, and soil decomposition (Smith et al., 2014). We selected 200m as the bin width after a sensitivity analysis to keep the total number of landforms in complex terrain reasonable (potential temperature difference between 100m and 200m bins < 0.3 °C; data not shown). Furthermore, using more topographic units (separate slopes into lower, mid, and upper slopes) did also not affect the simulation outcome and we thus opted for a 4-unit classification (see Appendix B). These configurations are subjective and should be tested before application und very different terrain or model scales.

## 5.2. Comparison to vegetation change proxy data

In general, regional vegetation cover of the past is difficult to quantify, as there are limited site-scale pollen, lake level, and midden proxy datasets available in Chile (Marchant et al., 2009). Pollen data from rodent midden (Quebrada del Chaco, 25.5° S, 2670-3550 m, Maldonado et al., 2005) suggest higher winter precipitation at the LGM, higher annual precipitation at 17-14 ka BP and higher summer precipitation at 14-11 ka BP in northern Chile, but results from another study at lower elevations indicate absolute desert conditions throughout the quaternary (Diaz et al., 2012) - which could be caused by regional to fog-precipitation. TraCE-21ka precipitation for site Pan de Azúcar (located at similar latitudes but at coastal lowlands) does not vary from PD conditions throughout the time series and as a result LPJ-GUESS simulates very little

vegetation cover (Fig. 9a). Other studies also suggest that LGM conditions were extremely dry, but that the transition to modern climatic conditions in this area occurred as a non-linear process of multiple moisture pulses over several centuries (Grosjean et al., 2001). This would imply that a coupled DVM-LEM model would thus underestimate episodes of high erosion potential due to poor model forcing data.

Pollen reconstructions from Laguna de Tagua Tagua 34.5° S 71.16° W (Heusser, 1990; Valero-Garcés et al., 2005) indicate the presence of extensive temperate woodlands at the LGM that match the simulated *Mesic Woodland* biome for this location in our simulations (Fig. 4a). Multiple lake sediment records of the region indicate dry and warm conditions at MH and the onset of more humid, but strongly seasonal, conditions (winter rainfall) in the late Holocene (Heusser, 1990; Villa-Martínez et al., 2003; Valero-Garcés et al., 2005). While LPJ-GUESS does simulate Mediterranean vegetation types (*Matorral* and

*Sclerophyllous Woodland*) for the MH and a shift to denser xeric woodlands in the late Holocene for this region, substantial variations in the precipitation regime as suggested by the lake records are not present in the TraCE-21ka data used.

Pollen records from the Chilean Lake District show a warming trend starting at 17,780 BP (Moreno and Videla, 2016), followed by a trend reversal with major cooling events (14,500 and 12,700 BP). In TraCE-21ka, a sharp cooling event at 14,500 is present (likely as an effect of the simulation setup of Meltwater Pulse 1A, Liu et al., 2009), but the second event is

not. In our transient simulations this event is reflected in changes of vegetation composition at three out of our four sites. This climatic change to colder and dryer conditions leads to a reduction of shrub PFTs at sites Sta. Gracia (Fig. 9b) and La Campana (Fig. 9c) that established with rising temperature and [$CO_2$] starting 18,000 BP. As a result, the site biome classification for Sta. Gracia briefly swings back to the initial *Steppe* state and the reduced fuel accumulation leads to fewer fires. Surface runoff is however not affected due to the low precipitation rates. In contrast, the same event at site La

Campana does not result in changes of the fire frequency probably due to the presence of sufficient fuel due to higher annual rainfall rates (>350 mm). At site Nahuelbuta this event briefly delays a transition from cold deciduous PFTs to a temperate evergreen/ needleleaved forest (Fig. 9d).

The early and mid-Holocene were the driest periods for central Chile (Heusser 1990; Valero-Garcés et al., 2005; Maldonado and Villagrán, 2006). This is also visible in the TraCE-21ka dataset (La Campana and Nahuelbuta, Fig 9c,d). However, the

annual precipitation differences in the TraCE-21ka data of this period compared to LGM and PD are relatively minor and we thus might not fully capture the true vegetation dynamics indicated by the proxy data of the region (centennial shifts from dry xeric to humid mesic vegetation).

Pollen proxy data by Villagrán (1988) indicates that coastal cool evergreen rainforests were shifted by 5° northwards during the LGM relative to their current position and that these areas were covered by forest mosaics/ parklands instead of todays'

dense forest. LPJ-GUESS simulations for LGM do reproduce this latitudinal biome shift (Fig. 5a). While we could not classify an open parkland vegetation type, the simulations show 5-20% lower FPC for this area at LGM (Fig. 6b) which can be interpreted as more open conditions. For the MH, temperate Valdivian rainforests similar to PD conditions did exist (Villagrán, 1988) and are also simulated by LPJ-GUESS (Fig. 5a)

LGM reductions of temperatures of up to 12 °C have been proposed for high altitudes in the tropics (Thompson et al., 1995),

and LPJ-GUESS also simulates a more expansive belt of high-elevation cold desert for LGM at the expense of high-Andean steppe (Fig. 5a,b). Furthermore, significantly lower tree lines and vegetation zones of up to 1500 m compared to PD are reported for LGM (Marchant et al., 2009), which in our simulations is reflected in smaller areas of the *Cold Deciduous Forest* biome and a shift to lower elevations (Fig. 5b).

In summary, the main forest transition in south-central Chile from LGM to PD was the postglacial expansion of forests (~15

ka BP). However, strong climate seasonality and forest clearing during early land utilization (as indicated by increased fire activities from 12–6 ka BP) and forest expansion into abandoned land after the Spanish colonial period (Lara et al., 2012) were other major, but anthropogenic, vegetation changes which were followed by intense land clearing for lumber extraction and farming (Armesto et al., 2010). Although significant for many areas, we opted to exclude anthropogenic land use in our

analysis as spatial intensities and general utilization is not well understood and hard to qualify and as we see this beyond the scope of this study.

## 5.3. Sensitivity of simulation to paleoclimate input data

As demonstrated in this study, climatic forcing of the ecosystem model is crucial for the simulated vegetation composition, biome establishment and associated vegetation cover and surface runoff, which are key controls of catchment denudation rates (Jeffery et al., 2014). The TraCE-21ka dataset was chosen since, to our knowledge, it is the only available dataset providing continuous transient monthly data. The low spatial resolution (T31/ ~3.75°) is however problematic in resolving regional scale heterogeneity and we therefore compare our LGM simulations with a regional paleoclimate model simulation (ECHAM5, resolution: T159/ 0.75°x0.75°, Mutz et al., 2018). Although the general latitudinal patterns of temperature deviations from PD are similar to TraCE-21ka, precipitation gradients and regional anomalies at LGM are stronger (see Fig. 3 for LGM anomaly plot of TraCE-21ka data and Fig. S1a,b for ECHAM5). While LGM precipitation anomalies in TraCE-21ka do not exceed 650 mm, precipitation rates in the ECHAM5 data vary by well over 1500 mm for the Andean highlands at 27-35° S and the coastal areas from 36-53° S but are markedly lower for the lower Andes and Tierra del Fuego (Fig. S1b). This leads to a different pattern of biome distribution for the LGM (Fig. 11). Due to lower temperatures, the cold deserts extend further north and a small zone of temperate rainforest exists at latitudes 40° S to 42° S, whereas *Magellanic Woodlands* are simulated with TraCE-21ka. A zone classified as *Magellanic Woodland* stretches from 30° S to 42° S along the Andean range that gives way to *Cold Deciduous Forest* at 28° S to 32° S at altitude. Temperate *Deciduous "Maule" Forest* exists north of 40° S and transitions into *Sclerophyllous Woodland* extending to 30° S, whereas TraCE-21ka results lead to *Steppe* at the lowlands and *Mesic Woodlands* at higher altitudes. Higher precipitation levels north of 30° S also lead to higher vegetation cover and larger areas of *Matorral* and *Arid Shrubland* and a reduction of desert conditions. This results in substantial differences of simulated foliar projected cover (Fig. 11b). While cover of *Steppe* and *Mesic Woodlands* simulated with TraCE-21ka is generally 10-20% higher versus the ECHAM5 simulations, the cover from 35° S to the Atacama Desert is substantially lower than ECHAM5 (higher precipitation rates for this region). Cover south of 42° S is substantially higher for TraCE-21ka as the colder conditions in ECHAM5 simulations generally prohibit the vegetation establishment.

The results show that choice of paleoclimate data clearly has an influence on simulated vegetation composition, but the impact on vegetation cover depends on the type and location of change. For instance, forest-to-forest transitions due to changes in temperature can have only little effect on FPC, while a different rate of annual precipitation at Mediterranean to semi-arid conditions can lead to substantial changes of FPC (Sta. Gracia: +200-500 mm precipitation in ECHAM5 LGM data results in > +20% FPC; see also Fig. 8a). Given the results from our study we might severely underestimate episodes of elevated erosion in future coupled model exercises and thus thorough sensitivity studies will be required.

## 5.4. Impact of atmospheric $CO_2$ on vegetation cover

The effectiveness of photosynthesis, and thus the plants ability to build-up biomass, has a large dependence on $[CO_2]$ (Farquhar et al., 1980; Hickler et al., 2015). Increases in vegetation biomass (expressed in LAI) were observed in our simulations but coincide with simultaneous rise of temperatures and $[CO_2]$ (Fig. 9b-d). As observed in the $CO_2$ sensitivity runs for LGM climate conditions, higher $[CO_2]$ concentrations lead to an expansion of vegetated areas at higher elevations and an expansion of forest biomes at the expense of herbaceous vegetation. Such changes and the magnitude of the $CO_2$ effect are consistent with earlier LGM simulations (Harrison and Prentice, 2003; Bragg et al., 2013). They suggest that $[CO_2]$ could have a crucial role also for understanding landscape evolution over longer time scales – a factor that again is not considered in previous studies of landscape evolution (i.e. Istanbulluoglu and Bras, 2005; Collins et al., 2004).

The strong limiting role of [$CO_2$] during the LGM is generally accepted, but to what extent substantial $CO_2$ fertilization effects still occur as concentrations increase to rise beyond the current stage is highly debated (Hickler et al., 2015). However, the LPJ-GUESS model with the enabled nitrogen cycle, and nitrogen limiting $CO_2$ effects, has been shown to generally reproduce experimental observations from "Free Air CO2 Enrichment" (FACE) experiments (e.g. Zaehle et al., 2014; Medlyn et at al., 2015; De Kauwe et al., 2017). Enabling the nitrogen cycle in LPJ-GUESS would be crucial if one aims at understanding vegetation response and landscape evolution at [$CO_2$] above present-day levels as it occurred in various episodes of Earths' history (i.e. the Miocene).

In general, we were able to show a temporal compatibility of paleo vegetation state, vegetation transitions, and simulated vegetation composition using our model (Sect. 5.2). Our results also indicate that the simulated vegetation is strongly influenced by climatic model drivers (see Sect. 5.3) and [$CO_2$], and by feedback mechanism occurring within the model (i.e. fire return intervals as controlled by climate and the production of fuel). Given the sensitivity of the vegetation cover to the climate forcing, careful consideration must be made of paleoclimate data and its characteristics and uncertainties. The TraCE-21ka data does not show high levels of variability, instead exhibiting only gradual changes over time (temperature, [$CO_2$]) or no strong, centennial to millennial scale trends (precipitation). In contrast, available proxy data suggest stronger variability, at least locally, although this is difficult to generalize for larger areas (see Sect. 5.2). Regional paleoclimate simulations with higher spatial resolution might be required to obtain better landscape-scale model forcing. Another approach could be to modify the climate-model-derived forcing based on proxy-data, such as increasing inter- and intra-annual variability within reasonable ranges (e.g. Giesecke et al., 2010).

## 6. Conclusions

In this study, we demonstrated how a DGVM can be applied to estimate vegetation features through time that can play an important role for evolution of landforms, and at a spatial and temporal resolution adequate for coupling with LEMs. The simulation also captures vegetation change drivers that are not explicitly represented in simplified vegetation representations used so far in LEMs, such as plant-physiological effects of changing [$CO_2$], fire dynamics that varies greatly with PFT composition and interaction with soil water resources through different rooting strategies. The sensitivity of landscape evolution to vegetation and climate changes is evaluated in the companion paper by Schmid et al. (this issue). Although our two studies stop short of presenting results from a fully coupled (dynamic vegetation and surface process) models, the results we present highlight a) how much vegetation likely changed in the Coastal Cordillera of Chile since the LGM (this study), and b) the general sensitivity of topography and erosion rates to the magnitudes of change identified here (Schmid et al., this issue). In future work we will implement a two-way coupling of LPJ-GUESS to LandLab. LPJ-GUESS will be driven by climate data and produce a continuous dataset of vegetation cover and surface hydrology that is passed to LandLab. Landlab will use this vegetation cover and simulated denudation rates and, in turn, will provide updated topography (and after a landform classification updated areal cover of landforms) and the associated soil depth information to LPJ-GUESS. The novel approach of representing sub-grid diversity in vegetation by landforms allows for an efficient computation with existing coarse-scale climate data and coupling to LEMs with higher spatial representations of topography.

From the experiments presented here, our main conclusions are as follows:

(1)   The regionally adapted version of LPJ-GUESS was able to simulate the latitudinal and altitudinal distribution of potential natural vegetation and the satellite-observed vegetation cover for present day conditions in most areas of Chile.

(2) While simulated MH vegetation did not differ substantially compared to PD PNV, simulated vegetation of the LGM indicates a marked northward shift of the biome distribution, a reduction of the tree line, and downward shift of vegetation zones at altitude. Vegetation cover was generally reduced compared to PD conditions and cold and hot desert were covering substantially larger areas of the simulation domain.

5 (3) Analysis of the results from transient site simulation indicate that temperature and [$CO_2$] did cause most of the observed shifts in vegetation composition and, in some cases, transitions between biomes over time. A sensitivity study highlighted the impact of 'CO$_2$-fertilization' on vegetation cover under LGM climate conditions.

(4) Comparisons with proxy data suggest that the coarse-scale climatic forcing does underestimate centennial to millennial climate variability. A combination of proxy-derived estimates and climate model results or higher resolution climate 10 models might be necessary to capture such variability.

(5) Our results show that vegetation cover in semi-arid to Mediterranean ecosystems responds strongly to changes in precipitation, while change of climatic conditions for temperate to boreal forest ecosystems do so only to a lesser extent.

(6) We consider the implementation of a landform classification a feasible tool to a) mediate between coarse DVM model 15 resolutions and generally higher resolution LEM with little computational expense and b) to account for sub-grid variability of micro-climate conditions that are otherwise absent from DVM simulations at larger scales. We envision that it will also allow new applications of LPJ-GUESS for research questions where a better representation of vegetation composition and state, especially the heterogeneity of the simulated sub-pixel vegetation, is important.

In summary, we suggest that coupling state-of-art dynamic vegetation modelling with LEMs has great potential for 20 improving our understanding of the evolution of landforms as the DVM using the landform approach can approximate spatial heterogeneity observed in the field that otherwise is not represented by standard DVM implementations. The FPC linked to topography structure will likely result in varying denudation rates in the landscape and have thus the potential to influence landscape evolution. The regional model adaptation and illustrated model improvements are an important step towards applying such a coupled model to the study area of EarthShape.

25 **Acknowledgements**

This study was funded as part of the German science foundation priority research program EarthShape: Earth surface shaping by biota (grant EH329-14-1 to T. A. E. and HI1538/5-1 to T. H.). We also thank the national park service of Chile (CONAF) for access to the study areas during field excursions.

30 **Competing interests**

The authors have no conflict of interest to declare.

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

**Table 1.** Major vegetation zones, associated plant functional types (PFT), and representative species (modified after Escobar Avaria, 2013; Luebert and Pliscoff, 2017; letters a-e are given to help the reader identify PFT and representative species).

| Biome | Plant functional type | Species |
|---|---|---|
| Arid Shrubland, Matorral, Sclerophyllous Woodland | (a) Temperate evergreen shrubs (TeE$_s$)<br>(b) Temperate raingreen shrubs (TeR$_s$)<br>(c) Temperate broadleaved evergreen sclerophyllous trees (TeBE$_{itscl}$) | *(a) Colliguaja odorifera, Porlieria chilensis, Fluorensia thurifera, Heliotropium stenohyllum,*<br>*(b) Retanilla trinervia, Trevoa quinquinervia, Acacia caven, Prosopis sp*<br>*(c) Quillaja saponaria, Lithraea caustica, Peumus boldus, Cryptocarya alba* |
| Temperate 'Maule' Forest | Temperate summergreen deciduous trees (TeBS$_{tm}$, TeBS$_{itm}$) | *Nothofagus alpine, N. glauca, N. obliqua, N. alessandrii* |
| Valdivian Rainforest (incl. North Patagonian Rainforest) | (d) Temperate broadleaved evergreen trees (TeBE$_{tm}$, TeBE$_{itm}$)<br><br>(e) Temperate needleaved evergreen trees (TeNE) | (d) *Embothrium coccineum, Weinmannia trichosperma, Nothofagus dombeyi, N. nitida, Eucryphia cordifolia, Drimys winteri, Laureliopsis philippiana, Aextoxicon punctatum, Luma apiculata, Persea lingue, Amomyrtus luma, Lomatia hirsuta.*<br>(e) *Fitzroya cuppressoides, Pilgerodendron uviferum, Saxegothaea conspicua, Podocarpus nubigena, Araucaria araucana* |
| Magellanic Rainforest/ woodlands | Boreal broadleaved evergreen trees (BBE) | *Nothofagus betuloides* (in hyperhumid areas) |
| Cold Deciduous Forest/ woodlands | Boreal broadleaved summergreen trees (BBS) | *Nothofagus pumilio, N. antarctica* (in humid areas) |
| (not dominant) (High-Andean) Steppe | Boreal evergreen shrub (BE$_s$)<br>Herbaceous vegetation (C3G) | *Bolax gummifera, Azorella selago*<br>Great floristic variability (from north to south): *Chaentanthera sphaeroidalis, Nastanthus spatolathus-Menonvillea spathulata, Oxalis adenophylla-Pozoa coriacea , Nassauvia dentata-Senecio portalesianus, Nassauvia pygmaea- Nassauvia lagascae.* |

**Table 2.** Areal extent of biomes in the simulation domain [units: percent of total area].

|  | LGM | MH | PD |
|---|---|---|---|
| Desert | 53.7 | 12.6 | 7.6 |
| Arid Shrubland | 1.9 | 4.5 | 5.6 |
| Matorral | 1.2 | 3.9 | 2.3 |
| Steppe | 13.6 | 11.6 | 12.4 |
| Sclerophyllous Woodland | - | 5.6 | 8.7 |
| Deciduous 'Maule' Forest | 0.4 | 2.1 | 2.1 |
| Mixed Forest | 0.4 | 0.4 | 0.6 |
| Valdivian Rainforest | 4.1 | 11.2 | 13.4 |
| Mesic Woodland | 6.6 | 2.9 | 2.7 |
| Cold Deciduous Forest | 1.7 | 15.3 | 12 |
| Magellanic Forest/ Woodland | 16.5 | 29.8 | 32.6 |

**Table 3.** Average foliar projected cover (FPC) and annual surface runoff for simulated biomes at present day (PD) and relative change within these PD areas for Last Glacial Maximum (Δ LGM, LGM-PD) and Mid Holocene (Δ MH, MH-PD).

| | Foliar projected cover | | | Surface runoff | | |
| | [%] | [% change] | | [mm yr$^{-1}$] | [% change] | |
| | PD | Δ LGM | Δ MH | PD | Δ LGM | Δ MH |
|---|---|---|---|---|---|---|
| Arid Shrubland | 16 | *-5* | *-2* | 0 | *0* | *0* |
| Matorral | 35 | *-9* | *-5* | 0 | *0* | *0* |
| Steppe | 39 | *-23* | *-7* | 1 | *+4* | *+1* |
| Sclerophyllous Woodland | 66 | *-6* | *-4* | 187 | *+35* | *+9* |
| Deciduous 'Maule' Forest | 85 | *-5* | *0* | 638 | *+114* | *+33* |
| Mixed Forest | 85 | *-42* | *-1* | 193 | *+24* | *+5* |
| Valdivian Rainforest | 88 | *-20* | *-1* | 910 | *+93* | *+63* |
| Mesic Woodland | 56 | *-21* | *-10* | 81 | *+57* | *+10* |
| Cold Deciduous Forest | 78 | *-44* | *-6* | 200 | *+62* | *+13* |
| Magellanic Forest/ Woodland | 77 | *-68* | *-7* | 966 | *+124* | *+35* |

**Figures**

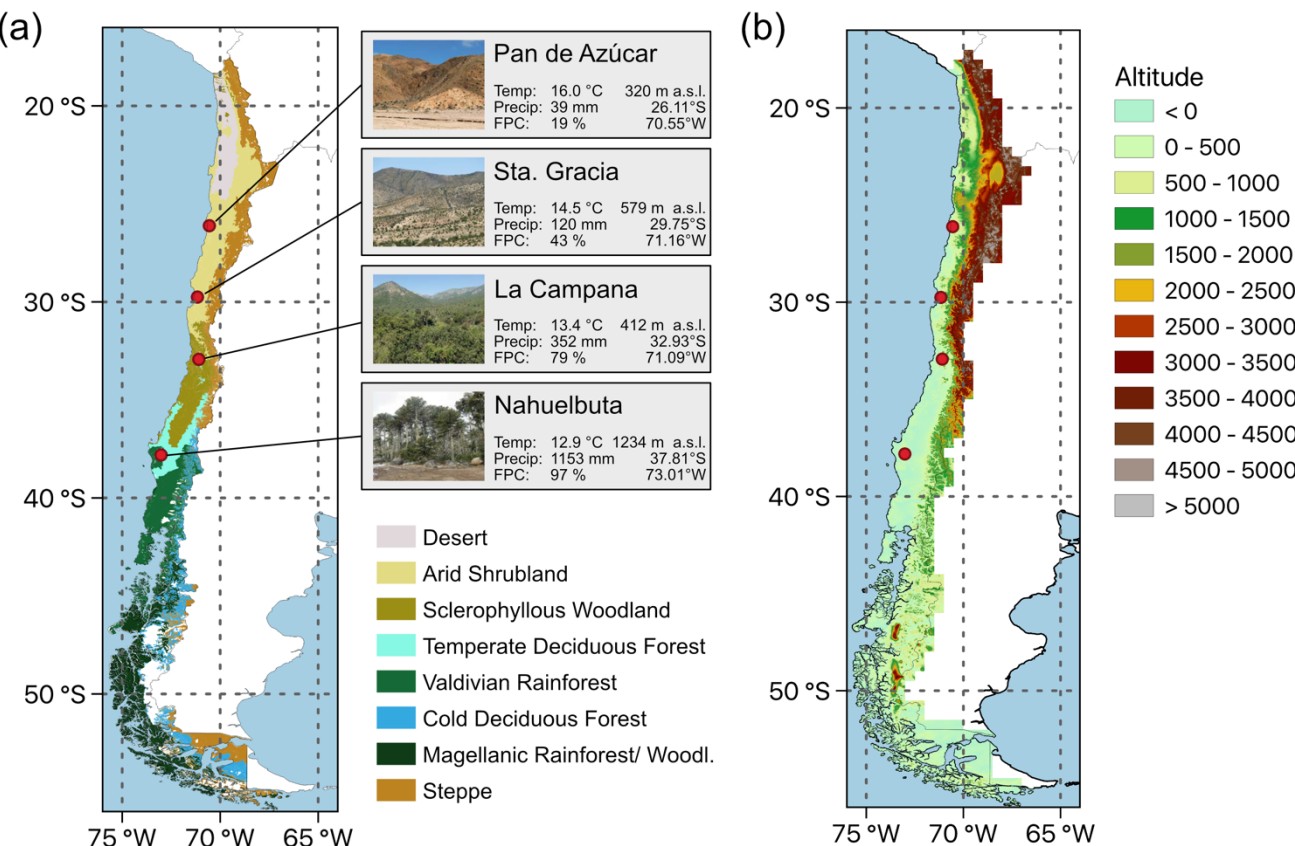

**Figure 1. (a)** Distribution of major vegetation zones in Chile and location of the four EarthShape SPP focus sites Pan de Azúcar, Sta. Gracia, La Campana and Nahuelbuta (temp: average annual temperature, precipitation: average annual precipitation - data: ERA-Interim 1960-1989, FPC: foliage projected cover (MODIS VCF v6, total vegetation cover, 2001-2016 average, Dimiceli et al., 2017). Vegetation zones are based on Luebert and Pliscoff (2017). **(b)** Elevation of the model domain.

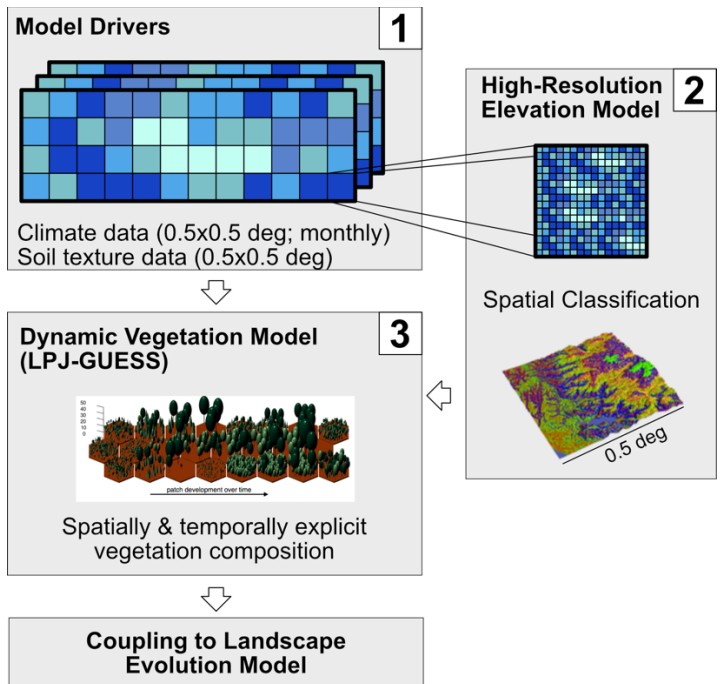

**Figure 2.** Schematic procedure of simulations. Coarse resolution model driving data (1) is disaggregated using a high-resolution elevation model and topographic landform classification (2) and the ecosystem model LPJ-GUESS then simulates vegetation state and dynamics using the landform classification to simulated topographic-adjusted patch composition (3). Vegetation cover and surface runoff results can then be passed on to a coupled landscape evolution model (LEM) (not implemented in presented study, see Schmid et al. in this issue for a description of the LEM).

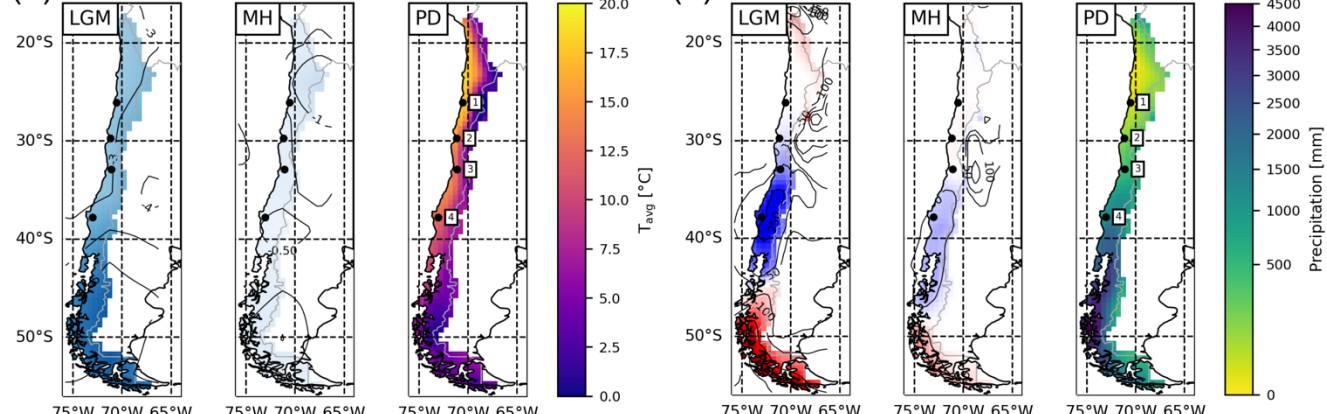

**Figure 3. (a)** Average annual temperature and **(b)** precipitation derived from the downscaled and bias-corrected TraCE-21ka transient paleoclimate data (Liu et al., 2009) for the Last Glacial Maximum (LGM), Mid Holocene (MH) and present day (PD) time-slices (data: average of 30-yr monthly data; 1: Pan de Azúcar; 2: Sta. Gracia; 3: La Campana; 4: Nahuelbuta).

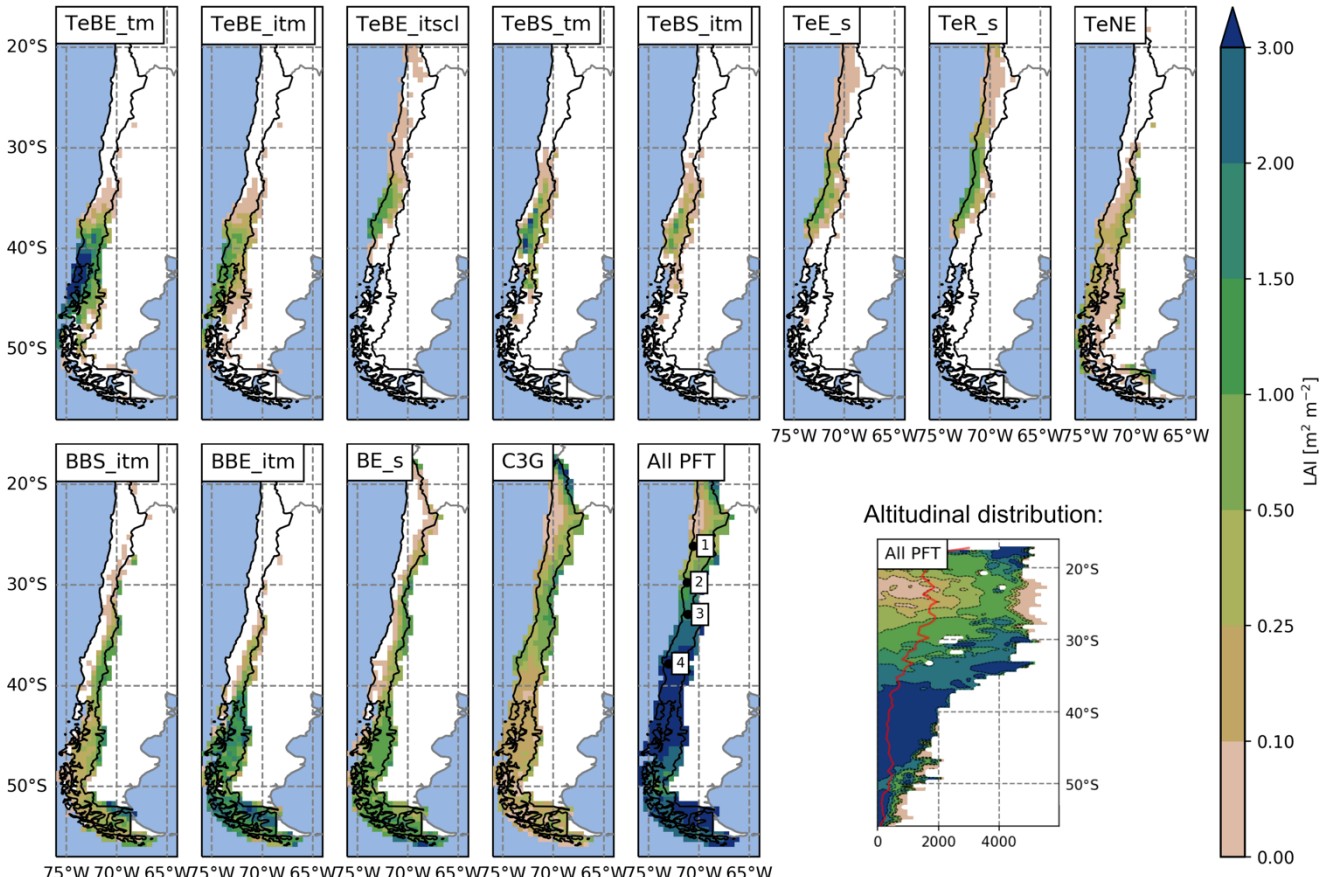

**Figure 4.** Spatial and altitudinal distribution of modelled plant functional types (PFT) for present day climate conditions (LAI: leaf area index [m² m⁻²], the altitudinal subplot represents zonal mean LAI; red line: average elevation). PFTs: TeBE$_{tm}$/ TeBE$_{itm}$ (temperate broadleaved evergreen trees; t = shade-tolerant; it = shade-intolerant; m = mesic), TeBE$_{itscl}$ (temperate broadleaved evergreen trees; scl = sclerophyllous), TeBS$_{im}$/ TeBS$_{itm}$ (temperate broadleaved summergreen trees), TeE$_s$ (temperate evergreen shrubs; s = shrub), TeR$_s$ (temperate raingreen shrubs), TeNE (temperate needleleaved evergreen trees), BBS$_{itm}$ (boreal broadleaved summergreen trees), BBE$_{itm}$: (boreal broadleaved evergreen trees), BE$_s$: (boreal evergreen shrubs), C3G (herbaceous vegetation). 1: Pan de Azúcar, 2: Sta. Gracia, 3: La Campana, 4: Nahuelbuta.

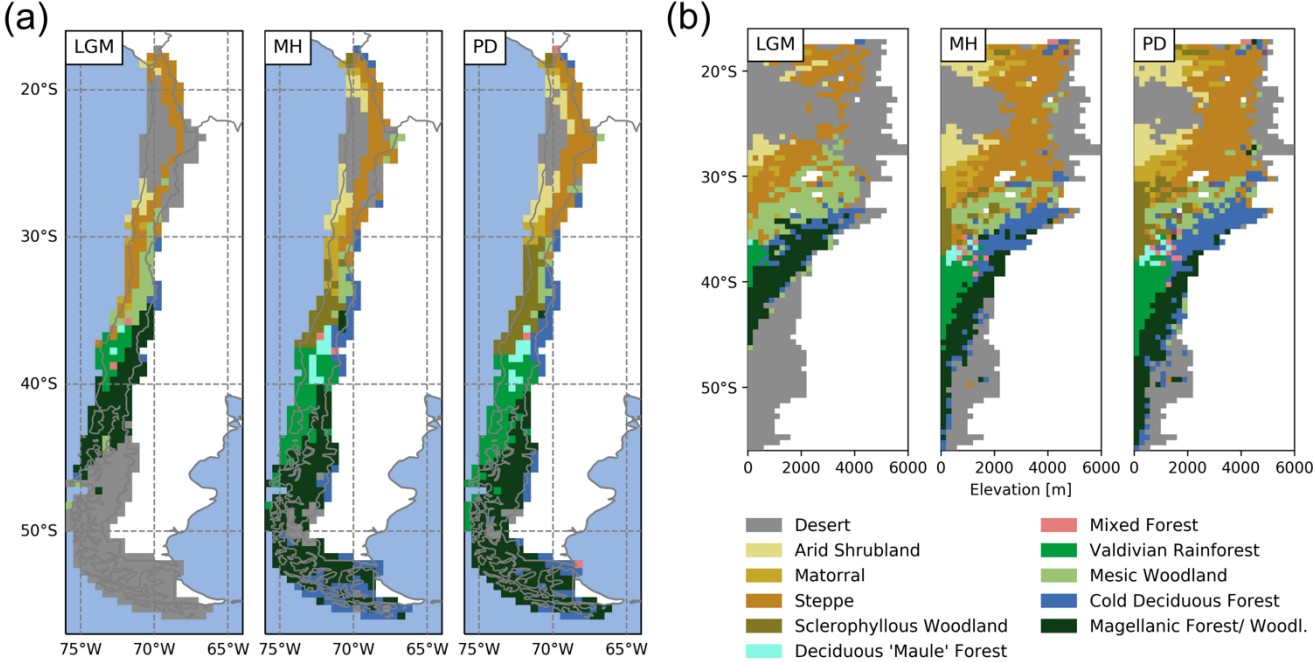

**Figure 5. (a)** Spatial and **(b)** altitudinal distribution of biomes for Last Glacial Maximum (LGM), Mid Holocene (MH) and present day (PD) (for biome classification decision tree see Fig. C1).

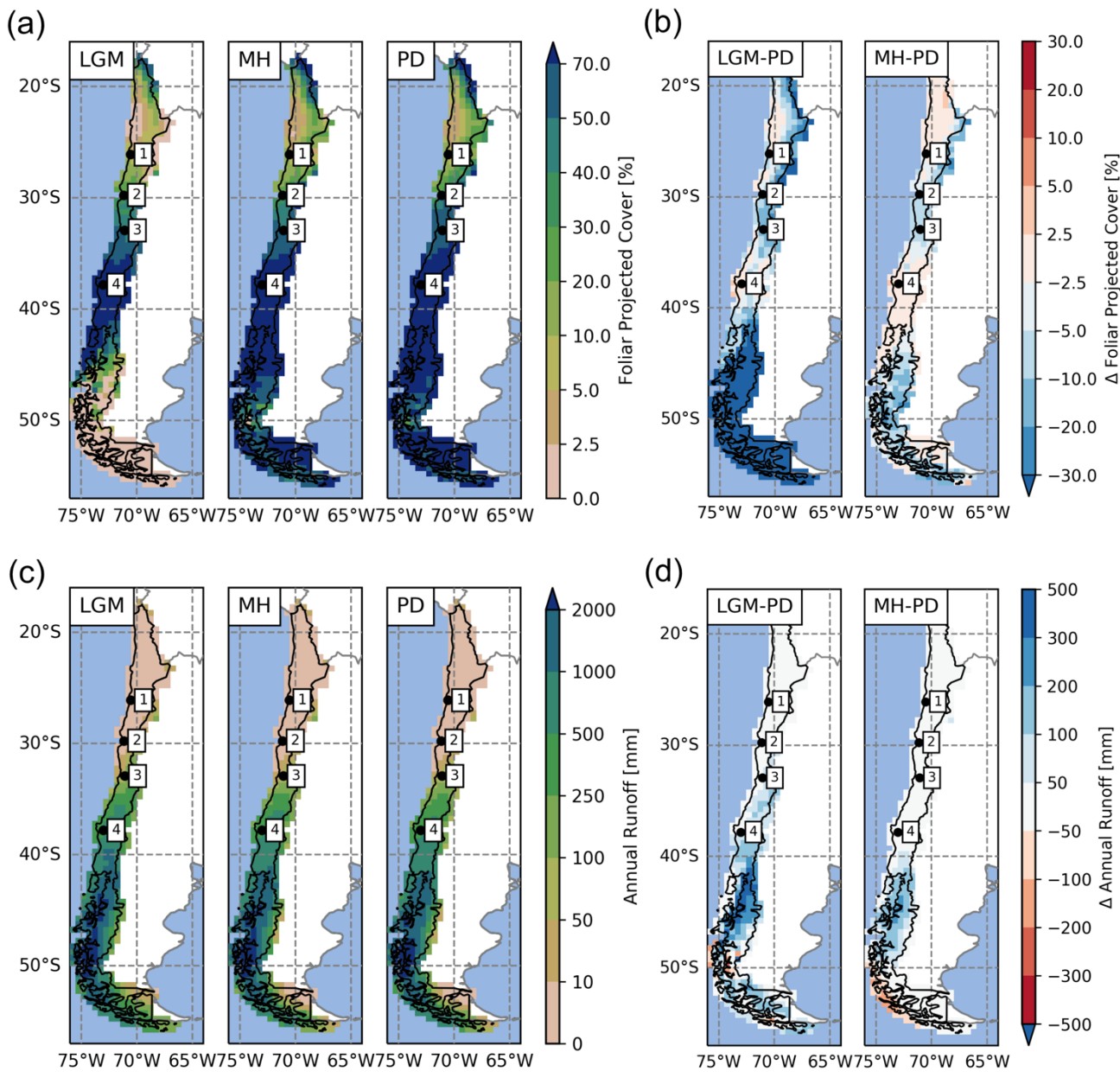

**Figure 6.** Spatial distribution of **(a)** foliar projected cover and **(c)** surface runoff simulated for Last Glacial Maximum (LGM), Mid Holocene (MH) and present day (PD). **(b)** and **(d)**: difference plots of LGM vs. PD (left) and MH vs. PD (right) for foliar projected cover and runoff, respectively (1: Pan de Azúcar, 2: Sta. Gracia, 3: La Campana, 4: Nahuelbuta).

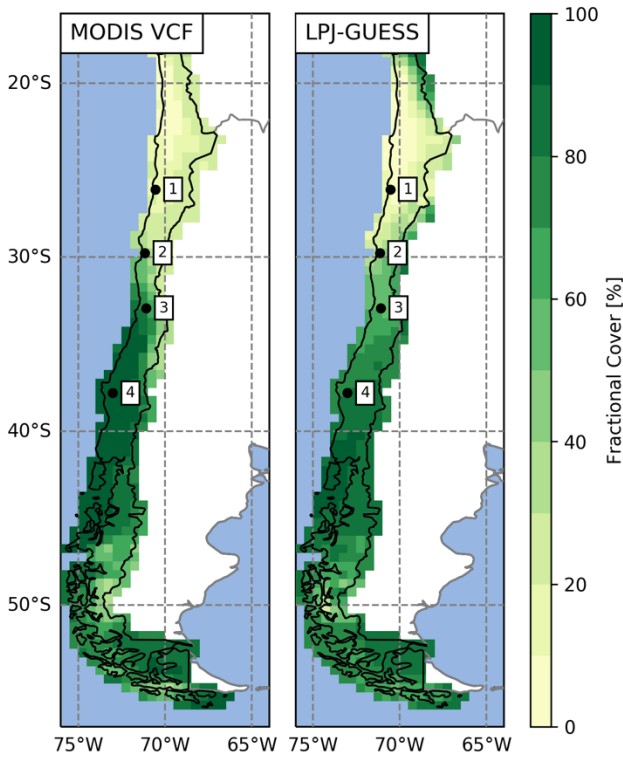

**Figure 7.** Comparison of satellite-derived vegetation cover and simulated average foliar projected cover of potential natural vegetation for present day (data: MODIS MOD44B VCF v6, total vegetation, 2001-2016 average; Dimiceli et al., 2017; 1: Pan de Azúcar, 2: Sta. Gracia, 3: La Campana, 4: Nahuelbuta).

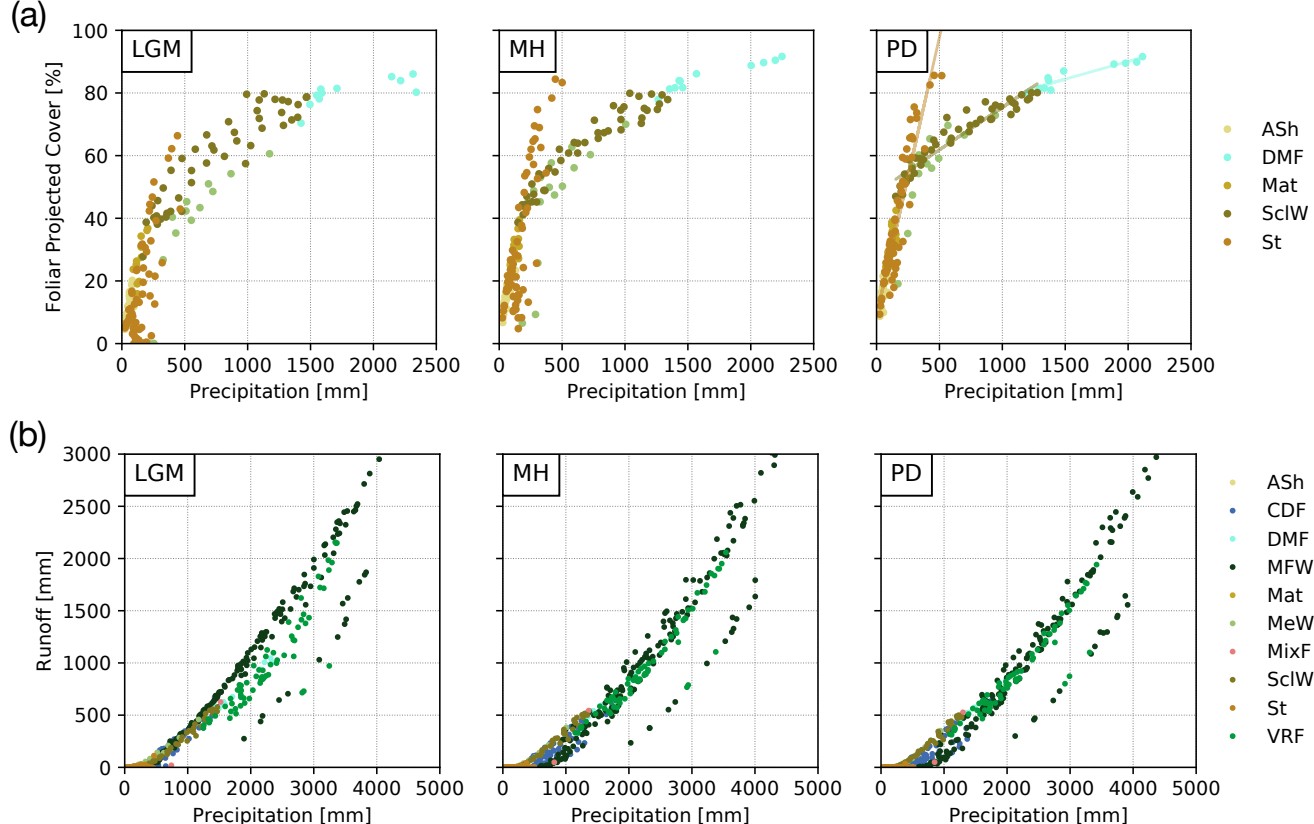

**Figure 8. (a)** Foliar projected cover (FPC) and **(b)** surface runoff as a function of annual average rainfall for Last Glacial
Maximum (LGM), Mid Holocene (MH) and present day (PD) (ASh: *Arid Shrubland*, CDF: *Cold Deciduous Forest*, DMF:
*Deciduous 'Maule' Forest*, MFW: *Magellanic Forest/ Woodland*, Mat: *Matorral*, MeW: *Mesic Woodland*, MixF: *Mixed
Forest*, SclW: *Sclerophyllous Woodland*, St: *Steppe*, VRF: *Valdivian Rainforest*; temperate and boreal forest biomes (CDF,
MixF, VRF, MFW) excluded from subplot (a) as they are also strongly dependent on temperature.

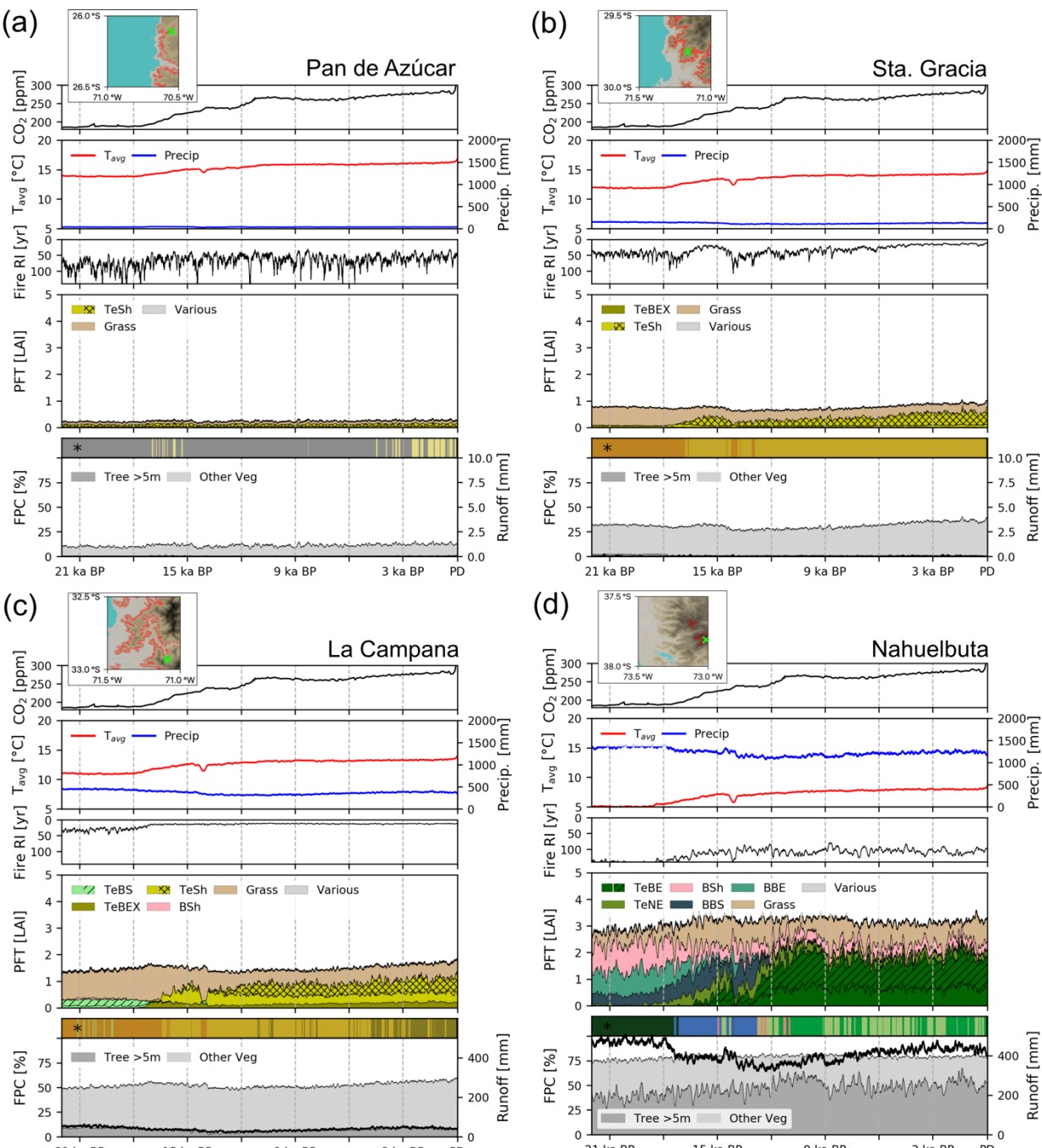

**Figure 9.** Transient simulations for the sites **(a)** Pan de Azúcar (a), **(b)** Sta. Gracia, **(c)** La Campana, and **(d)** Nahuelbuta. The insets show the location within the simulated 0.5°x0.5° grid cell and the area and location covered by the landform plotted in this figure (results given in other figures are an area-weighted aggregation of individual landform simulation results). Panels 1-3: $T_{avg}$: annual average temperature, Precip: annual precipitation, Fire RI: fire return interval. Panel 4: PFT: plant function type, (grouped) PFT abbreviations: TeSh (temperate shrub), Grass: herbaceous vegetation, TeBEX: sclerophyllous temperate broadleaved evergreen tree, TeBS: temperate broadleaved summergreen tree (hashed: shade-intolerant), BSh: boreal evergreen shrub, TeBE: temperate broadleaved evergreen tree, TeNE: temperate needleleaved evergreen tree, BBS: boreal broadleaved summergreen tree, BBE: boreal evergreen tree, Various: other PFTs (LAI < 0.05); plain: shade-intolerant, hatched: shade-tolerant, cross-hatched: raingreen. Panel 5: FPC: foliar projected cover, classification: trees and shrubs >5m (dark gray), herbaceous vegetation and small trees and shrubs (light gray); Runoff: simulated surface runoff; *) Biomization: for a legend on the applied classification see Fig. C1 and Fig. 4 for a color-coded legend. All data smoothed using 100-year averaging.

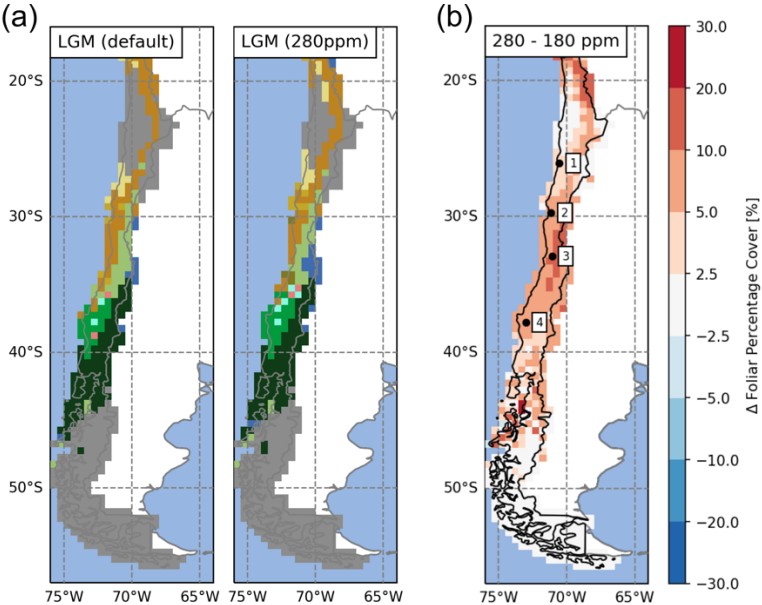

**Figure 10. (a)** Effect of atmospheric $CO_2$ ([$CO_2$]) concentrations on the simulated distribution of biomes for Last Glacial Maximum (LGM) time-slice simulations (left panel: default LGM setup of this study with [$CO_2$] = 180 ppm, right panel: control run with pre-industrial [$CO_2$] = 280ppm). **(b)** Difference plot of resulting foliar projected cover (280 ppm − 180 ppm). 1: Pan de Azúcar, 2: Sta. Gracia, 3: La Campana, 4: Nahuelbuta.

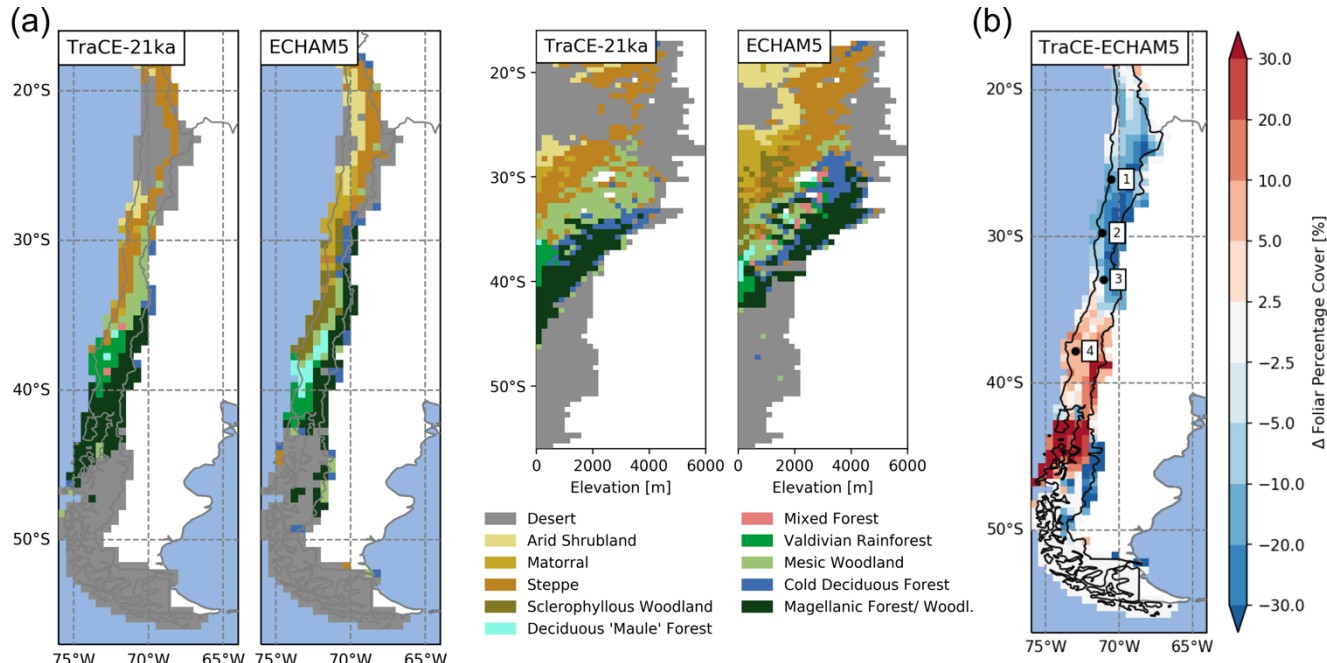

**Figure 11. (a)** Effect of paleoclimate input data used for LPJ-GUESS simulations on the spatial and altitudinal biome distribution and **(b)** effect on simulated foliage projected cover (given as difference between TraCE-21ka simulation and ECHAM5 simulation results, Mutz et al., 2018). For a comparison of differences between average temperatures and precipitation between the two climate datasets see Fig. 3 and Fig. S3. 1: Pan de Azúcar, 2: Sta. Gracia, 3: La Campana, 4: Nahuelbuta.

**Appendix A: Calculation of Monthly Wet Days**

It is a well-documented problem that climate models, such as CCSM3, have a tendency to overestimate the precipitation frequency in dry regions (e.g. Dai, 2006). Therefore, we parameterize the number of "wet days" (number of days in a month with rainfall greater than 0.1 mm day$^{-1}$) based on ERA-Interim daily climatology. It is assumed that the daily precipitation in a month follows a Gamma distribution, which is determined by the shape and scale parameters ($\alpha$ and $\beta$, respectively) as:

$$\begin{aligned} \alpha &= (x_{mean}/x_{std})^2, \\ \beta &= x_{std}^2/x_{mean} \end{aligned}$$

where $x_{mean}$ is the monthly mean precipitation, and $x_{std}$ its standard deviation (day-to-day variability). A characteristic feature of the Gamma distribution is its ability to attain two completely different shapes depending on the value of $\alpha$. If $\alpha < 1$ (typical for dry regions), the probability density attains maximum value at zero precipitation and decreases exponentially towards higher precipitation values, and if $\alpha > 1$ (typical for wet regions), the probability density function has a shape more reminiscent of a Gaussian distribution.

The number of wet days is estimated from the cumulative gamma distribution

$$F(x, \alpha, \beta) = \frac{1}{\beta^\alpha \Gamma(\alpha)} \int_0^x t^{\alpha - 1} \exp(-t/b) dt,$$

where $\Gamma$ is the Gamma function. The result of this equation is the probability that an observation will fall in the interval [0 x]. Hence for our purposes, the number of wet days ($n_{wet}$) is determined by

$$n_{wet} = n_{day}(1 - F(x_t, \alpha, \beta))$$

where $n_{day}$ is the number of days in a month, and $x_t$ the threshold value for a wet day (0.1 mm/day).

In our experiments the TraCE-21k climatology influences monthly $n_{wet}$ by modifying $x_{mean}$ at each grid cell. However, due to the poor representation of precipitation frequencies in the TraCE-21ka data, we use a monthly climatology of $x_{std}$ calculated from ERA-Interim.

## Appendix B: Implementation of landforms in LPJ-GUESS

LPJ-GUESS by default acknowledges within-grid cell variability of vegetation by the concept of patches (Smith et al., 2014). A patch represents a subset of the grid cell that usually is 0.5°x0.5° in size. Patches are not spatially registered to any

particular location within this cell. By definition, they represent an area of 0.1 ha - the assumed maximum area a mature tree might cover. The replication of these patches ensures that stochastic events (i.e. vegetation establishment and mortality, fire) effect only subsets of a grid cell and allow the model to simulated gap-dynamics and succession. Studies usually are configured with n >= 100 patch-replications to prevent the stochastic events from dominating simulated average grid cell results. In this study we introduce 'landforms' into LPJ-GUESS. The aim is to address two problems. First, the default grid

cell size (0.5x0.5) does not allow to address observed landscape heterogeneity (i.e. local site conditions, topographic structure of a catchment). While LPJ-GUESS has been applied at higher resolutions, the lack of high-quality environmental forcing for these resolutions make this approach often impractical. Second, this new approach does allow us to link two models of different model resolutions (LPJ-GUESS and climate drivers 0.5°x0.5°, LEM LandLab 100x100m) for approximating the true DEM characteristics into homogeneous landform units that aim to characterize the dominant

topographic units within this 0.5°x0.5° grid cell. Thus, the landform concept can be used to mediate the information exchange between these two models. In the implemented landform concept we define a set of patch groups for each landform (i.e. a subset of the grid that shares the same topographic features similar elevation, slope and aspect). The classification is based on a high-resolution elevation model of the grid cell (SRTM1 data, 30m), but in a future coupled-model the high-resolution DEM will be provided and continuously updated by the landscape evolution model coupled to

LPJ-GUESS. The model forcing (0.5°x0.5° climate and soil texture) is modified for the defined landforms of a given grid cell (see also Fig. B1).

### Classification of landforms and modification of environmental drivers

In order to classify landforms, we use the elevation and computed slope and aspect of the grid cells of the high-resolution

DEM. Furthermore, aspect and slope are used to compute a topographic position index (TPI) - the difference between elevation and the elevation of surrounding positions (focal radius 300 m, see Weiss 2001). The TPI and slope values are then used to classify positions into discrete topographic classes (here: ridges (TPI > 1 SD), mid-slopes (TPI > -1 SD and TPI < 1 SD and slope >= 6°), valleys (TPI <= -1 SD) and flats (TPI > -1 SD and TPI < 1 SD and slope < 6°). These classes are then stratified by elevation intervals to finally form the landforms (the number of landforms per grid cell depends on complexity

of the terrain: mean: 17; 25% quantile: 11, 75% quantile: 23). The average elevation, slope and aspect are then used to adapt the environmental forcing for this landform.

In this study, we adapt the landform surface temperature via the elevation difference of the 0.5°x0.5° grid cell elevation $E_{GC}$ and the average elevation of the high-resolution DEM occupied by a landform ($E_{LF}$) and adjust the temperature with the global lapse rate $\gamma$ of -6.5 °C km$^{-1}$ (see Eqn. 2).

$$T_{LF} = T_{GC} + \gamma(E_{LF} - E_{GC})$$

Furthermore, we adapt the amount of absorbed radiation based on the landform slope, aspect and time of the year. The solar declination ($\delta$) at any given day in the year (doy: day of year) is calculated in LPJ-GUESS as follows (Prentice et al., 1993,

all angles in radians):

$$\delta = -23.4 \times \cos\left(2\pi \times \frac{doy + 10.5}{365}\right)$$

The solar angle at noon is calculated from the latitude (lat) and δ as:

$$A_Z = lat - \delta$$

The corrected radiation at the landform ($R_{LF}$) for north or south facing slopes depending on their aspect (ψ) and slope (β)
angle is then calculated from grid cell radiation in LPJ-GUESS ($R_{GC}$):

$$R_{LF} = R_{GC} \times \left(\frac{1}{\cos A_Z}\right) \times \cos(|A_Z| \pm \beta \times \cos(\beta \times |\psi|))$$

Finally, the soil depth of the landform is adjusted based on the TPI of the landform. The default soil depth (DSD) of 1.5m is
scaled by multiplying the height of the lower soil layer (default: 1m) with fixed values for ridges (0.25), mid-slope positions
(0.5), and valleys (1.5) resulting in total soil depth of 0.75m, 1m, and 2m, respectively.
In the future coupled model actual soil depth will be provided by the landscape evolution model.

**Model setup with landforms**

Instead of running LPJ-GUESS with one environmental condition for all n patches (n >= 100) of a grid cell, the model is
now executed for each landform and its adjusted environmental conditions n times (n=15). In both model setups the grid cell
results are reported as the average results over all n patches (Fig. C1). However, in the landform setup the results are
reported as area-weighted averages (area fraction = landform fraction within the grid cell). Patches within a landform are
averaged like in the default model setup. In a model-coupling setup the model results per landform will be disaggregated
back onto the high-resolution DEM of the landscape evolution model to provide a spatially explicit vegetation cover.

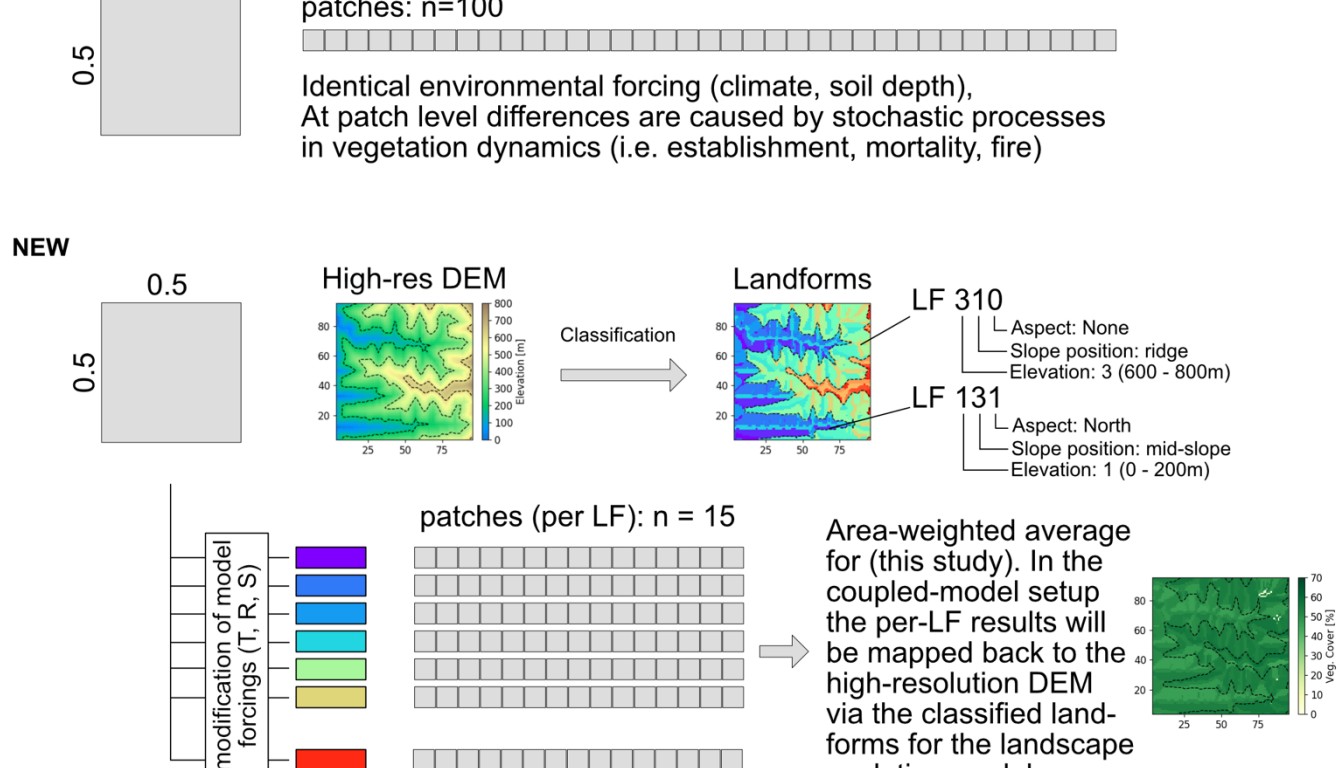

**Figure B1.** Conceptual difference between original LPJ-GUESS patch setup and the new landform addition (T: temperature,
R: radiation, S: soil depth, LF: landform).

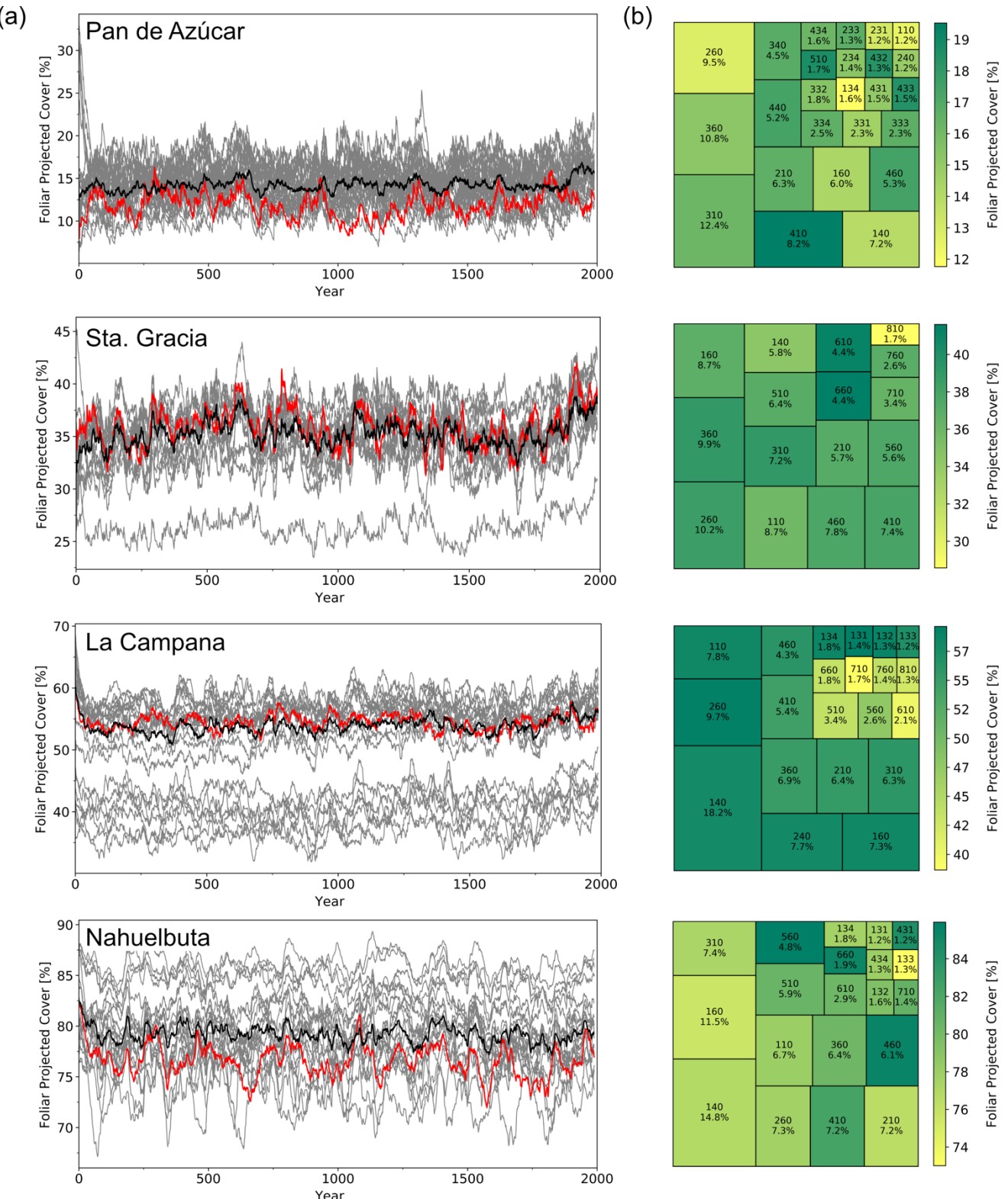

**Figure B2. (a)** Change of foliar projected cover for the individual landforms (gray), the area-weighted mean of all landforms (black) and the default LPJ-GUESS setup with no landforms (red) for the last 2000 years of the transient simulations. **(b)** Areal extent and foliar projected cover of the individual landforms in the four simulated grid cells (averaged for the last 100 years of the timeseries; text: landform id code and percentage of total grid cell covered by this landform in %).

# Appendix C: Plant Functional Type setup and biomization scheme

**Table C1.** Plant Functional Type (PFT) characteristics used in this study. Climate classes are associated with differing photosynthesis optimum temperatures and base respiration rates (see Smith et al., 2001; Te: temperate, B: boreal; M: Mediterranean was newly introduced with PS temperatures min: 0, low: 17, high: 27, max: 40; resp. coefficient: 1.0). $k_{allom1}$ = constant in allometry equations (Smith *et al.*, 2001; higher values equal wider crowns); $T_{c,min}$ = minimum coldest-month temperature for survival; $T_{c,max}$ = maximum coldest-month temperature for establishment; $GDD_5$ = minimum degree-day sum above 5 °C for establishment; $fAWC$ = minimum growing-season (daily temperature > 5°C) fraction of available soil water holding capacity in the first soil layer; $r_{fire}$ = fraction of individuals surviving fire; $k_{la:sa}$ = leaf area to sapwood cross-sectional area ratio; $z_1$ = fraction of roots in first soil layer (the reminder being allocated to second soil layer); $a_{leaf}$ = leaf longevity; $a_{ind}$ = maximum, non-stressed longevity; $CA_{max}$ = maximum woody crown area. r: base respiration rate (modified after Hickler et al., 2012).

| PFT | Climate | $r$ (g C g N$^{-1}$ d$^{-1}$) | Lifeform | $k_{allom1}$ | $T_{c,min,s}$ (°C) | $T_{c,min}$ (°C) | $T_{c,max}$ (°C) | $T_{wmin}$ | $GDD_5$ (°C d) | $fAWC$ | Shade tol. | $r_{fire}$ | $k_{la:sa}$ | $z_1$ | $a_{leaf}$ (yr) | $a_{ind}$ (yr) | $CA_{max}$ (m$^2$) |
|---|---|---|---|---|---|---|---|---|---|---|---|---|---|---|---|---|---|
| TeBE$_{tm}$ | Te | 0.055 | tree | 250 | -1 | 0 | 15 | | 900 | 0.3 | tolerant | 0.1 | 6000 | 0.7 | 2.0 | 500 | 30 |
| TeBE$_{itm}$ | Te | 0.055 | tree | 250 | -1 | 0 | 15 | | 900 | 0.3 | intolerant | 0.1 | 6000 | 0.7 | 2.0 | 400 | 30 |
| TeBE$_{itscl}$ | Te/ M | 0.055 | tree | 250 | 1 | 4 | 18.8 | | 2400 | 0.01 | intolerant | 0.5 | 4000 | 0.5 | 2.0 | 250 | 30 |
| TeBS$_{tm}$ | Te | 0.055 | tree | 250 | -14 | -13 | 6 | 5 | 1800 | 0.3 | tolerant | 0.2 | 6000 | 0.6 | 0.5 | 500 | 30 |
| TeBS$_{itm}$ | Te | 0.055 | tree | 250 | -14 | -13 | 6 | 5 | 1800 | 0.3 | intolerant | 0.2 | 6000 | 0.6 | 0.5 | 400 | 30 |
| TeE$_s$ | Te/ M | 0.055 | shrub | 100 | 1 | 1 | - | | 2600 | 0.001 | intolerant | 0.5 | 3000 | 0.5 | 2.0 | 100 | 10 |
| TeR$_s$ | Te/ M | 0.055 | shrub | 100 | 1 | 1 | - | | 2800 | 0.001 | intolerant | 0.5 | 3000 | 0.5 | 1.0 | 50 | 10 |
| TeNE | Te | 0.055 | tree | 150 | -7 | -7 | 22 | | 600 | 0.3 | intolerant | 0.5 | 5000 | 0.7 | 2.0 | 400 | 30 |
| BBS$_{itm}$ | B | 0.11 | tree | 250 | -30 | -30 | 3 | | 150 | 0.1 | intolerant | 0.1 | 6000 | 0.6 | 0.5 | 300 | 30 |
| BBE$_{itm}$ | B | 0.11 | tree | 250 | -30 | -30 | 5 | | 250 | 0.5 | intolerant | 0.1 | 6000 | 0.8 | 2.0 | 400 | 30 |
| BE$_s$ | B | 0.11 | shrub | 100 | | - | 4.5 | | 150 | 0.3 | intolerant | 0.1 | 2000 | 0.8 | 2.0 | 50 | 10 |
| C3G | - | 0.055 | herbac. | - | - | - | - | | - | 0.1 | - | 0.5 | - | 0.9 | 0.75 | - | - |

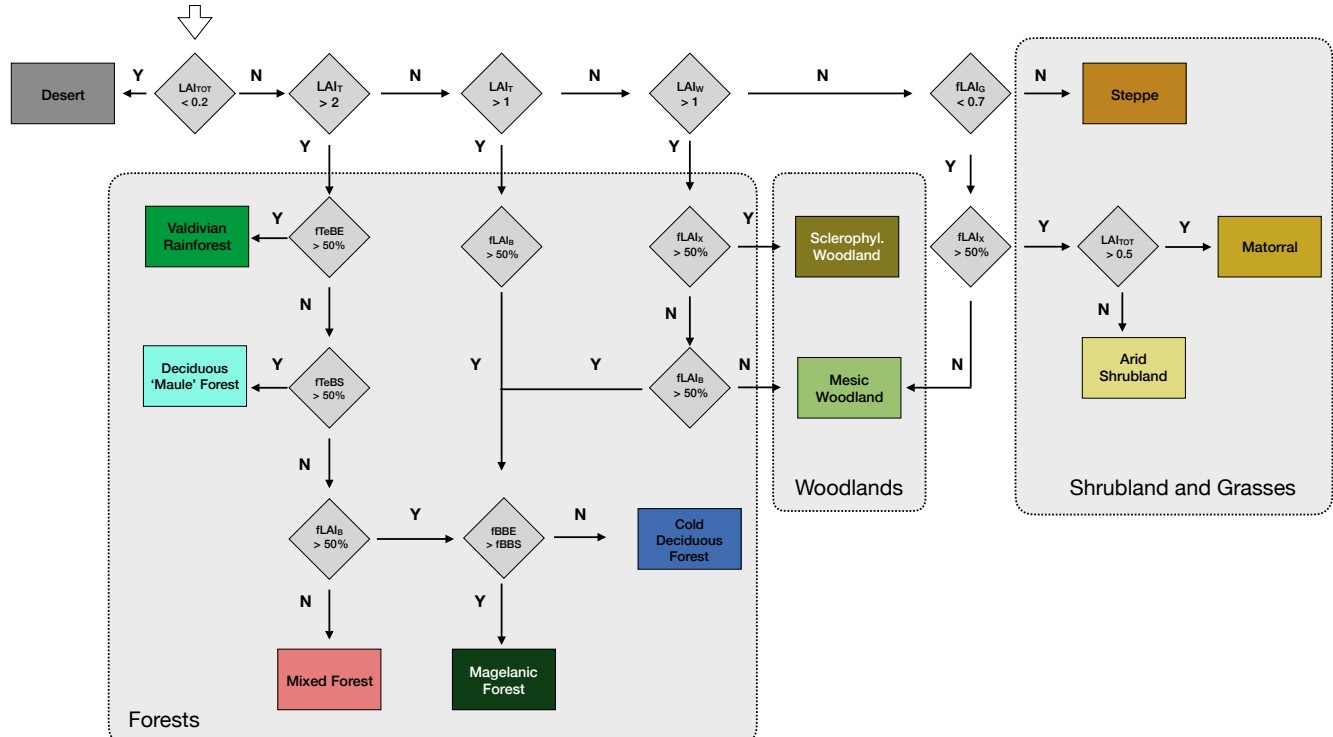

**Figure C1.** Flowchart of the decision tree used to classify plant functional types (PFTs) into biomes. $LAI_{TOT}$: LAI of all PFTs; $LAI_T$: LAI of tree PFTs; $LAI_W$: LAI of trees + shrubs; $LAI_G$: LAI herbaceous PFT; $LAI_X$: LAI of xeric PFTs; $LAI_B$: LAI of boreal PFTs; $fLAI_n$: fraction of LAI pf $PFT_n$ versus its peers (all tree PFTs, boreal PFTs, etc.); TeBE: temperate broadleaved evergreen tree; TeBS: temperate broadleaved summergreen tree; BBE: boreal broadleaved evergreen tree; BBS: boreal broadleaved summergreen tree.