# Peer review of "Effect of changing vegetation and precipitation on denudation (part 1): Predicted vegetation composition and cover over the last 21 thousand years along the Coastal Cordillera of Chile"

_Earth Surface Dynamics, 2018_

## Referee Comment (RC1) · Anonymous Referee #1 · 30 Mar 2018

Review of "Effect of changing vegetation on denudation (part1) : Predicted vegetation composition and cover over the 21 thousand years along the Coastal Cordillera of Chile" by Werner et al., submitted to Earth Surf . Dynam. Discuss.

The main objective of this paper is to demonstrate that a dynamic vegetation model can be used to simulate vegetation at temporal and spatial scales consistent with those of landscape evolution models (as stated by the authors themselves at the beginning of the discussion section), towards a coupling of both types of models. The study focuses on the vegetation of the Coastal Cordillera in Chile and its evolution since the

last glacial maximum. It uses the LPJ-GUESS dynamic vegetation model forced by climatic outputs of a transient simulation (TraCE-21ka) of the Community Climate System Model version 3 (CCSM3) from the last glacial maximum to the present. The results are analysed along the two main variables that are important for land denudation process in the landscape evolution model, i.e., vegetation cover fraction and surface runoff.

There are mainly two aspects which, to my knowledge, can be considered as novel in this paper: (1) the development of specific plant functional types for the Coastal Cordillera in Chile and their use in a dynamic vegetation model to study vegetation changes since the last glacial maximum, and (2) the development of landforms within a dynamic vegetation model. Thus, the paper deserves publication. It is moreover well written and relatively well organised. It can be published after a minor revision addressing the few comments summarized below.

Major comments

I have two main comments that the authors should address in their revised version:

(1) It is a bit strange to find the validation of the model near the end of the paper, in the Discussion section (Evaluation of predicted PNV), and furthermore this validation should be more quantitative. This section should come much earlier in the paper, probably at the beginning of the Results section. Other parts of the Discussion section could also be transferred to the Results section (but maybe near the end of the section): the sensitivity tests performed in subsections 5.3 and 5.4; these are results and not just discussion. More importantly, I feel that the validation of subsection 5.1 should be improved. As it is now, it is limited to a visual comparison of biome maps, as well of the foliage projection cover map predicted by the model with the vegetation cover map derived from MODIS data. You should provide at least some statistics for this comparison. Also, there is no validation of runoff, while it is reported as a very important variable for the landscape evolution model.

(2) It is not clear to me that this study fulfils the objective of demonstrating the ability

of a dynamic vegetation model to produce results useful for the spatial and temporal scales of landscapes evolution models. The authors develop a landform sub-model, but they do not really test it. They just present some transient evolution for one landform in each of four given model pixels. However, we do not know how far the use of the landform sub-model improves model prediction. It would be useful to illustrate the landform results for a given pixel (in addition to the altitudinal profiles that I guess use the landforms). Also the results of simulations with the landforms should be compared to those obtained with a model without landforms. How far does it improve the comparison with observed vegetation, or with MODIS vegetation cover? How far the results are affected by the adjustment of radiation for slope and aspects, or by the change in soil depth from the valley to the mountain slope or ridge? Landform modelling is a novel aspect put forward by this paper. So, it is important to discuss it more fully.

Minor comments

Introduction

- p. 2 lines 15-20: this paragraph provides a review of the use of DGVMs for paleoclimatic applications. They, however, mostly refer to studies performed with LPG-GUESS. Please, please provide also examples of studies performed with other DGVMs.

Methods

- p. 4, line 25: "We approximate the fraction A of the land surface ..." instead of " We approximate the land surface ..." - p. 4, line 37: Field capacity looks strange here. This would mean that a bucket approach is used in both layers. However, since drainage is not possible below field capacity (this is its definition), it would mean that subsurface runoff and percolation rate through the second layer are always zero in your model. Please check. - p. 5, line 15: you use a constant average lapse rate of 6.5°C/km, whereas the lapse rate could significantly vary, especially in desert areas where it could tend towards the dry adiabatic value of 9.7°C/km. Moreover, other climate variables can change significantly with elevation in mountain areas, such as

**ESurfD**
precipitation, cloudiness and air relative humidity. - p. 6, section 3.4: for PD, you use the 1960-1989 period. Does the atmospheric $CO_2$ for PD correspond to the mean $CO_2$ during this period? If so, it is significantly larger than the Holocene mean value and it is thus necessary to perform a pre-industrial simulation in addition to PD, in order to separate the $CO_2$ and the climate effect in the difference between PD and MH.

Results

- p. 7, line 20: "the Deciduous 'Maule' Forest occurs as total rainfall decreases and rainfall seasonality increases." According to table 1, the 'Maule' Forest is a temperate forest made of temperate summergreen trees, not raingreen trees. So, we would expect that it is the seasonality of temperature and not rainfall that determines the occurrence of these trees. Please be more precise on the processes that link this forest to rainfall seasonality. - p. 8 lines 19-21 and table 3: FPC in the south is lower during MH than at PD. Why? According to Fig. 3, in the south, the climate is wetter and colder during MH. We would expect larger FPC. Is the difference due to $CO_2$, which is larger at PD? This needs to be commented. - p. 8 line 35: "... between PFTs that might otherwise be lost ..."

Discussion

- p. 10, line 19: "The surface runoff simulated here was found to be consistent with expected patterns" – This is not really true, since no validation of runoff has been made. - p. 12, line 38: Hickler et al., 2015; Zhu et al., 2016 – Please refer to earlier literature, this has been discussed much earlier by many authors.

Table 1

- Please provide, as far as possible, example species for all PFTs

Table 2

- it might be interesting to also list in this table the PD biome areal extent from the observed map of figure 1, in order to compare them with the model

**ESurfD**
[Figure]

Table 3

- according to the legend, the table lists PD absolute values, but relative changes (in % ??) with respect to PD for the LGM and MH. However, the title at the top of the table, runs over three columns, which is misleading, because it suggests that all values are % cover or mm yr-1, i.e., absolute changes. Please revise.

Figure 1

- it might be useful to provide a map of elevation next to the vegetation map

Figure 8

- Legend, line 11: " dark grey" instead of "darkgrey"

Appendix A

- p. 32, line 8: ". . . completely different shapes . . ." instead of ". . . completely shapes. . ."

---

## Referee Comment (RC2) · Anonymous Referee #2 · 1 Apr 2018

This study aims to provide a basis for importance of accounting transient vegetation changes in landscape evolution models. By demonstrating the ability of a DVM to simulate vegetation at compatible scales with landscape evolution models, authors' objective is to highlight the potential for coupling vegetation and landscape modelling.

The study is well executed and well written in general. However, I could only recommend the publication of the manuscript after addressing some major points. The overall quality and the meticulousness of the paper suggest that such revisions are within the reach of the authors. Please see the comments below.

[Figure]

**GENERAL COMMENTS**

Neither adaption of a DVM for a particular study system nor simulating the past transient vegetation dynamics with a DVM is newsworthy anymore, unless novel methods are introduced in their application. Which brings us to the novelty of this paper: the coupling (or rather, the preparation towards coupling) of a DVM to a landscape evolution model. However, the manuscript fails to describe the steps that makes this coupling possible and discuss the approach with sufficient detail.

For example, in the final paragraph authors claim "In summary, we suggest that coupling state-of-art dynamic vegetation modelling with landscape evolution models has great potential for improving our understanding of the evolution of landforms" whereas this is not the essence of the current text. The text currently merely reports the simulated vegetation composition and cover over the last 21K years in fairness to the second part of its title. However, as I mentioned (although maybe not for Coastal Cordillera of Chile) this has been done multiple times by now. What distinguishes this study from such previous studies in terms of its potential to improve landscape evolution models and estimates of denudation rates?

Is it the improved ability of a regionally parameterized DVM to reproduce regional vegetation? Which is, by the way, only evaluated qualitatively and only through visual comparison, whereas more quantitative approaches are available in the literature. Then, comparison of results with a globally parameterized version is also necessary. Is it the importance of using a model that explicitly simulates the hydrological cycle and outputs runoff, evaporation, evapotranspiration directly, say, instead of indirect calculations of these variables from simulated vegetation cover? Then, comparison with such indirect calculations and their evaluation against data is necessary. Is it the introduction of landforms in a DVM and getting the topography as close as possible (P.8, L.5)? Then, the version with landforms should be tested against a version without, at least at the four sites. Besides, in my opinion, this novelty itself is not sufficiently explained, please see specific comments below.

Although the questions are raised in the introduction, what makes a DVM useful over the simplified vegetation representations used so far in landscape evolution models, or a particular DVM more useful than others, for its coupling with landscape evolution models is left untested and unanswered in the paper. And some of the relevant bits of information (e.g. P.10, L.16-22) are buried deep.

I invite authors to rethink about their last sentence "The current simulations are an important step towards applying such a coupled model to the study area of EarthShape" and their main conclusions listed few lines above that. None of their main conclusions is about or linked back to the importance or potentials of such coupling. This paper should clearly convey how much more we learn about vegetation from DVMs -or from your particular version of a DVM- that is crucial to know for improved predictions of landscape evolution, that otherwise we could not know.

SPECIFIC COMMENTS

Introduction

P2., L.21: Could you provide examples of vegetation processes influencing erosive processes on comparable temporal scales?

P2., L.24-25: Please provide citation for the 120 ppm $CO_2$ compensation point.

Background

This is a good place to include another short section to inform the reader about climate-vegetation interplay on erosive processes in Chile so that they can follow interpretation of results later. What does high precipitation-high vegetation cover or low precipitation-low vegetation cover lead to? Are types of vegetation rooting strategies relevant? Basically, guide readers to pay attention to certain aspects in the coming sections.

Methods

Eqn (1) is not referenced in the methods, and "n" and "A" are not mentioned.

Landform classification

If I understand correctly, the landforms are affecting simulations via temperature, radiation and soil depth, right? And the temperature difference is calculated with a fixed lapse rate (P.5, L.15)? Whether this is a value authors calculated or obtained from literature is not clear. How were the adjustments to the radiation received by a landform made using the slope and aspect (P.5, L.16)? There is no further explanation/equation. Ideally, a script could be provided for reproducibility of this section. Could you elaborate why no adjustment was applied to the precipitation? Could you also report how many simulation entities (grid cells/landforms) you started and ended up with after landform classification, and how much it would be different if you were to statistically downscale all the grid cells to obtain the same spatial scale? The contrast might help highlight the strength of this approach.

Table 1: Please provide what subscripts (e.g. i-t-m) stand for here as well.

P.6, L.7: almost repeating information with P.5, L.27-28.

P.6, L.17: no further information is provided about how the downscaling and bias-correction was performed. If the authors followed a previous study, please cite. Otherwise, please provide sufficient information or scripts for its reproduction.

P.8, L.24-26 and Figure 7: Authors use statements like general / strong correlation, but do not report any metric like correlation coefficient. Please provide numerical comparisons. Are there statistically significant differences in these relationships between periods or between biomes?

P.8, L.35: A low hanging fruit for authors would be to compare transient vegetation dynamics for a single landform to an averaged grid cell version (as opposed to re-running simulations without landforms to test the extent of improvement provided by landform approach), and discuss the importance of resulting differences for erosive processes.

Discussion

P.10, L.3: Comparison of model simulations to observational PD vegetation should
have come by now. Ideally, right after section 4.1.

Most of the section 5.1 can be moved to results.

P.10, L.30-34: Seems like something to tackle with landforms. I.e. Why not apply a
correction for precipitation?

Section 5.2: Although it is good that authors provide a comprehensive comparison of
past vegetation to proxy data, this discussion is again not linked back to the big picture
of why this is important for a potential coupling of vegetation-landscape modeling. For
instance, authors could cite some palaeohydrological study and contribute its interpre-
tations with their findings.

Or they could discuss their findings in relation to landscape processes, such as (P.9,
L21) "Despite pronounced changes in vegetation composition, FPC only increases
from approx. 51% (LGM) to 59% (PD)", (P.8, L24-25) "While the general correlation
of FPC to precipitation can also observed for LGM, the variability in mesic and xeric
woodlands appears to be larger." How could these translate to erosive processes?
Could other simplified vegetation representations provide similar information or are
these where advantages of DVMs come into play?

In the discussion, authors could further discuss what we have learned over or built
upon Collins et al. (2004) and Istanbulluoglu and Bras (2005) as these studies were
mentioned in the introduction (P.2, L.6)

Conclusion

P.13, L.16-17: How can we know? There was not a single comparison to such studies
with simplified vegetation representation in the discussion.

P.13, L.22: Could authors elaborate on what their planned next steps are?

Could the authors summarize their findings into a brief roadmap/checklist for the community? Say, if I have a DVM that I would like to couple with a landscape model, what advice should I follow in the light of this study?

———————————————

---

## Editor Comment (EC1) · R. Hodge (Editor) · 10 May 2018

The reviewers agree that this is a well-executed and well written paper. The two reviewers identify very similar areas that the authors need to address when replying to these reviews. They identify that the main advance of this work is the inclusion of 'landforms' into the dynamic vegetation model, but both think that the effect of this change needs further exploration and more quantitative validation. The reviewers make some suggestions for how this might be undertaken (e.g. further analysis, comparison with literature or additional model runs), and I would encourage the authors to think about

how they can address this.

The overall aim of this pair of papers is to move forwards towards coupling a dynamic vegetation model and a landscape evolution model. This would be a notable advance, however this paper also needs to stand on its own. A more robust analysis of the new landform component of the model would help to achieve this by demonstrating the novelty and significance of this work in its own right, rather than just in the context of the larger project.

---

## Author Response (AR1)

**RESPONSE TO REVIEWS – Esurf Manuscript: esurf-2018-14**

**Effect of changing vegetation on denudation (part 1): Predicted vegetation composition and cover over the last 21 thousand years along the Coastal Cordillera of Chile**
**By: Werner et al.**

Responses in blue, original comment in black.

**Response to Associate Editor: Rebecca Hodge (Univ. Durham)**

The reviewers agree that this is a well-executed and well written paper. The two reviewers identify very similar areas that the authors need to address when replying to these reviews. They identify that the main advance of this work is the inclusion of 'landforms' into the dynamic vegetation model, but both think that the effect of this change needs further exploration and more quantitative validation. The reviewers make some suggestions for how this might be undertaken (e.g. further analysis, comparison with literature or additional model runs), and I would encourage the authors to think about how they can address this.

The overall aim of this pair of papers is to move forwards towards coupling a dynamic vegetation model and a landscape evolution model. This would be a notable advance, however this paper also needs to stand on its own. A more robust analysis of the new landform component of the model would help to achieve this by demonstrating the novelty and significance of this work in its own right, rather than just in the context of the larger project.

Dear Prof. Hodge:

We would like to thank the referees and the editor for the thorough review, the positive feedback and the suggested improvements for our submitted manuscript. We do follow the points raised and hope that our revision addresses them in full – making the manuscript more useful and interesting to the readers. In the few instances where we disagree with a reviewers' comment we reason why we do not think the implementation of the requested addition would be a good idea (often a substantial extension of the manuscript or a shift of focus from the original manuscript intent).

In particular we implemented the following larger changes to the manuscript to address both reviewers' concerns:

- We added a new Appendix B that illustrates the implemented landform modification approach and details the modifications of site conditions that lead to the observed variability of simulated FPC at the landform level. Furthermore, we added new figures S4 and B1 that showcases the difference between default and landform simulation mode for the four EarthShape focus sites and the conceptual design and added a new section to the discussion (5.1). We hope that this further underlines the novelty of the presented research

- In addition, we improved Fig. 1, and Table 1 and 3 as requested

- As requested by both reviewers we restructured the results and discussion sections and the manuscript should now be structured more clearly
- The climate and vegetation sections were improved and linked to the updated Table 1
- We added quantitative analysis to the evaluation of simulated FPC and also added statistical analysis where appropriate
- We also tried to improve the document to explain why our chosen approach is useful - and in fact a necessity - for our future model coupling stage. In addition, we linked to aspects of landscape evolution and erosion throughout the manuscript as suggested by Reviewer 2. We hope that these changes better convey our strategy and link to our upcoming work of a fully-coupled DVM-LEM model

Attached to this document you find our line-by-line response to the referees' comments. At the end of the document the revised manuscript with track-changes is attached.

Sincerely,
Christian Werner (on behalf of all authors)

**Response to RC1**

We thank the anonymous reviewer for the kind words and positive assessment of the submitted manuscript. Below we first reply to the two major points raised and follow with a detailed line-by-line response to the minor comments.

Major comment 1: Manuscript section organization and improvement of validation

It is a bit strange to find the validation of the model near the end of the paper, in the Discussion section (Evaluation of predicted PNV), and furthermore this validation should be more quantitative. This section should come much earlier in the paper, probably at the beginning of the Results section. Other parts of the Discussion section could also be transferred to the Results section (but maybe near the end of the section): the sensitivity tests performed in subsections 5.3 and 5.4; these are results and not just discussion. More importantly, I feel that the validation of subsection 5.1 should be improved. As it is now, it is limited to a visual comparison of biome maps, as well of the foliage projection cover map predicted by the model with the vegetation cover map derived from MODIS data. You should provide at least some statistics for this comparison. Also, there is no validation of runoff, while it is reported as a very important variable for the landscape evolution model.

The organization of result and discussion section could have been clearer (as also noticed by Reviewer 2). To improve the readability we restructured parts of the result and discussion section. The evaluation of PNV distribution (biome locations) for PD conditions was merged into section 4.2. The comparison of simulated foliar projected cover with MODIS-based vegetation cover estimates was combined in result section 4.3. Furthermore, we introduced a new section that describes the results from our $CO_2$ sensitivity simulation experiment (formerly covered in 5.3). We did however keep the section describing the results from an alternative paleo climate dataset (ECHAM5) in the discussion section, as its not part of the TraCE-21ka-based simulation

ensemble of this paper and was merely included to illustrate the importance of climate data for DVM simulation results.

Furthermore, we added more quantifications of discrepancies between modelled and observed FPC to the relevant section (mean absolute error, MAE). However, we want to note that the evaluation of both FPC and biome distribution is complicated for multiple reasons (which was also our reason for our simplistic visual characterization of the results). First, mismatches of biome classifications can be the result from multiple causes. The biome map given in Fig. 1a is based on an aggregated version of the floristic units (Luebbert and Pliscoff, 2017) that do not necessarily align with a PFT- and physiological-based classification. Also the large number units and their often very specific mosaic of co-occurring species make a clear separation into major biomes (that are based on very few characteristic PFTs) often impossible. For instance, Matorral and Sclerophyllous Woodland were not easily separable based on these units.

A thorough runoff validation is unfortunately not possible in the scope of this study (esp. due to a lack of spatially explicit runoff data). However, as Part II (Schmid et al., 2018) only uses FPC yet, we opted to postpone this issue to the planned future publication of the coupled DVM-LEM model which will include FPC and runoff effects. We thus added a paragraph that acknowledges this limitation and we revised all sections that concern runoff results. We still want to include runoff data is it will be considered in future work (as mentioned).

We added a quantitative assessment of simulated FPC against MODIS vegetation cover (Sect. 4.3) and correlation analysis when appropriate.

Major comment 2: Request for more details of landform effect in simulations

It is not clear to me that this study fulfils the objective of demonstrating the ability of a dynamic vegetation model to produce results useful for the spatial and temporal scales of landscapes evolution models. The authors develop a landform sub-model, but they do not really test it. They just present some transient evolution for one landform in each of four given model pixels. However, we do not know how far the use of the landform sub-model improves model prediction. It would be useful to illustrate the landform results for a given pixel (in addition to the altitudinal profiles that I guess use the landforms).

The reason to present only a single landform in Fig. 9 was to clearly show the PFT transitions with time. If one reports the average PFT composition of all landforms in a grid cell these trends are masked/ harder to showcase (but all simulations reported in the manuscript are run using the landform approach).

Another reason for originally not including the full details of the implemented landform approach was that we assumed this would bring the document to an unfeasible length. However, since both reviewers ask for more details of the approach we now include a new Appendix C that explains the conceptual approach and the modifications of site conditions for the landforms in detail. The landform simulation mode in general improved the simulation especially in semi-arid to Mediterranean locations as a heterogeneous landscape representation (i.e. valley landforms and higher altitude locations with lower temperatures) generally led to higher vegetation cover (access to more soil water, lower temperature with less water-limiting conditions).

We included a new figure to the supplement that highlights the differences in simulated FPC for individual landforms, the area-weighted average FPC from those landforms and the original LPJ-GUESS simulation for the last 2000 years of the transient simulation runs of the four focus sites. As can be observed (Fig. S4a), the implemented landform approach has differing effects for the four focus sites. The area-weighted average FPC at site Sta. Gracia closely resembles the results from the default simulation mode (apart from landform 810 of high altitudes that only covers 1.7% of the grid cell area, Fig. S4b). Average-landform and default results for site La Campana also differ only marginally. However, here a set of landforms of higher altitudes has a substantially lower FPC than the average (~ - 15%). Variation at the hyper-arid site Pan de Azucar is lower (as is the FPC), but generally higher than the default simulation (which aligns with MODIS observations for the site, where the default model underestimates satellite-observed cover). The higher FPC in the new model setup is likely a result of deeper soil profiles of flat and valley landforms that allow a longer water storage versus the default uniform 1.5m soil assumption of LPJ-GUESS. The larger variability of FPC at site Nahuelbuta can be attributed to the rel. Large altitudinal variation in this 0.5x0.5 grid cell (coast to mountainous terrain) and is thus likely a temperature effect.
From site explorations (see Fig. 1 for impressions from the four locations) it is clear that vegetation is not distributed uniformly in the landscapes. Thus, the higher spatial diversity of simulated FPC in the landform approach can be assumed to more realistically describe true FPC in these areas and should thus also lead to more non-uniform erosion rates when FPC is spatially disaggregated on a high-resolution landscape utilized by an LEM.

We added a section (5.1) to the discussion section that points at these results.

Also the results of simulations with the landforms should be compared to those obtained with a model without landforms. How far does it improve the comparison with observed vegetation, or with MODIS vegetation cover? How far the results are affected by the adjustment of radiation for slope and aspects, or by the change in soil depth from the valley to the mountain slope or ridge? Landform modelling is a novel aspect put forward by this paper. So, it is important to discuss it more fully.
We acknowledge that a thorough analysis of individual effects on simulated FPC would be interesting, but we deem this outside the scope of this manuscript as it would lead to a substantial extension of the paper which is already very long. However, we added section that discusses the observed differences to default model simulations (see answer above).

Minor comments

P. 2 lines 15-20: this paragraph provides a review of the use of DGVMs for paleoclimatic applications. They, however, mostly refer to studies performed with LPJ-GUESS. Please, please provide also examples of studies performed with other DGVMs.
LPJ-GUESS related paper did indeed dominate this section due to the widespread use of the model for these kind of simulations. We added the references Bragg et al., 2013 (BIOME4 model), Cowling et al., 2008 (LGM to PD simulation study for Africa using TRIFFID), Hopcroft et al., 2017 (a multi model study for the Holocene Sahara greening) to provide results from other models. In addition, we replace Shellito and Sloan (2006) Part 1 with the companion

paper Part 2 since it investigates the possibilities DVM in more detail (the authors use the NCAR LSM-DGVM).
Furthermore, we added Snell et al. (2014) in the introduction paragraph to DGVMs as it is a nice review paper for readers new to the field (the authors also discuss multiple models and their strong points and weaknesses).

Bragg, F. J., Prentice, I. C., Harrison, S. P., Eglinton, G., Foster, P. N., Rommerskirchen, F., and Rullkötter, J.: Stable isotope and modelling evidence for $CO_2$ as a driver of glacial–interglacial vegetation shifts in southern Africa, Biogeosciences, 10, 2001-2010, https://doi.org/10.5194/bg-10-2001-2013, 2013

Cowling, S. A., Cox, P. M., Jones, C. D., Maslin, M. A., Peros, M., and Spall, S. A.: Simulated glacial and interglacial vegetation across Africa: implications for species phylogenies and trans-African migration of plants and animals. Global Change Biology, 14: 827-840, doi:10.1111/j.1365-2486.2007.01524.x, 2008.

Hopcroft, P. O., P. J. Valdes, A. B. Harper, and D. J. Beerling (2017), Multi vegetation model evaluation of the Green Sahara climate regime, *Geophys. Res. Lett.*, *44*, 6804–6813, doi:10.1002/2017GL073740.

Shellito, C. J. and Sloan, L. C.: Reconstructing a lost Eocene Paradise, Part II: On the utility of dynamic global vegetation models in pre-Quaternary climate studies, Glob. Planet. Change, 50(1), 18-32, doi: 10.1016/j.gloplacha.2005.08.002, 2006.

Snell, R. S., Huth, A. , Nabel, J. E., Bocedi, G. , Travis, J. M., Gravel, D. , Bugmann, H. , Gutiérrez, A. G., Hickler, T. , Higgins, S. I., Reineking, B. , Scherstjanoi, M. , Zurbriggen, N., and Lischke, H.: Using dynamic vegetation models to simulate plant range shifts. Ecography, 37: 1184-1197. doi:10.1111/ecog.00580, 2014.

P. 4, line 25: "We approximate the fraction A of the land surface . . ." instead of "We approximate the land surface ..."
Corrected

P. 4, line 37: Field capacity looks strange here. This would mean that a bucket approach is used in both layers. However, since drainage is not possible below field capacity (this is its definition), it would mean that subsurface runoff and percolation rate through the second layer are always zero in your model. Please check
LPJ-GUESS uses indeed a bucket model (Gerten et al., 2004; Smith et al., 2014; Seiler et al., 2015). The relevant section from Gerten et al., 2004 p254: "The model diagnoses surface runoff (R1) and subsurface runoff (R2) from the excess of water over field capacity of the upper and the lower soil layer, respectively. In addition, the amount of water percolating through the second soil layer is assumed to contribute to subsurface runoff (…)"
Baseflow is not explicitly mentioned in Gerten et al., 2004, however Seiler et al. 2015 state: "Precipitation enters the soil until the upper layer is saturated, while any additional precipitation is lost as surface runoff. Soil water evaporates from the upper 20cm, depending on potential

evaporation and soil water content. Soil water percolates from the upper to the lower soil layer, until the lower soil layer is saturated, in which case excess water is lost as drainage. Water contained in the lower soil layer can leave the soil as baseflow at a given rate."

We rephrased the paragraph to: ". In LPJ-GUESS water enters the top soil layer as precipitation until this layer is fully saturated (excess water is lost as surface runoff and evaporation removes water from a 20cm sub-horizon of the top layer). During precipitation days, water can percolate from the top to the lower layer until the lower layer is saturated (excess water is lost as drainage). In addition, water of the lower layer can drain as baseflow with a fixed drainage rate (Gerten et al., 2004; Seiler et al., 2015). The model does neither consider lateral water movement between grid cells nor routing in a stream network (in this study we report the surface runoff component only)."

Seiler, C., R. W. A. Hutjes, B. Kruijt, and T. Hickler (2015), The sensitivity of wet and dry tropical forests to climate change in Bolivia. *J. Geophys. Res. Biogeosci.*, 120, 399–413. doi: 10.1002/2014JG002749.

P. 5, line 15: you use a constant average lapse rate of 6.5∘C/km, whereas the lapse rate could significantly vary, especially in desert areas where it could tend towards the dry adiabatic value of 9.7∘C/km. Moreover, other climate variables can change significantly with elevation in mountain areas, such as precipitation, cloudiness and air relative humidity.
We use 6.5 °C/km as it is an accepted global standard average value used by the climate science community. The reason is that this value is close to the global average, and is also the defined lapse rate in the International Standard Atmosphere (ISA) (e.g. Vaughan 2015). However, we acknowledge that the lapse rate varies a lot in space and time over multiple time scales (ranging from sub-daily to climatological). The high (spatial and temporal) variability is attributed to many features, associated with both atmospheric therodynamics and dynamics, e.g. radiative conditions, moisture content and large-scale atmospheric circulation. Hence, while a higher lapse would potentially be a better approximation for drier site (e.g. Pan de Azucar), this might not be the case for other sites with different atmospheric conditions. In addition, in our case the lapse rate correction is applied for the surface air temperature, which means that we would need to account for the surface conditions (e.g. vegetation type and potential snow cover) when estimating the lapse rate. To study the behavior of the near-surface lapse rate would require details of the atmospheric boundary layer on sub-daily to seasonal time scales. Although this would be an interesting exercise on its own, it is well outside the scope of this study. Another complicated issue is that we are dealing with paleo conditions. Unfortunately, there are no observational proxy records of past lapse rates. However, it is likely to believe that past lapse rates were different compared to the present because of differences in the atmospheric circulation, as well as radiative and surface conditions. While a value close to the dry adiabat (7-9 C/km) of the near-surface lapse rate might be a good approximation for present deserts, it is not certain that this is true also for past climates with different insolation conditions. For example, during episodes of lower insolation it is possible that the surface would be significantly cooler on average, and hence force the near-surface lapse rate toward lower values. Hence, since we cannot account for all possible uncertainties related to the spatial and temporal variations of the lapse rate, we decided to use one recognized value (the standard lapse rate) for all the sites in the

study. However, we did include some of this discussion as well as potential implications of the chosen lapse rate in the revised manuscript.

Vaughan, W. W.: BASIC ATMOSPHERIC STRUCTURE AND CONCEPTS, Standard Atmosphere, 12-16, 2015.

P. 6, section 3.4: for PD, you use the 1960-1989 period. Does the atmospheric CO2 for PD correspond to the mean CO2 during this period? If so, it is significantly larger than the Holocene mean value and it is thus necessary to perform a pre-industrial simulation in addition to PD, in order to separate the CO2 and the climate effect in the difference between PD and MH.

We use annual $CO_2$ concentrations throughout the simulations (transient and time-slice simulations) and the preceding spin-up periods. Indeed, this concentration is higher than the Holocene average, but we fail to see how this makes a Preindustrial time-slice run necessary. We assess the impact of atmospheric $CO_2$ concentrations in the LGM evaluation runs (identical temperature/ precipitation regime; modified $CO_2$ concentration), but do not aim to separate temperature and CO2 effects on for PD and MH periods. We did however refer to the relevant paragraph in the results section and mention a possible effect of $CO_2$ in the observed differences between FPC for various time slices (MH – PD).

P. 7, line 20: "the Deciduous 'Maule' Forest occurs as total rainfall decreases and rainfall seasonality increases." According to table 1, the 'Maule' Forest is a temperate forest made of temperate summergreen trees, not raingreen trees. So, we would expect that it is the seasonality of temperature and not rainfall that determines the occurrence of these trees. Please be more precise on the processes that link this forest to rainfall seasonality.

The reviewer is correct that the TeBS PFT variants that define the DMF biome are of summergreen phenology (species: *Nothofagus glauca*, *N. obliqua, N. alessandrii*). Especially *N. obliqua* is a very versatile species in the mesotemperate climate and covers large areas (Amigo & Rodríguez Guitián, 2011). Reviewer 2 is also correct that the sentence was not very clearly worded as we only mentioned that they occur north of the Valdivian evergreen forests (that receive higher rainfall that the areas where TeBS dominate). However, clearly, temperature and temp. seasonality is a driving factor for the emergence of these species at these latitudes. We thus rephrased the sentence to make this clearer.

P. 8 lines 19-21 and table 3: FPC in the south is lower during MH than at PD. Why? According to Fig. 3, in the south, the climate is wetter and colder during MH. We would expect larger FPC. Is the difference due to CO2, which is larger at PD? This needs to be commented.

According to the TraCE climate data, average MH temperature at latitudes 45-53 S was 0.5 deg C colder then PD. Precipitation totals were however higher from 35-46 S and lower for the southern areas (47-54). A reduced FPC for latitudes south of 45 (Fig. 6b) thus align more with the temperature difference. Furthermore, the PFT distribution maps of MH and (Fig. S1) and PD (Fig. 4) indicate that most reduction of FPC might be attributed to a smaller extent of temperate broadleaf evergreen PFTs (TeBE_tm, TeBE_itm) due to them being outcompeted on lower temperatures by the boreal PFT types. While lower $CO_2$ concentrations could also lead to reduced FPC as illustrated by the LGM $CO_2$ sensitivity simulations (Fig. 10), we do not observe a general FPC reduction (however $CO_2$ concentration is lower for all areas at MH).

Thus, differences in FPC are likely due to changes in PFT composition (FPC per PFT depends on the PFT properties, the balance between PFT compositions can vary with little changes in environmental envelope and are also a result of successional establishment) and/ or the differences in $[CO_2]$ (as different PFTs can benefit from higher $CO_2$ concentrations in different ways - i.e. relationship of $CO_2$ assimilation and transpiration loss, phenological differences etc.). A clear attribution is not possible but we improved the sentence to avoid confusing the readers.

P. 8 line 35: ". . . between PFTs that might otherwise be lost . . ."
Corrected

P. 10, line 19: "The surface runoff simulated here was found to be consistent with expected patterns" – This is not really true, since no validation of runoff has been made.
We agree with the reviewers. Due to a lack of available data for a thorough large-scale comparison we only provide a descriptive evaluation. However, as we plan a future use of FPC and runoff in a coupled-model we still wanted to include runoff results here for completeness. We changed the sentence to: "The surface run-off simulated here was found to be consistent with the general expected patterns, although a thorough analysis was not possible and will be included in future work." In general we added caution notes in the manuscript whenever we discuss runoff results to make the reader aware of this.

P. 12, line 38: Hickler et al., 2015; Zhu et al., 2016 – Please refer to earlier literature, this has been discussed much earlier by many authors.
We added the classic article by Farquhar et al., 1980 and removed Zhu et al. 2016 to keep the reference list reasonable (we kept Hickler et al., 2015 as it provides a nice summary of the current view on the effect of atmospheric $CO_2$ levels on plant physiology based on observations and the implementation in vegetation models).

Table 1. Please provide, as far as possible, example species for all PFTs
We extended the table with a comprehensive list of example species.

Table 2. It might be interesting to also list in this table the PD biome areal extent from the observed map of figure 1, in order to compare them with the model
While we agree with the reviewer that this addition would be interesting, we did not include this column as not all biome classes of the model setup align with the simplified biome classification presented in Fig. 1. The biome map of Fig.1 was generated by aggregating floristic classification units of Luebert and Pliscoff, 2017. However, the scheme does not allow to easily discriminate for instance between Matorral and Sclerophyllous Woodland. Furthermore, these classifications often list complex topographic units that do not easily translate into the biome system used.

Table 3. According to the legend, the table lists PD absolute values, but relative changes (in % ??) with respect to PD for the LGM and MH. However, the title at the top of the table, runs over three columns, which is misleading, because it suggests that all values are % cover or mm yr-1, i.e., absolute changes. Please revise.
We agree with the reviewer that this was potentially confusing for the readers and added the suggested improvement (column-specific units) to the table.

Figure 1. It might be useful to provide a map of elevation next to the vegetation map
We changed Figure 1 as requested and added the elevation map as a second plot panel (now Fig 1. a, b).

Figure 8. Legend, line 11: "dark grey" instead of "darkgrey"
Corrected

Appendix A. P. 32, line 8: "... completely different shapes..." instead of "...completely shapes.. ."
Corrected

**Response to RC2**

We thank the anonymous reviewer for the general support of this paper. Below we first reply to the general comments raised and follow with a detailed line-by-line response to the specific comments.

General comments

Neither adaption of a DVM for a particular study system nor simulating the past transient vegetation dynamics with a DVM is newsworthy anymore, unless novel methods are introduced in their application. Which brings us to the novelty of this paper: the coupling (or rather, the preparation towards coupling) of a DVM to a landscape evolution model. However, the manuscript fails to describe the steps that makes this coupling possible and discuss the approach with sufficient detail.

Concerning the novelty of our application of the DVM we slightly disagree with the reviewer and would like to emphasize that:
1. The site-specific application of a DVM to understand vegetation changes from the LGM to present is novel. We know of few other studies that have done a calibrated and transient application of this nature.
2. As mentioned in the text, this study covers the German Priority research program EarthShape study areas. Available proxy data for past vegetation in these study areas is limited, and the transient simulations presented here provide a backbone of paleo vegetation predictions that are essential for understanding other elements of EarthShape (e.g. soil formation, nutrient cycling, climate-vegetation interactions with surface processes). Thus, we anticipate this manuscript will very useful to EarthShape participants, and hopefully a broader audience, for a necessary step needed to understand present day observations of biota and surface processes.

Concerning the reviewers comment that the coupling preparation (i.e. the landforms approach) is not well described: we agree, and apologize for the confusion. We have modified the section and also added a more detailed explanation as Appendix C to explain this better. This study mainly provides technical achievements in terms of the transient simulations from LGM to present and downscaling of those results to spatial scales that can be linked to landscape evolution modeling. Our coupling at this point is very basic, and we quite simply use the changes in predicted vegetation cover produced from the DGVM as a guide for the amplitude of change we impose in

the landscape evolution model. For example, in the companion paper by Schmid et al. (2018, this issue) we use the best, currently available, parameterizations for how vegetation influence hillslope and surface water flow processes. Currently, this parameterization requires knowledge of vegetation cover. While this is simplistic, there is no more detailed approach currently available, that can also be scaled to the millennial timescales considered. Our future work will aim towards improving this, but in this set of current papers we start with the simplest approach to evaluate if the vegetation cover changes predicted are even important for surface processes studies. Quite interestingly, these vegetation cover changes we predict are in fact important, and we'll build upon that in the future.
The manuscript text has been updated to reflect the above comment. We hope this satisfies the reviewers concerns and helps to clarify any confusion we caused.

For example, in the final paragraph authors claim "In summary, we suggest that coupling state-of-art dynamic vegetation modelling with landscape evolution models has great potential for improving our understanding of the evolution of landforms" whereas this is not the essence of the current text. The text currently merely reports the simulated vegetation composition and cover over the last 21K years in fairness to the second part of its title. However, as I mentioned (although maybe not for Coastal Cordillera of Chile) this has been done multiple times by now. What distinguishes this study from such previous studies in terms of its potential to improve landscape evolution models and estimates of denudation rates?
Is it the improved ability of a regionally parameterized DVM to reproduce regional vegetation? Which is, by the way, only evaluated qualitatively and only through visual comparison, whereas more quantitative approaches are available in the literature. Then, comparison of results with a globally parameterized version is also necessary. Is it the importance of using a model that explicitly simulates the hydrological cycle and outputs runoff, evaporation, evapotranspiration directly, say, instead of indirect calculations of these variables from simulated vegetation cover? Then, comparison with such indirect calculations and their evaluation against data is necessary. Is it the introduction of landforms in a DVM and getting the topography as close as possible (P.8, L.5)? Then, the version with landforms should be tested against a version without, at least at the four sites. Besides, in my opinion, this novelty itself is not sufficiently explained, please see specific comments below."
We thank the reviewer for highlighting these points, and we've modified the manuscript to address the concerns of the reviewer. In particular we (a) introduce a paragraph where we highlight again why the chosen approach is superior to existing methodologies (although a true evaluation of the effect of these differences can only be shown when the model actually is coupled and thus has to be addressed in future work), we (b) revisited the relevant sections to clearly state the aims of this paper: (i) develop and evaluate a regional DVM parametrization, (ii) introduce a new approach to efficiently simulated sub-grid heterogeneity and enable future coupling between models of different spatial resolutions, (iii) investigate the temporal succession of biome composition for the four ES focus sites and the resulting changes of vegetation cover and runoff from LGM to PD, (iv) investigate the effect of paleo climate model drivers and changes of atmospheric $CO_2$ on the simulated vegetation and vegetation composition. And (c) we added a quantitative evaluation of simulated FPC (MAE between model results and MODIS data, Sect. 4.3).
We address many of the points by introducing a section illustrating the effect of the new landform component on simulated vegetation cover (Appendix B, Fig. S4) and also explain the

novelty of this approach in more detail (see also response to Reviewer 1). We extended the section explaining why the global DVM setup is insufficient (need for regional PFT adaptation) for this application and also checked all sections discussing the simulated runoff results. Even though the capability of LPJ-GUESS and other DVMs to simulated a hydrological cycle in detail and thus account for the effects of i.e. surface runoff and transpiration on denudation rates makes them in our view a great tool for this research, we did not try to fully investigate the simulated hydrological fluxes in this study as a) part II (Schmid et al., 2018) currently does not use surface runoff in the first simulations and b) little data is currently available to thoroughly check the results. For reasons of completeness and the interested readers we still report those numbers here, although the major focus if this manuscript rests on the simulation of vegetation cover. However, the future coupled model setup will include surface runoff effects on landscape evolution and we intent to then also include a full evaluation of those model results. To make this clearer to the reader, we carefully checked all sections where runoff is reported and added cautious notes. We also added a section that illustrated the effect of the new landform mode on FPC for the four ES sites as requested (Section 5.1)

Although the questions are raised in the introduction, what makes a DVM useful over the simplified vegetation representations used so far in landscape evolution models, or a particular DVM more useful than others, for its coupling with landscape evolution models is left untested and unanswered in the paper. And some of the relevant bits of information (e.g. P.10, L.16-22) are buried deep.
I invite authors to rethink about their last sentence "The current simulations are an important step towards applying such a coupled model to the study area of EarthShape" and their main conclusions listed few lines above that. None of their main conclusions is about or linked back to the importance or potentials of such coupling. This paper should clearly convey how much more we learn about vegetation from DVMs - or from your particular version of a DVM - that is crucial to know for improved predictions of landscape evolution, that otherwise we could not know."
We did expand the discussion (5.1) to discuss the effects of landforms on FPC and how this is useful for a future coupling to a LEM.
We also did change the mentioned section to the following:

(6) We consider the implementation of a landform classification a feasible tool to a) mediate between coarse DVM model resolutions and generally higher resolution LEM with little computational expense and b) to account for sub-grid variability of micro-climate conditions that are otherwise absent from DVM simulations at larger scales

In summary, we suggest that coupling state-of-art dynamic vegetation modelling with LEMs has great potential for improving our understanding of the evolution of landforms as the DVM using the landform approach can approximate spatial heterogeneity observed in the field that otherwise is not represented by standard DVM implementations. The FPC linked to topography structure will likely result in varying denudation rates in the landscape and have thus the potential to influence landscape evolution. The regional model adaptation and illustrated model improvements are an important step towards applying such a coupled model to the study area of EarthShape.

Specific comments

P2., L.21: Could you provide examples of vegetation processes influencing erosive processes on comparable temporal scales?
We added a paragraph illustrating the relationship between vegetation and erosive processes.

"Acosta et al. (2015) showed that [10]Be-derived mean catchment denudation rates are lower for steeper but vegetated hillslopes in the Rwenzori Mountains and the Kenya Rift Flanks than the erosion rates for sparsely-vegetated, lower-gradient hillslopes within the Kenya Rift zone. Jeffery et al. (2014) highlighted that vegetation cover plays a major role in controlling Central Andean topography which links directly to the potential erosion in those areas. On a shorter timescale, Vanacker et. al. 2007 determined that the removal of natural vegetation due to land use change increases sediment yield from catchments significantly, while catchments with high vegetation-cover, natural or artificial, return to their natural benchmark erosion rates after reforestation."

Vanacker, V., von Blanckenburg, F., Govers, G., Molina, A., Poesen, J., Deckers, J., and Kubik, P: Restoring dense vegetation can slow mountain erosion to near natural benchmark levels. Geology, 35(4), 303–306, doi: doi.org/10.1130/G23109A.1, 2007.

P2., L.24-25: Please provide citation for the 120 ppm $CO_2$ compensation point.
This was a typo. The correct value should be 150 ppm. We also added the reference Lovelock and Whitfield (1982).

Background
This is a good place to include another short section to inform the reader about climate-vegetation interplay on erosive processes in Chile so that they can follow interpretation of results later. What does high precipitation-high vegetation cover or low precipitation- low vegetation cover lead to? Are types of vegetation rooting strategies relevant? Basically, guide readers to pay attention to certain aspects in the coming sections.
We did include a brief transitional paragraph as suggested but, in light of the overall length of the manuscript, decided against a longer section. We did however try to expanded relevant parts of the introduction to cover the questions raised.

Methods
Eqn (1) is not referenced in the methods, and "n" and "A" are not mentioned.
Corrected

Landform classification
If I understand correctly, the landforms are affecting simulations via temperature, radiation and soil depth, right? And the temperature difference is calculated with a fixed lapse rate (P.5, L.15)? Whether this is a value authors calculated or obtained from literature is not clear. How were the adjustments to the radiation received by a landform made using the slope and aspect (P.5, L.16)? There is no further explanation/equation. Ideally, a script could be provided for reproducibility of this section. Could you elaborate why no adjustment was applied to the precipitation? Could you

also report how many simulation entities (grid cells/landforms) you started and ended up with after landform classification, and how much it would be different if you were to statistically downscale all the grid cells to obtain the same spatial scale? The contrast might help highlight the strength of this approach.

We acknowledge that the original manuscript was lacking detail in this section and extended the explanation of how the landform-approach alters site conditions and thus vegetation development and cover. As some of these questions were already raised by RC1 we refer to our response given there. In short, the lapse rate for temperature correction for landform elevation differences was based on the International Standard Atmosphere (e.g., Vaughan, 2015).

Two key technical advantages of this approach are that a) we do not have to match the high-resolution of the landscape evolution model when the two models are coupled (which might be wasteful computationally as large sections of the sub-grid topography might share topographic conditions and thus would produce identical outputs). The landforms act as a mediator between the coarse resolution model drivers and help to aggregate topographic characteristics and then disaggregate vegetation cover. Further, b) we can keep the general model infrastructure (i.e. model drivers and resolutions) as is and do not have to create a separate down-scaled driving dataset.

We added a new Appendix C that illustrates the concept and implementation of the landforms in LPJ-GUESS in detail.

Vaughan, W. W.: BASIC ATMOSPHERIC STRUCTURE AND CONCEPTS, Standard Atmosphere, 12-16, 2015.

Table 1: Please provide what subscripts (e.g. i-t-m) stand for here as well.
We added the extra abbreviation information in the caption as suggested.

P.6, L.7: almost repeating information with P.5, L.27-28.
We deleted the duplicate sentence on page 5.

P.6, L.17: no further information is provided about how the downscaling and bias-correction was performed. If the authors followed a previous study, please cite. Otherwise, please provide sufficient information or scripts for its reproduction.

Following Hempel et al. (2013), we used an additive bias-correction for the temperature, and multiplicative corrections for the precipitation and downward shortwave radiation (the same technique was also used in e.g. O'ishi and Abe-Ouchi 2011). The reason why we use multiplicative instead of additive corrections for precipitation and radiation is due to the fact that these fields cannot be less than zero. After the bias-correction, the resulting anomalies were interpolated to the ERA-Interim grid using bilinear interpolation. We have clarified all this in the main text.

Hempel, S., Frieler, K., Warszawski, L., Schewe, J., and Piontek, F.: A trend-preserving bias correction - the ISI-MIP approach, Earth Syst. Dynam., 4, 219-236, doi: 10.5194/esd-4-219-2013, 2013.

O'ishi, R., and A. Abe-Ouchi (2011), Polar amplification in the mid-Holocene derived from dynamical vegetation change with a GCM, *Geophys. Res. Lett.*, 38, L14702, doi: 10.1029/2011GL048001.

P.8, L.24-26 and Figure 7: Authors use statements like general / strong correlation, but do not report any metric like correlation coefficient. Please provide numerical comparisons. Are there statistically significant differences in these relationships between periods or between biomes?
We agree with the reviewer that the claim was not justified by hard evidence. We change section and added an analysis of corellation coeffcents between time-slices and selected biomes.

P.8, L.35: A low hanging fruit for authors would be to compare transient vegetation dynamics for a single landform to an averaged grid cell version (as opposed to re-running simulations without landforms to test the extent of improvement provided by landform approach), and discuss the importance of resulting differences for erosive processes.
This is indeed a good idea and we added this to the manuscript (Section 5.1, Fig. S4). See also our response to R1.

P.10, L.3: Comparison of model simulations to observational PD vegetation should have come by now. Ideally, right after section 4.1.
Most of the section 5.1 can be moved to results.
This was indeed a problem in the original manuscript organization and also identified by Reviewer 1. We thus broke this section up and moved descriptions of the model results into the relevant result sections and merged the discussion paragraphs into the main discussion section.

P.10, L.30-34: Seems like something to tackle with landforms. I.e. Why not apply a correction for precipitation?
While a correction of precipitation by adding assumed fog precipitation could be used, the uncertainty in the occurrence and the small scale of this phenomena would add a large uncertainty to our simulations and would also be impossible to validate for past times. Furthermore, these effects are dependent on stratification of the lower atmosphere, the proximity to the coast (and sea surface temperature) and thus beyond reach for a general model parametrization that is intended to be applicable for larger areas and long time periods. We did add a paragraph discussing the omission of a precipitation correction in section 5.1.

Section 5.2: Although it is good that authors provide a comprehensive comparison of past vegetation to proxy data, this discussion is again not linked back to the big picture of why this is important for a potential coupling of vegetation-landscape modeling. For instance, authors could cite some palaeohydrological study and contribute its interpretations with their findings.
Or they could discuss their findings in relation to landscape processes, such as (P.9, L21) "Despite pronounced changes in vegetation composition, FPC only increases from approx. 51% (LGM) to 59% (PD)", (P.8, L24-25) "While the general correlation of FPC to precipitation can also observed for LGM, the variability in mesic and xeric woodlands appears to be larger." How could these translate to erosive processes? Could other simplified vegetation representations provide similar information or are these where advantages of DVMs come into play?
We added a clarification as for why we did a thorough comparison with paleo vegetation record (i.e. to demonstrate the capability of the model to simulated conditions of differencing climatic

conditions and the resulting changes of vegetation composition and cover). However, one result of this study was that unfortunately the global paleo climate dataset TraCE-21ka seems to underrepresent substantial hygric changes observed by proxy records. Therefore, a detailed comparison of our runoff results to palaeohydrological studies is not helpful. We made sure that we mention this data limitation at the relevant sections in the text. We also extended the section by discussing the effects on erosive processes. In general, as mentioned in other responses, we checked that we provide a better link from our study part1 to the general topic of vegetation effects on erosion dynamics.

In the discussion, authors could further discuss what we have learned over or built upon Collins et al. (2004) and Istanbulluoglu and Bras (2005) as these studies were mentioned in the introduction (P.2, L.6)
We are a bit reluctant to add a discussion section about this as we do not have the actual model-coupling in place and thus cannot report findings of the effects (this is work in progress and will be evaluated in detail once we have a fully-coupled model).
However the mentioned papers treat vegetation cover as a cumulative value of total ground cover. Our study shows that for some model domain, even if the cumulative change in vegetation cover may be small, there exists a large change in PFT distribution which should be considered when thinking about the effectiveness of surface processes. Future studies should try to incorporate the individual effects that different PFT's (e.g. differences in LAI lead to different values of rainfall interception, different root densities and distributions lead to different values of soil cohesion etc.) exert on the land surface into the description of their landscape evolution models.
We did highlight advantages of a state-of-the-art DVM in the text over traditional landscape evolution model vegetation representations when appropriate (i.e. fire dynamics, response to changes in $[CO_2]$ (section 5 and 5.4).

P.13, L.16-17: How can we know? There was not a single comparison to such studies with simplified vegetation representation in the discussion.
We rephrased the sentence to: "The simulation also captures vegetation change drivers that are not explicitly represented in simplified vegetation representations used so far in landscape evolution models, such as plant-physiological effects of changing $[CO_2]$, fire dynamics that varies greatly with PFT composition and interaction with soil water resources through different rooting strategies."

P.13, L.22: Could authors elaborate on what their planned next steps are?
We expanded the paragraph with some detailed next steps: "In future work we will implement a two-way coupling of the dynamic vegetation LPJ-GUESS to the landscape evolution model LandLab. LPJ-GUESS will be driven by climate data and produce a continuous dataset of vegetation cover and surface hydrology that is passed to LandLab. Landlab will use this vegetation cover and simulated denudation rates and, in turn, will provide updated topography (and after a landform classification updated areal cover of landforms) and the associated soil depth information to LPJ-GUESS."

Could the authors summarize their findings into a brief roadmap/checklist for the community? Say, if I have a DVM that I would like to couple with a landscape model, what advice should I follow in the light of this study?

Unfortunately we do not see that a general roadmap can be provided, as DVM greatly differ in model composition and process representation. However, one general guidance would be that one needs to bridge the scales of these models. We believe that running DVMs at LEM resolutions is only feasible at the small catchment level. But even in these small-scale studies the duplication of vegetation properties for similar DEM cells and landscape positions seems wasteful and could be avoided by adopting the proposed landform approach allowing for a more efficient simulation.

We added this suggestion as an additional bullet point to the final conclusions: "We consider the implementation of a landform classification a feasible tool to a) mediate between coarse DVM model resolutions and generally higher resolution LEM with little computational expense and b) to account for sub-grid variability of micro-climate conditions that are otherwise absent from DVM simulations at larger scales"

**Effect of changing vegetation on denudation (part 1): Predicted vegetation composition and cover over the last 21 thousand years along the Coastal Cordillera of Chile**

Christian Werner[1], Manuel Schmid[2], Todd A Ehlers[2], Juan Pablo Fuentes-Espoz[3], Jörg Steinkamp[1], Matthew Forrest[1], Johan Liakka[4], Antonio Maldonado[5], Thomas Hickler[1,6]

[1] Senckenberg Biodiversity and Climate Research Centre (SBiK-F), Senckenberganlage 25, 60325 Frankfurt, Germany
[2] University of Tuebingen, Department of Geosciences, Wilhelmstrasse 56, 72074 Tuebingen, Germany
[3] Department of Silviculture and Nature Conservation, University of Chile, Av. Santa Rosa 11315, La Pintana, Santiago RM, Chile
[4] Nansen Environmental and Remote Sensing Center, Bjerknes Centre for Climate Research, Thormøhlens gate 47, N-5006 Bergen, Norway
[5] Centro de Estudios Avanzados en Zonas Áridas (CEAZA), Raúl Bitrán 1305, La Serena, Chile
[6] Department of Physical Geography, Geosciences, Goethe-University, Altenhoeferallee 1, 60438 Frankfurt, Germany

*Correspondence to:* Christian Werner (christian.werner@senckenberg.de)

**Abstract**

Vegetation is crucial for modulating rates of denudation and landscape evolution as it stabilizes and protects hillslopes and intercepts rainfall. Climate conditions and atmospheric $CO_2$ concentration ([$CO_2$]) influence the establishment and performance of plants and thus have a direct influence on vegetation cover. In addition, vegetation dynamics (competition for space, light, nutrients and water) and stochastic events (mortality and fires) determine the state of vegetation, response times to environmental perturbations, and the successional development. In spite of this, state-of-art reconstructions of past transient vegetation changes have not been accounted for in landscape evolution models. Here, a widely used dynamic vegetation model (LPJ-GUESS) was used to simulate vegetation composition/ cover and surface runoff in Chile for the Last Glacial Maximum (LGM), Mid Holocene (MH) and present day (PD). In addition, we conducted transient vegetation simulations from LGM to PD for four sites of the Coastal Cordillera of Chile at a spatial and temporal resolution adequate for coupling with landscape evolution models.

Using a regionally-adapted parametrization, LPJ-GUESS was capable of reproducing present day potential natural vegetation along the strong climatic gradients of Chile and simulated vegetation cover was also in line with satellite-based observations. Simulated vegetation during the LGM differed markedly from PD conditions. Coastal cold temperate rainforests where displaced northward by about 5° and the tree line and vegetation zones were at lower elevations than at PD. Transient vegetation simulations indicate a marked shift in vegetation composition starting with the past-glacial warming that coincides with a rise in [$CO_2$]. Vegetation cover between the sites ranged from 13% (LGM: 8%) to 81% (LGM: 73%) for the northern Pan de Azúcar and southern Nahuelbuta sites, respectively, but did not vary by more than 10% over the 21,000 yr simulation. A sensitivity study suggests that [$CO_2$] is an important driver of vegetation changes and, thereby, potentially landscape evolution. Comparisons with other paleoclimate model driver highlight the importance of model input on simulated vegetation.

In the near future, we will directly couple LPJ-GUESS to a landscape evolution model (see companion paper) to build a fully-coupled dynamic-vegetation/ landscape evolution model that is forced with paleoclimate data from atmospheric general circulation models.

**1. Introduction**

On the macro scale, it has been suggested that sediment yields from rivers exhibit a non-linear relationship with changing vegetation (Langbein and Schumm, 1958). Although this relationship is controversial (e.g. Riebe et al., 2001; Gyssels et al., 2005), previous work highlights that vegetation is likely a first order control on catchment denudation rates (Acosta et al., 2015; Collins et al., 2004; Istanbulluoglu and Bras, 2005; Jeffery et al., 2014). While relatively simple vegetation descriptions have been included in landscape evolution modelling (LEM) studies (Collins et al., 2004; Istanbulluoglu and Bras, 2005), these descriptions do not include explicit representations of plant competition for water, light and nutrients or stand dynamics which are key to determine the progression of vegetation state.

Dynamic Global Vegetation Models (DGVMs) were created as state-of-art tools for representing the distribution of vegetation types, vegetation dynamics (forest succession and disturbances by, e.g., fire), vegetation structure and biogeochemical exchanges of carbon water and other elements between the soil, the vegetation and the atmosphere (Prentice et al., 2007; Snell et al., 2014). Interactions with the climate system have been a special focus, including both transient response to climatic changes and using DGVMs as land-surface schemes of Earth system models (i.e. Cramer et al., 2001; Bonan, 2008; Reick et al., 2013; Yu et al., 2016). DGVMs are instrumental for understanding the impact of future climate change on vegetation (i.e. Morales et al., 2007; Hickler et al., 2012) as well as studying feedbacks between changing vegetation on climate (i.e. Raddatz et al., 2007; Brovkin et al., 2009). In addition, DGVMs have been utilised to better understand past vegetation changes, ranging from the Eocene (Liakka et al., 2014; Shellito and Sloan, 2006) and Late Miocene (Forrest et al., 2015) to the Last Glacial Maximum (LGM; ~21,000 BP) to the Mid Holocene (MH, ~6,000 BP) (i.e. Harrison and Prentice, 2003; Allen et al., 2010; Prentice et al., 2011; Bragg et al., 2013; Huntley et al., 2013; Hopcroft et al., 2017). Using these models, it has been shown that vegetation often responds with substantial time lags to changes in climate (Hickler et al., 2012, Huntley et al., 2013). Such transient changes are likely to influence erosion rates and catchment denudation. Acosta et al. (2015) showed that [10]Be-derived mean catchment denudation rates are lower for steeper but vegetated hillslopes in the Rwenzori Mountains and the Kenya Rift Flanks than the erosion rates for sparsely-vegetated, lower-gradient hillslopes within the Kenya Rift zone. Jeffery et al. (2014) highlighted that vegetation cover plays a major role in controlling Central Andean topography which links directly to the potential erosion in those areas. 
[revised manuscript text omitted]
, radius 300m, Weiss 2001) that classifies the DEM into discrete topographic classes based on a focal neighborhood analysis (here: ridges, mid-slopes, valleys and flats). These classes are then stratified by elevation intervals to finally form the landforms. The average elevation, slope and aspect are then used to adapt the environmental forcing for this landform.

In this study, we adapt the landform surface temperature via the elevation difference of the 0.5°x0.5° grid cell elevation $E_{GC}$ and the average elevation of the high-resolution DEM occupied by a landform ($E_{LF}$) and adjust the temperature with the global lapse rate γ of -6.5 °C km$^{-1}$ (see Eqn. 2).

$$T_{LF} = T_{GC} + \gamma(E_{LF} - E_{GC})$$

Furthermore, we adapt the amount of absorbed radiation based on the landform slope, aspect and time of the year. The solar declination (δ) at any given day in the year (doy: day of year) is calculated in LPJ-GUESS as follows (Prentice et al., 1993, all angles in radians):

$$\delta = -23.4 \times \cos\left(2\pi \times \frac{doy + 10.5}{365}\right)$$

The solar angle at noon is calculated from the latitude (lat) and δ as:

$$A_Z = lat - \delta$$

[revised manuscript text omitted]

---

## Author Response (AR2)

Manuscript: esurf-2018-14

**Effect of changing vegetation and precipitation on denudation (part 1): Predicted vegetation composition and cover over the last 21 thousand years along the Coastal Cordillera of Chile**

**By: Werner et al.**

**Response to Associate Editor: Rebecca Hodge (Univ. Durham)**

Responses in blue, original comment in black.

Dear Mrs. Hodge.

We want to thank you for your suggestions and kind words. In the following we want to address your general comments. Afterwards we give a point-by-point response to the minor comments of reviewer 2 and your PDF notes.
We hope that you find our improved paper suitable for publication.

Kind regards (also on behalf of the other authors),

Christian Werner

**Comments**

I also have some comments that I would like you to consider; the major points are below, and the minor points are in the attached annotated file.

As I see it, the main novelty of this paper is the implementation of the landforms into LPJ-GUESS. However, I think that this aspect of the paper could be more prominent still; although this version is an improvement on the previous one, the details of the landform method are still only presented in an appendix.

Figure S4 is a really useful addition that shows the impact of the landform aspect of the model, but it is not in the main paper. Furthermore, there is not that much in the discussion considering the assumptions/decisions that have to be made in implementing the landform method; for example, what is the justification for the different soil depths? Is that the optimum number of topographic classes?
How do you decide on an appropriate elevation interval? How might these decisions affect your results? Instead the discussion focusses more on the overall model application.

I think that it would help to emphasize the novelty of your work if you could move Fig S4, and maybe Appendix B, into the main paper, along with developing Section 5.1

of the discussion to address some of the decisions that have to be made in developing the landform model.

There is already some overlap between Appendix B and the methods, so not all the appendix text would have to move into the methods. You could then move some other figures (e.g. 10 and 11) into the supplement, and slim down some other sections (e.g. Section 5, maybe 4.3 and 4.4).

Supplement: text in Fig S4b is hard to read.

Indeed, the new landform simulation mode is a key feature of the presented work. However, our primary focus was to demonstrate that the model (in part due to the added improvements) is capable to simulate past vegetation cover for the chosen model domain and is thus suitable for future model-coupling to asses vegetation-landscape evolution dynamics for the climatic gradients captured by the EarthShape focus sites. Furthermore, we wanted to demonstrate the importance of dynamic vegetation (and thus its environmental controls, i.e. atmospheric $CO_2$ and climate forcing) for the simulation outcome and why a dynamic vegetation model should be well suited to respond to these dynamics. Also, simpler vegetation modelling approaches do not capture the effect of changes in $CO_2$ on vegetation. For the stated reason we prefer to not cut back on the sections concerning model outcomes of different $CO_2$ settings and model input drivers (we thus also kept figures 10/11).

The landform model addition is indeed an important improvement for the model and a key outcome of this project with a range of potential useful applications for the DGVM science domain and thus deserves a good illustration. We this improved on presenting this to the reader as suggested by moving details for the landform addition to the main manuscript and also made more pointed references to its importance and effects. For instance, we moved the updated figure S4 to the main manuscript (now Appendix B) and extended the relevant discussion section (addressing your questions about the justification for the landform setup choices). We also highlight this model improvement in the conclusion and abstract.

**Minor points (as positioned in the PDF file)**

P1. Please change the title to be consistent with the part 2 paper: Effect of changing vegetation and precipitation on denudation – Part 1:

Corrected

P2. Can you be more specific about Jeffry et al? This doesn't explain to me what they did/found out.

We changed the sentence too: "Jeffery et al. (2014) investigated how interdependent climate and vegetation properties affect Central Andean topography. They found that mean hill slope gradient correlates most strongly with percent vegetation cover, and that climate influences on topography are mediated by vegetation."

P5. In this new text, it might be useful to specify which rates (infiltration, surface run-off etc) are affected by vegetation cover. You mention this in the previous sentence, but it's not clear whether/how the influences of vegetation applies to LPJ-GUESS.

We changed the sentence too: "In addition, the hydrological cycle is also affected by PFT-specific interception and transpiration rates that are a function of PFT-specific parameters and development stage. Thus vegetation is modulating infiltration via interception (that is a function of vegetation cover) and runoff (via plant uptake and transpiration of water) under the given environmental constraints."

P6. Suggest reinstating some of the deleted text ("While a higher...") that goes on to explain why you don't use site specific lapse rates in this model.

We added the previously deleted clarification again

P10. Fig S4 is a useful addition - move to the main paper?

Originally, we were reluctant to add yet another figure to the main manuscript, however it indeed nicely illustrates the effect of the model change. We thus moved it into the main document (Appendix B).

P12. Discrepancy between what and what?

In order to clarify the sentence, we changed it to: "the largest regional discrepancy of observed and simulated vegetation cover to occur at coastal areas between 30° S to 36° S, (…)"

P12. Would this paragraph be better later on in the paper? It's a bit odd to be discussing the implications of the differences between the model and proxy data, when you don't present these differences until Section 5.2. Or, should the material in 5.2 be moved to earlier in the paper?

We moved the paragraph to the end of the discussion to summarize key points of the previous subsections. It should now be a better experience to the readers.

By 'which' I assume that you mean the landform version of the model, but it's not clear.

We re-phrased to: "The larger vegetation cover simulated sing the landform approach also better aligns with MODIS observations for the site, where the default model underestimates satellite-observed cover."

Can you explain how the TPI is used in this classification; do certain values of the TPI map onto different topographic classes?

We extended to sentence to: "In order to classify landforms, we use the elevation and computed slope and aspect of the grid cells of the high-resolution DEM. Furthermore, aspect and slope are used to compute a topographic position index (TPI) - the difference between elevation and the elevation of surrounding positions (focal radius 300m, see Weiss 2001). The TPI and slope values are then used to

classify positions into discrete topographic classes (here: ridges (TPI > 1 SD), mid-slopes (TPI > -1 SD and TPI < 1 SD and slope >= 6°), valleys (TPI <= -1 SD) and flats (TPI > -1 SD and TPI < 1 SD and slope < 6°)."

So how many landforms does this typically give you per grid cell?

This information is given in the document in section 3.2 page 6. We also provide this information in the mentioned section now: "These classes are then stratified by elevation intervals to finally form the landforms (the total number of landforms per grid cell depends on complexity of the terrain: mean: 16; 25% quantile: 11, 75% quantile: 23)."

Do these values come from anywhere, e.g. papers reporting spatial variation in soil depth?

These values are subjective estimates by the authors and also designed to – on average – retain the previous model assumption of global average soil depth of 1.5 m. In future model coupling actual soil depth information per landform will be provided by LandLab.

**Reviewer 2**

The manuscript by Werner et al. is improved after the revisions. I thank the authors for their detailed responses and discussions. Context is now made clearer, methods are explained in sufficient detail and limitations listed more openly to guide future research in this area. I found the new figures B1 and S4 particularly helpful. I believe the manuscript in this form constitutes a concrete contribution to the scientific literature. I have only minor comments (page and line numbers refer to the marked up version):

We thank the reviewer for the kind words.

P6.L10: From their responses I can see that authors put thought in the lapse rate decision. However, before delving into sophisticated methods, they could have fitted spatial regressions to calculate the lapse rates themselves as a function of topography and other conditions. Although this is also not ideal (statistical modeling might still miss more dynamic relationships), at least authors could have tested whether different lapse rates result in differences in their simulations (e.g. increase/decrease agreement between simulated and observed vegetation).

Also, in their response authors claim they "cannot account for all possible uncertainties related to variations of the lapse rate", but they should. In general, simulations that are done with single sets of parameters/drivers/initial conditions are bound to such criticism. A different choice of lapse rate/parameters/drivers can (most of the time, will) change results. Therefore, models should be run in ensembles and results should be reported with associated uncertainty estimates. These are thoughts for future reference.

We still deem out chosen approach reasonable but acknowledge that a more sophisticated (dynamic) statistical downscaling method could improve the simulations (however, the suggested ensemble approach to develop such a sub-module is beyond the scope of the presented manuscript).

P6.L14: Maybe refer Appendix B here as it explains how the slope and aspect were utilized to adjust the incoming radiation received by the landform.

Added the reference to Appendix B

P6.L24: Do you mean Appendix B?

Corrected

P8.L5: Do you mean Table C1?

Corrected

P9.L18: Townshend et al 2017 is missing from the references.

This was corrected to the correct citation reference Dimiceli et al., 2015

P9.L21: The north-westernmost tip also shows a distinct discrepancy.

We think that the reviewer actually refers to the Andean highlands in the north-easternmost tip of the map and extended the sentence to: "The most distinct regional discrepancy can be observed at coastal areas between 30° S to 36° S and the Andean highlands in the north of the model domain."

P9.L22: How do these values compare with the literature? i.e. does 11.9% MAE count as successful?

We are not aware of an appropriate reference that we can cite here but deem this a good result especially given that a) MODIS observes actual current vegetation cover and not potential vegetation, b) was found to have quality issues at high cover rates (saturation artifact, Sexton et al. 2013, International Journal of Digital Earth), and c) LPJ-GUESS is by design not calibrated to local observations but calculates vegetation composition and properties dynamically.

P9.L24 Typo "tends to underestimate" Also why does LPJ-GUESS tends to underestimate FPC of deciduous PFTs? Is this a paramerization issue or model structural error?

We corrected the typo. The underestimation of deciduous PFTs was observed in different previous studies (i.e. Smith et al., 2014) but not resolved as of yet. The effect is especially observed using the global PFT mode. We assume that the PFTs in the current parametrization have a competitive disadvantage against the evergreen PFTs. It has been suggested that this could be an effect of the smaller nitrogen supply levels for plant physiology (leaf development, fine root growth).

P13.L12: typo, *diverse

Fixed

Manuscript: esurf-2018-14

**Effect of changing vegetation and precipitation on denudation (part 1): Predicted vegetation composition and cover over the last 21 thousand years along the Coastal Cordillera of Chile**

**By: Werner et al.**

**Response to Associate Editor: Rebecca Hodge (Univ. Durham)**

Responses in blue, original comment in black.

Dear Mrs. Hodge.

We want to thank you for your suggestions and kind words. In the following we want to address your general comments. Afterwards we give a point-by-point response to the minor comments of reviewer 2 and your PDF notes.
We hope that you find our improved paper suitable for publication.

Kind regards (also on behalf of the other authors),

Christian Werner

**Comments**

I also have some comments that I would like you to consider; the major points are below, and the minor points are in the attached annotated file.

As I see it, the main novelty of this paper is the implementation of the landforms into LPJ-GUESS. However, I think that this aspect of the paper could be more prominent still; although this version is an improvement on the previous one, the details of the landform method are still only presented in an appendix.

Figure S4 is a really useful addition that shows the impact of the landform aspect of the model, but it is not in the main paper. Furthermore, there is not that much in the discussion considering the assumptions/decisions that have to be made in implementing the landform method; for example, what is the justification for the different soil depths? Is that the optimum number of topographic classes?
How do you decide on an appropriate elevation interval? How might these decisions affect your results? Instead the discussion focusses more on the overall model application.

I think that it would help to emphasize the novelty of your work if you could move Fig S4, and maybe Appendix B, into the main paper, along with developing Section 5.1

of the discussion to address some of the decisions that have to be made in developing the landform model.

There is already some overlap between Appendix B and the methods, so not all the appendix text would have to move into the methods. You could then move some other figures (e.g. 10 and 11) into the supplement, and slim down some other sections (e.g. Section 5, maybe 4.3 and 4.4).

Supplement: text in Fig S4b is hard to read.

Indeed, the new landform simulation mode is a key feature of the presented work. However, our primary focus was to demonstrate that the model (in part due to the added improvements) is capable to simulate past vegetation cover for the chosen model domain and is thus suitable for future model-coupling to asses vegetation-landscape evolution dynamics for the climatic gradients captured by the EarthShape focus sites. Furthermore, we wanted to demonstrate the importance of dynamic vegetation (and thus its environmental controls, i.e. atmospheric $CO_2$ and climate forcing) for the simulation outcome and why a dynamic vegetation model should be well suited to respond to these dynamics. Also, simpler vegetation modelling approaches do not capture the effect of changes in $CO_2$ on vegetation. For the stated reason we prefer to not cut back on the sections concerning model outcomes of different $CO_2$ settings and model input drivers (we thus also kept figures 10/11).

The landform model addition is indeed an important improvement for the model and a key outcome of this project with a range of potential useful applications for the DGVM science domain and thus deserves a good illustration. We this improved on presenting this to the reader as suggested by moving details for the landform addition to the main manuscript and also made more pointed references to its importance and effects. For instance, we moved the updated figure S4 to the main manuscript (now Appendix B) and extended the relevant discussion section (addressing your questions about the justification for the landform setup choices). We also highlight this model improvement in the conclusion and abstract.

**Minor points (as positioned in the PDF file)**

P1. Please change the title to be consistent with the part 2 paper: Effect of changing vegetation and precipitation on denudation – Part 1:

Corrected

P2. Can you be more specific about Jeffry et al? This doesn't explain to me what they did/found out.

We changed the sentence too: "Jeffery et al. (2014) investigated how interdependent climate and vegetation properties affect Central Andean topography.  They found that mean hill slope gradient correlates most strongly with percent vegetation cover, and that climate influences on topography are mediated by vegetation."

P5. In this new text, it might be useful to specify which rates (infiltration, surface run-off etc) are affected by vegetation cover. You mention this in the previous sentence, but it's not clear whether/how the influences of vegetation applies to LPJ-GUESS.

We changed the sentence too: "In addition, the hydrological cycle is also affected by PFT-specific interception and transpiration rates that are a function of PFT-specific parameters and development stage. Thus vegetation is modulating infiltration via interception (that is a function of vegetation cover) and runoff (via plant uptake and transpiration of water) under the given environmental constraints."

P6. Suggest reinstating some of the deleted text ("While a higher...") that goes on to explain why you don't use site specific lapse rates in this model.

We added the previously deleted clarification again

P10. Fig S4 is a useful addition - move to the main paper?

Originally, we were reluctant to add yet another figure to the main manuscript, however it indeed nicely illustrates the effect of the model change. We thus moved it into the main document (Appendix B).

P12. Discrepancy between what and what?

In order to clarify the sentence, we changed it to: "the largest regional discrepancy of observed and simulated vegetation cover to occur at coastal areas between 30° S to 36° S, (…)"

P12. Would this paragraph be better later on in the paper? It's a bit odd to be discussing the implications of the differences between the model and proxy data, when you don't present these differences until Section 5.2. Or, should the material in 5.2 be moved to earlier in the paper?

We moved the paragraph to the end of the discussion to summarize key points of the previous subsections. It should now be a better experience to the readers.

By 'which' I assume that you mean the landform version of the model, but it's not clear.

We re-phrased to: "The larger vegetation cover simulated sing the landform approach also better aligns with MODIS observations for the site, where the default model underestimates satellite-observed cover."

Can you explain how the TPI is used in this classification; do certain values of the TPI map onto different topographic classes?

We extended to sentence to: "In order to classify landforms, we use the elevation and computed slope and aspect of the grid cells of the high-resolution DEM. Furthermore, aspect and slope are used to compute a topographic position index (TPI) - the difference between elevation and the elevation of surrounding positions (focal radius 300m, see Weiss 2001). The TPI and slope values are then used to

classify positions into discrete topographic classes (here: ridges (TPI > 1 SD), mid-slopes (TPI > -1 SD and TPI < 1 SD and slope >= 6°), valleys (TPI <= -1 SD) and flats (TPI > -1 SD and TPI < 1 SD and slope < 6°)."

So how many landforms does this typically give you per grid cell?

This information is given in the document in section 3.2 page 6. We also provide this information in the mentioned section now: "These classes are then stratified by elevation intervals to finally form the landforms (the total number of landforms per grid cell depends on complexity of the terrain: mean: 16; 25% quantile: 11, 75% quantile: 23)."

Do these values come from anywhere, e.g. papers reporting spatial variation in soil depth?

These values are subjective estimates by the authors and also designed to – on average – retain the previous model assumption of global average soil depth of 1.5 m. In future model coupling actual soil depth information per landform will be provided by LandLab.

**Reviewer 2**

The manuscript by Werner et al. is improved after the revisions. I thank the authors for their detailed responses and discussions. Context is now made clearer, methods are explained in sufficient detail and limitations listed more openly to guide future research in this area. I found the new figures B1 and S4 particularly helpful. I believe the manuscript in this form constitutes a concrete contribution to the scientific literature. I have only minor comments (page and line numbers refer to the marked up version):

We thank the reviewer for the kind words.

P6.L10: From their responses I can see that authors put thought in the lapse rate decision. However, before delving into sophisticated methods, they could have fitted spatial regressions to calculate the lapse rates themselves as a function of topography and other conditions. Although this is also not ideal (statistical modeling might still miss more dynamic relationships), at least authors could have tested whether different lapse rates result in differences in their simulations (e.g. increase/decrease agreement between simulated and observed vegetation).

Also, in their response authors claim they "cannot account for all possible uncertainties related to variations of the lapse rate", but they should. In general, simulations that are done with single sets of parameters/drivers/initial conditions are bound to such criticism. A different choice of lapse rate/parameters/drivers can (most of the time, will) change results. Therefore, models should be run in ensembles and results should be reported with associated uncertainty estimates. These are thoughts for future reference.

We still deem out chosen approach reasonable but acknowledge that a more sophisticated (dynamic) statistical downscaling method could improve the simulations (however, the suggested ensemble approach to develop such a sub-module is beyond the scope of the presented manuscript).

P6.L14: Maybe refer Appendix B here as it explains how the slope and aspect were utilized to adjust the incoming radiation received by the landform.

Added the reference to Appendix B

P6.L24: Do you mean Appendix B?

Corrected

P8.L5: Do you mean Table C1?

Corrected

P9.L18: Townshend et al 2017 is missing from the references.

This was corrected to the correct citation reference Dimiceli et al., 2015

P9.L21: The north-westernmost tip also shows a distinct discrepancy.

We think that the reviewer actually refers to the Andean highlands in the north-easternmost tip of the map and extended the sentence to: "The most distinct regional discrepancy can be observed at coastal areas between 30° S to 36° S and the Andean highlands in the north of the model domain."

P9.L22: How do these values compare with the literature? i.e. does 11.9% MAE count as successful?

We are not aware of an appropriate reference that we can cite here but deem this a good result especially given that a) MODIS observes actual current vegetation cover and not potential vegetation, b) was found to have quality issues at high cover rates (saturation artifact, Sexton et al. 2013, International Journal of Digital Earth), and c) LPJ-GUESS is by design not calibrated to local observations but calculates vegetation composition and properties dynamically.

P9.L24 Typo "tends to underestimate" Also why does LPJ-GUESS tends to underestimate FPC of deciduous PFTs? Is this a paramerization issue or model structural error?

We corrected the typo. The underestimation of deciduous PFTs was observed in different previous studies (i.e. Smith et al., 2014) but not resolved as of yet. The effect is especially observed using the global PFT mode. We assume that the PFTs in the current parametrization have a competitive disadvantage against the evergreen PFTs. It has been suggested that this could be an effect of the smaller nitrogen supply levels for plant physiology (leaf development, fine root growth).

P13.L12: typo, *diverse

Fixed

[revised manuscript text omitted]